# Optimizing Attention with Mirror Descent: Generalized Max-Margin Token Selection

## Abstract

Attention mechanisms have revolutionized several domains of artificial intelligence, such as natural language processing and computer vision, by enabling models to selectively focus on relevant parts of the input data. While recent work has characterized the optimization dynamics of gradient descent (GD) in attention-based models and the structural properties of its preferred solutions, less is known about more general optimization algorithms such as mirror descent (MD). In this paper, we investigate the convergence properties and implicit biases of a family of MD algorithms tailored for softmax attention mechanisms, with the potential function chosen as the $p$-th power of the $\ell_p$-norm. Specifically, we show that these algorithms converge in direction to a generalized hard-margin SVM with an $\ell_p$-norm objective when applied to a classification problem using a softmax attention model. Notably, our theoretical results reveal that the convergence rate is comparable to that of traditional GD in simpler models, despite the highly nonlinear and nonconvex nature of the present problem. Additionally, we delve into the joint optimization dynamics of the key-query matrix and the decoder, establishing conditions under which this complex joint optimization converges to their respective hard-margin SVM solutions. Lastly, our numerical experiments on real data demonstrate that MD algorithms improve generalization over standard GD and excel in optimal token selection.

## 1 Introduction

Attention mechanisms (Bahdanau et al., 2014) have transformed natural language processing (NLP) and large language models (LLMs). Initially developed for encoder-decoder recurrent neural networks (RNNs), attention enables the decoder to focus on relevant input segments rather than relying solely on a fixed-length hidden state. This approach became fundamental in transformers (Vaswani et al., 2017), where attention layers—computing softmax similarities among input tokens—are the architecture's backbone. Transformers have driven rapid advancements in NLP with models like BERT (Devlin et al., 2019) and ChatGPT (OpenAI, 2023), and have become the preferred architecture for generative modeling (Chen et al., 2021b; Ramesh et al., 2021), computer vision (Dosovitskiy et al., 2021; Radford et al., 2021), and reinforcement learning (Driess et al., 2023; Chen et al., 2021a). This has led to increased exploration of the mathematical foundations of attention's optimization.

To understand optimization dynamics of attention mechanisms, Tarzanagh et al. (2024; 2023) studied the *implicit bias* of gradient descent (GD) in a binary classification setting with a fixed linear decoder. This bias refers to the tendency of GD to learn specific weight characteristics when multiple valid solutions exist. For example, in linear logistic regression on separable data, GD favors solutions aligned with the max-margin class separator (Soudry et al., 2018; Ji & Telgarsky, 2018). Similarly, Tarzanagh et al. (2023; 2024) propose a model resembling a hard-margin Support Vector Machine (SVM)—specifically, ($\ell_p$-AttSVM) with $p = 2$—which maximizes the margin between optimal and non-optimal input tokens based on their softmax logits. These studies show that as training progresses, the combined key-query weights $W(k)$ of the model increasingly align with the locally optimal solution $W_{\mathrm{mm}}^{\alpha}$—the minimizer of ($\ell_p$-AttSVM) with $p = 2$. Expanding on these insights, Vasudeva et al. (2024a) explores global directional convergence and the convergence rate of GD under specific conditions. Sheen et al. (2024) further extends these findings by relaxing assumptions about the convergence of regularized paths for the $(W_K, W_Q)$ parameterization of the key-query matrix, showing that gradient flow minimizes the nuclear norm of the key-query weight $W = W_K W_Q^\top$.

**Contributions.** While the aforementioned works provide insights into the implicit bias and token selection properties of attention mechanisms, their analyses are limited to simplistic GD models. A broader understanding of general descent algorithms such as the mirror descent (MD) family and

their token selection properties is missing. We address this by examining a family of `MD` algorithms designed for softmax attention, where the potential function is the $p$-th power of the $\ell_p$-norm, termed $\ell_p$-`AttGD`. This generalizes both $\ell_p$-GD (Azizan & Hassibi, 2018; Sun et al., 2022; 2023) and attention GD (Tarzanagh et al., 2024; 2023), enabling the exploration of key aspects of attention optimization via $\ell_p$-`AttGD`.

*Implicit bias of $\ell_p$-`AttGD` for attention optimization.* Building on Tarzanagh et al. (2023); Vasudeva et al. (2024a); Sheen et al. (2024), we consider a one-layer attention model for binary classification. Specifically, given a dataset $(X_i, y_i, z_i)_{i=1}^n$ where $X_i \in \mathbb{R}^{T \times d}$ represents inputs with $T$ tokens, $y_i \in \{\pm 1\}$ is the label, and $z_i \in \mathbb{R}^d$ is the comparison token, we study a single-layer attention model $f(X_i, z_i) := v^\top X_i^\top \sigma(X_i W z_i)$, where $\sigma(\cdot)$ is the softmax function, $W$ is the key-query matrix, and $v$ is a linear decoder. Our goal is to separate a *locally optimal* token $\alpha_i \in [T]$ of each input sequence $X_i$ from the rest via an empirical risk minimization problem (ERM) with a smooth decreasing loss. We extend the SVM formulation in Tarzanagh et al. (2023) to ($\ell_p$-AttSVM), defining a hard-margin SVM using the $\ell_p$-norm instead of the $\ell_2$-norm. The solution $W_{\mathrm{mm}}^\alpha$ separates the locally optimal tokens $(\alpha_i)_{i=1}^n$ with *generalized* maximum margin (see Section 2). Theorem 3 provides sufficient conditions for $\ell_p$-`AttGD` to converge locally, in direction, to $W_{\mathrm{mm}}^\alpha$. Moreover, Theorem 2 shows that $\|W(k)\|_{p,p}$ diverges as $k \to \infty$. These results characterize the implicit bias towards ($\ell_p$-AttSVM) in separating locally optimal tokens, extending previous work to a broader class of algorithms. While Theorem 3 and Tarzanagh et al. (2024; 2023) offer insights into optimization dynamics for $p = 2$, the finite-time convergence rate of $\ell_p$-`AttGD` for selecting locally optimal tokens remains unexplored.

*Convergence rate of $\ell_p$-`AttGD` to the solution of ($\ell_p$-AttSVM).* Theorem 4 establishes convergence rates for $\ell_p$-`AttGD`, showing that the iterates $W(k)$, for large $k$, satisfy that $D_\psi\left(W_{\mathrm{mm}}^\alpha / \|W_{\mathrm{mm}}^\alpha\|_{p,p}, W(k)/\|W(k)\|_{p,p}\right)$ decreases at an inverse poly-log rate, where $D_\psi(\cdot, \cdot)$ denotes the Bregman divergence (Bregman, 1967b); see Definition 1. Despite optimizing a highly nonlinear, nonconvex softmax function, we achieve a convergence rate similar to GD in linear binary classification (Ji & Telgarsky, 2018, Theorem 1.1). Compared to the recent polynomial rate $O(k^{-3/4})$ in (Vasudeva et al., 2024a, Theorem 1) for optimizing attention, our rate is logarithmic and slower, but applicable to standard GD and `MD` for locally optimal token selection. Importantly, we do not require the near-orthogonality of tokens assumption used in Vasudeva et al. (2024a).

*Generalized Max-Margin Solutions and Joint Optimization of $(v, W)$.* We study the joint problem under logistic loss using $\ell_p$-norm regularization path, where (ERM) is solved under $\ell_p$-norm constraints, examining the solution trajectory as these constraints relax. Since the problem is linear in $v$, if the attention features $\bar{X}_i = X_i^\top \sigma(X_i W z_i)$ are separable by their labels $y_i$, $v$ acts as a generalized max-margin classifier (Azizan et al., 2021). Inspired by Tarzanagh et al. (2024; 2023), we show that under suitable geometric conditions, $W$ and $v$ generated by $\ell_p$-norm regularization path converge to their respective max-margin solutions (Theorem 5 in the appendix).

Finally, we provide extensive numerical experiments on real and synthetic data, demonstrating that `MD` algorithms improve generalization over standard GD, excelling in optimal token selection and suppressing non-optimal tokens.

## 2 PRELIMINARIES

**Notations.** Let $N \geq 1$ and $[N] = \{1, 2, \ldots, N\}$. Vectors are denoted by lowercase letters (e.g., $a$), with components $a_i$, and matrices by uppercase letters (e.g., $A$). The minimum and maximum of scalars $a$ and $b$ are $a \wedge b$ and $a \vee b$, respectively. For a vector $v \in \mathbb{R}^d$, the $p$-norm is $\|v\|_p = (\sum_{i=1}^d |v_i|^p)^{1/p}$. For a matrix $M \in \mathbb{R}^{d \times d}$, the $p, p$-norm is $\|M\|_{p,p} = (\sum_{i=1}^d \sum_{j=1}^d |M_{ij}|^p)^{1/p}$. When $p = 2$, these are the Euclidean norm for vectors and the Frobenius norm for matrices. For any two matrices $X, Y$ of the same dimensions, we define $\langle X, Y \rangle := \mathrm{trace}(X^\top Y)$. Throughout, for a differentiable function $f : \mathbb{R}^{d \times d} \to \mathbb{R}$, we define $D_f : \mathbb{R}^{d \times d} \times \mathbb{R}^{d \times d} \to \mathbb{R}$ as

$$D_f(W, V) := f(W) - f(V) - \langle \nabla f(V), W - V \rangle. \tag{1}$$

Asymptotic notations $\mathcal{O}$ and $\Omega$ hide constant factors, and all logarithms are natural ($e$-base).

**Single-head attention model.** Given input sequences $X, Z \in \mathbb{R}^{T \times d}$ with length $T$ and embedding dimension $d$, the output of a single-head (cross-)attention layer is computed as:

$$\mathrm{softmax}(X W_Q W_K^\top Z^\top) X W_V,$$

where $W_Q, W_K \in \mathbb{R}^{d \times d_1}$, $W_V \in \mathbb{R}^{d \times d_2}$ are trainable key, query, value matrices, respectively; $\text{softmax}(XW_QW_K^\top Z^\top)$ is the attention map; and $\text{softmax}(\cdot) : \mathbb{R}^{T \times T} \to \mathbb{R}^{T \times T}$ denotes the row-wise softmax function applied row-wise on $XW_QW_K^\top Z^\top$. Similar to Tarzanagh et al. (2024; 2023), we reparameterize the key-query product matrix as $W := W_QW_K^\top \in \mathbb{R}^{d \times d}$, and subsume the value weights $W_V$ within the prediction head $v \in \mathbb{R}^d$. Suppose the first token of $Z$, denoted by $z$, is used for prediction. Then, the attention model can be formulated as

$$f(X, z) = v^\top X^\top \sigma(XWz), \tag{2}$$

where $\sigma : \mathbb{R}^T \to \mathbb{R}^T$ is the softmax function on vectors.

**Attention-based empirical risk minimization.** We consider a one-layer attention model (2) for binary classification. Consider the dataset $(X_i, y_i, z_i)_{i=1}^n$, where $X_i \in \mathbb{R}^{T \times d}$ is the input with $T$ tokens each of dimension $d$, $y_i \in \{\pm 1\}$ is the label, and $z_i \in \mathbb{R}^d$ is the token used for comparison. We use a smooth decreasing loss function $l : \mathbb{R} \to \mathbb{R}$ and study empirical risk minimization (ERM):

$$\min_{v \in \mathbb{R}^d, W \in \mathbb{R}^{d \times d}} \quad \mathcal{L}(v, W) := \frac{1}{n} \sum_{i=1}^n l\left(y_i v^\top X_i^\top \sigma\left(X_i W z_i\right)\right). \tag{ERM}$$

Throughout, we will use $\mathcal{L}(W)$ to denote the objective of (ERM) with fixed $v$.

The highly nonlinear and nonconvex nature of the softmax operation makes the training problem described in (ERM) a challenging nonconvex optimization task for $W$, even with a fixed $v$. Next, we provide an assumption on the loss function necessary to demonstrate the convergence of MD for margin maximization within the attention mechanism.

**Assumption A.** *Within any closed interval, the loss function $l : \mathbb{R} \to \mathbb{R}$ is strictly decreasing and differentiable, and its derivative $l'$ is bounded and Lipschitz continuous.*

Assumption A aligns with the assumptions on loss functions in Tarzanagh et al. (2024; 2023). Commonly used loss functions, such as $l(x) = e^{-x}$, $l(x) = -x$, and $l(x) = \log(1 + e^{-x})$, satisfy this assumption.

**Preliminaries on mirror descent.** We review the MD algorithm (Blair, 1985) for solving attention-based (ERM). Mirror descent is defined using a *potential function*. We focus on differentiable and strictly convex potentials $\psi$ defined on the entire domain $\mathbb{R}^{d \times d}$. Note that in general, the potential function is a convex function of Legendre type (Rockafellar, 2015, Section 26). We call $\nabla \psi$ the *mirror map*. The natural "distance" associated with the potential $\psi$ is given by the Bregman divergence (Bregman, 1967a).

**Definition 1** (Bregman Divergence). *For a strictly convex function $\psi : \mathbb{R}^{d \times d} \to \mathbb{R}$, the expression $D_\psi(\cdot, \cdot)$ defined in (1) is called the Bregman divergence.*

An important example of a potential function is $\psi = \frac{1}{2} \| \cdot \|_F^2$. In this case, the Bregman divergence simplifies to $D_\psi(W, V) = \frac{1}{2} \|W - V\|_F^2$; For more details, see Bauschke et al. (2017). MD with respect to the mirror map $\psi$ is a generalization of GD where the Bregman divergence is used as a measure of distance. Given a stepsize $\eta > 0$, the MD algorithm is as follows:

$$W(k + 1) \leftarrow \operatorname*{arg\,min}_{W \in \mathbb{R}^{d \times d}} \left\{ \eta^{-1} D_\psi(W, W(k)) + \langle \nabla \mathcal{L}(W(k)), W \rangle \right\}. \tag{MD}$$

Equivalently, MD can be written as $\nabla \psi(W(k+1)) = \nabla \psi(W(k)) - \eta \nabla \mathcal{L}(W(k))$; see Bubeck et al. (2015); Juditsky & Nemirovski (2011). A useful fact about the Bregman divergence is that it is non-negative and $D_\psi(W, V) = 0$ if and only if $W = V$.

**Preliminaries on attention SVM.** Following Tarzanagh et al. (2024; 2023), we use the following definition of token scores.

**Definition 2** (Token Score). *For prediction head $v \in \mathbb{R}^d$, the score of token $X_{it}$ is $\gamma_{it} = y_i v^\top X_{it}$.*

It is important to highlight that the score is determined solely based on the *value embeddings* $v^\top X_{it}$ of the tokens. The softmax function $\sigma(\cdot)$ minimizes (ERM) by selecting the token with the highest score (Tarzanagh et al., 2023, Lemma 2). Using (2), Tarzanagh et al. (2023) defines globally optimal tokens $(\text{opt}_i)_{i=1}^n$, with each $\text{opt}_i$ maximizing the score for $X_{i\text{opt}_i}$. For our MD analysis, we primarily consider locally optimal tokens, as they are more general than globally optimal ones. Locally optimal tokens (Tarzanagh et al., 2024; 2023) are characterized by having scores that surpass those of nearby

tokens, we formalize the notion of nearby tokens later in Definition 3 on locally optimal tokens and support tokens. Intuitively, these are the tokens that locally minimize (ERM) upon selection and can be defined based on *support tokens*. Before presenting the mathematical notion of locally optimal tokens, we provide the formulation of the attention SVM problem. Given a set of (locally) optimal token indices $(\alpha_i)_{i=1}^n \in [T]^n$, Tarzanagh et al. (2023) defines the following hard-margin attention SVM problem, which aims to separate, with maximal margin, (locally) optimal tokens from the rest of the tokens for every input sequence:

$$W_{\text{mm}}^\alpha := \underset{W \in \mathbb{R}^{d \times d}}{\arg\min} \|W\|_F \text{ s.t. } (X_{i\alpha_i} - X_{it})^\top W z_i \geq 1, \text{ for all } t \in [T] - \{\alpha_i\}, i \in [n]. \quad (3)$$

The constraint $(X_{i\alpha_i} - X_{it})^\top W z_i \geq 1$ indicates that in the softmax probability vector $\sigma(X_i W z_i)$, the $\alpha_i$ component has a significantly higher probability compared to the rest, and so these problems solve for a sort of probability separator that has the lowest norm.

**Definition 3** (Globally and Locally Optimal Tokens). *Consider the dataset* $(X_i, y_i, z_i)_{i=1}^n$.

**1.** *The tokens with indices* $\texttt{opt} = (\texttt{opt}_i)_{i=1}^n$ *are called globally optimal if they have the highest scores, given by* $\texttt{opt}_i \in \arg\max_{t \in [T]} \gamma_{it}$.

**2.** *Fix token indices* $(\alpha_i)_{i=1}^n$ *for which* (3) *is feasible to obtain* $W_{\text{mm}}^\alpha$. *Let the support tokens* $\mathcal{T}_i$ *for the* $i^{th}$ *data be the set of tokens* $\tau$ *such that* $(X_{i\alpha_i} - X_{i\tau})^\top W_{\text{mm}}^\alpha z_i = 1$. *The tokens with indices* $(\alpha_i)_{i=1}^n$ *are called locally optimal if, for all* $i \in [n]$ *and* $\tau \in \mathcal{T}_i$, *the scores per Def. 2 obey* $\gamma_{i\alpha_i} > \gamma_{i\tau}$.

It is worth noting that token scoring and optimal token identification can help us understand the importance of individual tokens and their impact on the overall objective. A token score measures how much a token contributes to a prediction or classification task, while an optimal token is defined as the token with the highest relevance in the corresponding input sequence (Tarzanagh et al., 2024; 2023). For illustration, please refer to Figure 1.

## 3 IMPLICIT BIAS OF MIRROR DESCENT FOR OPTIMIZING ATTENTION

### 3.1 OPTIMIZING ATTENTION WITH FIXED HEAD $v$

In this section, we assume the prediction head is fixed and focus on the directional convergence of MD and its token selection property through the training of the key-query matrix $W$. The analysis will later be expanded in Section 3.2 to include the joint optimization of both $v$ and $W$.

We investigate the theoretical properties of the main algorithm of interest, namely MD with $\psi(\cdot) = \frac{1}{p}\| \cdot \|_{p,p}^p$ for $p > 1$ for training (ERM) with fixed $v$. We shall call this algorithm $\ell_p$-*norm AttGD* because it naturally generalizes attention training via GD to $\ell_p$ geometry, and for conciseness, we will refer to this algorithm by the shorthand $\ell_p$-AttGD. As noted by Azizan et al. (2021), this choice of mirror potential is particularly of practical interest because the mirror map $\nabla\psi$ updates become *separable* in coordinates and thus can be implemented *coordinate-wise* independently of other coordinates.

$$\forall \ i,j \in [d], \quad \begin{cases} [W(k+1)]_{ij} \leftarrow \left| [W(k)]_{ij}^+ \right|^{\frac{1}{p-1}} \cdot \text{sign}\left([W(k)]_{ij}^+\right), \\ [W(k)]_{ij}^+ := |[W(k)]_{ij}|^{p-1}\text{sign}([W(k)]_{ij}) - \eta[\nabla\mathcal{L}(W(k))]_{ij}. \end{cases} \quad (\ell_p\text{-AttGD})$$

The algorithm will still incur additional overhead compared to gradient descent, but this overhead is linear in the size of the trainable parameters for both time and space. We discuss this further in the Appendix. In the following, we first identify the conditions that guarantee the convergence of $\ell_p$-AttGD. The intuition is that, for attention to exhibit implicit bias, the softmax nonlinearity should select the locally optimal token within each input sequence. Tarzanagh et al. (2023) shows that under certain assumptions, training an attention model using GD causes its parameters' direction to converge.

This direction can be found by solving a simpler optimization problem, such as (3), which selects the locally optimal token. Depending on the attention model's parameterization, the attention SVM varies slightly. In this work, we generalize (3) using the $\ell_p$-norm as follows:

**Definition 4** (Attention SVM with $\ell_p$–norm Objective). *For a dataset* $\{(X_i, y_i, z_i)\}_{i=1}^n$ *with* $y_i \in \{\pm 1\}$, $X_i \in \mathbb{R}^{T \times d}$, *and token indices* $(\alpha_i)_{i=1}^n$, $\ell_p$-*based attention SVM is defined as*

$$W_{\text{mm}}^\alpha := \underset{W \in \mathbb{R}^{d \times d}}{\arg\min} \|W\|_{p,p}$$

$$\text{subj. to } (X_{i\alpha_i} - X_{it})^\top W z_i \geq 1, \text{ for all } t \in [T] - \{\alpha_i\}, i \in [n]. \quad (\ell_p\text{-AttSVM})$$

Problem ($\ell_p$-AttSVM) is strictly convex, so it has unique solutions when feasible. Throughout this paper, we assume feasibility, which means there exists a matrix $W$ that linearly separates the logits $X_{i\alpha_i}^\top W z_i$ from the logits $X_{it}^\top W z_i$ for all $t \in [T] \setminus \{\alpha_i\}$ and $i \in [n]$. It is worth noting that this is not a strong assumption. For example, under mild overparameterization, $d \geq \max\{T - 1, n\}$, the problem is almost always feasible (Tarzanagh et al., 2023, Theorem 1). Next, we assert that the solution to the ($\ell_p$-AttSVM) problems determines the direction that the attention model parameters approach as the training progresses.

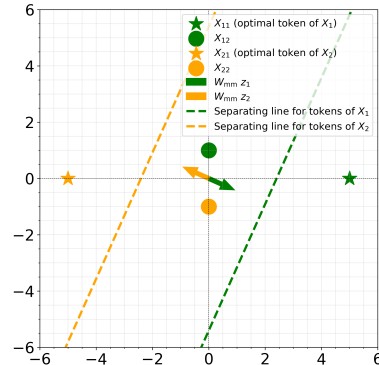

Figure 1: Visualization of ($\ell_p$-AttSVM) for $p = 3$.

**Example 1.** *Consider the matrices $X_1 = [5, 0; 0, 1]$ and $X_2 = [-5, 0; 0, -1]$ with $y_1 = -y_2 = 1$. Let $X_{i1}$ be the optimal token and $X_{it}$ be the others. Solving Problem ($\ell_p$-AttSVM) with $p = 3$ and setting $z_i = X_{i1}$, we obtain the solution $W_{mm} := W_{mm}^\alpha =$* [0.03846, 0; −0.00769, 0]. *Figure 1 illustrates how the optimal tokens $X_{11}$ and $X_{21}$ are separated by the dashed decision boundaries. These boundaries are orthogonal to the vectors $W_{mm} z_i$ and indicate the hyperplanes that separate the sequences based on the optimal token in each case.*

**Theorem 1** ($\ell_p$–norm Regularization Path)**.** *Suppose Assumption A on the loss function holds. Consider the ridge-constrained solutions $W^{(R)}$ of (ERM) defined as*

$$W^{(R)} := \operatorname*{arg\,min}_{W \in \mathbb{R}^{d \times d}} \mathcal{L}(W) \quad \text{subj. to} \quad \|W\|_{p,p} \leq R. \qquad (\ell_p\text{-AttRP})$$

*Then, $\lim_{R \to \infty} W^{(R)}/R = W_{mm}^{opt}/\|W_{mm}^{opt}\|_{p,p}$, where $W_{mm}^{opt}$ is the solution of ($\ell_p$-AttSVM), with $\alpha_i$ replaced by $opt_i$.*

Theorem 1 shows that as the regularization strength $R$ increases, the optimal direction $W^{(R)}$ aligns more closely with the max-margin solution $W_{mm}^\alpha$. This theorem, which allows for globally optimal tokens (see Definition 3), does not require any specific initialization for the $\ell_p$-AttRP algorithm and demonstrates that max-margin token separation is an essential feature of the attention mechanism.

Next, we present the convergence of MD applied to (ERM). Under certain initializations, the parameter's $\ell_p$-norm increases to infinity during training, with its direction approaching the ($\ell_p$-AttSVM) solution. To describe these initializations, we introduce the concept of cone sets.

**Definition 5.** *Given a square matrix $W \in \mathbb{R}^{d \times d}$, $\mu \in (0, 1)$, and some $R > 0$,*

$$S_{p,\mu}(W) := \left\{ W' \in \mathbb{R}^{d \times d} \mid D_\psi \left( \frac{W}{\|W\|_{p,p}}, \frac{W'}{\|W'\|_{p,p}} \right) \leq \mu \right\}, \qquad (4a)$$

$$C_{p,\mu,R}(W) := S_\mu(W) \cap \{W' \mid \|W'\|_{p,p} \geq R\}. \qquad (4b)$$

These sets contain matrices with a similar direction to a reference matrix $W$, as captured by the inner product in $S_\mu(W)$. For $C_{p,\mu,R}(W)$, there is an additional constraint that the matrices must have a sufficiently high norm. We note that $S_{p,\mu}(W)$ and $C_{p,\mu,R}(W)$ reduce to their Euclidean variants as described in Tarzanagh et al. (2024; 2023). With this definition, we present our first theorem about the norm of the parameter increasing during training.

**Theorem 2.** *Suppose Assumption A holds. Let $(\alpha_i)_{i=1}^n$ be locally optimal tokens as per Definition 3. Consider the sequence $W(k)$ generated by Algorithm $\ell_p$-AttGD. For a small enough stepsize $\eta$, if $W(0) \in C_{p,\mu,R}(W_{mm}^\alpha)$ for some dataset-dependent constants $\mu, R > 0$, then we have $\lim_{k \to \infty} \|W(k)\|_{p,p} = \infty$.*

**Remark 1.** *The condition on the stepsize $\eta$ is that it must be sufficiently small so that $\psi(\cdot) - \eta\mathcal{L}(\cdot)$ remains convex for the matrices $W$ along the path traced by the iterates $W(k)$. Specifically, there exists an index $k$ and a real number $r \in [0, 1]$ such that $W = rW(k) + (1 - r)W(k + 1)$. This restriction applies to all theorems in this paper that require a sufficiently small stepsize $\eta$.*

This theorem implies that the parameters will increase and diverge to infinity, justifying the need to characterize the convergence of their direction.

**Theorem 3** (Convergence of $\ell_p$-AttGD)**.** *Suppose Assumption A holds. Let $(\alpha_i)_{i=1}^n$ be locally optimal tokens as per Definition 3. Consider the sequence $W(k)$ generated by Algorithm $\ell_p$-AttGD.*

*For a small enough stepsize $\eta$, if $W(0) \in C_{p,\mu,R}(W_{\mathrm{mm}}^\alpha)$ for some dataset-dependent constants $\mu > 0, R > \exp(2)$, then*

$$\lim_{k \to \infty} \frac{W(k)}{\|W(k)\|_{p,p}} = \frac{W_{\mathrm{mm}}^\alpha}{\|W_{\mathrm{mm}}^\alpha\|_{p,p}}.$$

These theorems show that as the parameters grow large enough and approach a locally optimal direction, they will keep moving toward that direction.

**Theorem 4** (Convergence Rate of $\ell_p$-`AttGD`). *Suppose Assumption A holds. Let $(\alpha_i)_{i=1}^n$ be locally optimal tokens as per Definition 3. Consider the sequence $W(k)$ generated by Algorithm $\ell_p$-`AttGD`. For a small enough stepsize $\eta$, if $W(0) \in C_{p,\mu,R}(W_{\mathrm{mm}}^\alpha)$ for some $\mu > 0, R > \exp(2)$, then*

$$D_\psi \left( \frac{W_{\mathrm{mm}}^\alpha}{\|W_{\mathrm{mm}}^\alpha\|_{p,p}}, \frac{W(k)}{\|W(k)\|_{p,p}} \right) = \mathcal{O} \left( \begin{cases} \frac{\log \log k}{\log k} & \text{if } p > 2, \\ \frac{(\log \log k)^2}{\log k} & \text{if } p = 2, \\ \frac{1}{(\log k)^{p-1}} & \text{otherwise.} \end{cases} \right). \tag{5}$$

Note that, in the left-hand side of (5), there is a dependence on $p$ in the Bregman divergence $D_\psi$ itself as well. Despite optimizing a highly nonlinear, nonconvex softmax function, we achieve a convergence rate similar to GD in linear binary classification (Ji & Telgarsky, 2018, Theorem 1.1) (up to a $\log \log k$ factor). The theorems we prove hinge on the parameter entering the set $W(k) \in C_{p,\mu,R}(W_{\mathrm{mm}}^\alpha)$ with a high enough norm. Since we aim to show the parameter converges in direction to the cone center, $W_{\mathrm{mm}}^\alpha$, we need conditions ensuring the parameters remain in the cone. We formalize this in Lemma 17 and prove that for any $\mu > 0$ and locally optimal tokens $(\alpha_i)_{i=1}^n$ (Definition 3), there exist constants $R, \mu' > 0$ depending on the dataset and $\mu$, such that if $W(0) \in C_{p,\mu',R}(W_{\mathrm{mm}}^\alpha)$, then $W(k) \in C_{p,\mu,R}(W_{\mathrm{mm}}^\alpha)$ for all $k$, meaning the iterates remain within a larger cone; see Figure 2.

For Theorem 2, we show in Lemma 11 that at any timestep $k \geq 0$, the norm of the $W$ parameter evolves in the following manner,

$$\|W(k+1)\|_{p,p}^{p-1} \geq \|W(k)\|_{p,p}^{p-1}$$
$$+ \frac{\eta}{\|W(k)\|_{p,p}} \langle -\nabla \mathcal{L}(W(k)), W(k) \rangle.$$

With the above, to prove Theorem 2, it is enough to show that $\langle -\nabla \mathcal{L}(W(k)), W(k) \rangle$ is positive and large enough to keep the norm increasing to infinity. Specifically, in Lemma 9 we show that there exist dataset-dependent constants $R, \delta, \mu > 0$ such that for all $W, V \in C_{p,\mu,R}(W_{\mathrm{mm}}^\alpha)$ with $\|V\|_{p,p} = \|W_{\mathrm{mm}}^\alpha\|_{p,p}$,

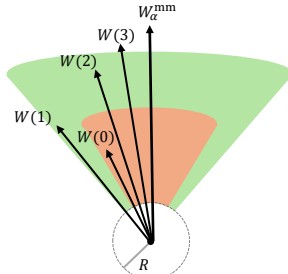

Figure 2: Illustration of Lemma 17. $W(k), \forall k > 0$ are within the larger set.

$$-\langle \nabla \mathcal{L}(W), V \rangle = \Omega \left( \exp \left( -\frac{\|W\|_{p,p}}{\|W_{\mathrm{mm}}^\alpha\|_{p,p}} \left( 1 + \frac{1}{2}\delta \right) \right) \right) > 0.$$

Theorem 3 is a direct consequence of Theorem 4, which extends the analysis that is done for Lemma 17 by providing a tighter bound on how thin the cone set $C_{p,\mu,R}$ may be for later iterates.

## 3.2 TRAINING DYNAMICS OF MIRROR DESCENT FOR JOINT OPTIMIZATION OF $W$ AND $v$

This section explores the training dynamics of jointly optimizing the prediction head $v$ and attention weights $W$. Unlike Section 3.1, the main challenge here is the evolving token scores $\gamma$ influenced by the changing nature of $v$. This requires additional technical considerations beyond those in Section 3.1, which are also addressed in this section. Given stepsizes $\eta_W, \eta_v > 0$, we consider the

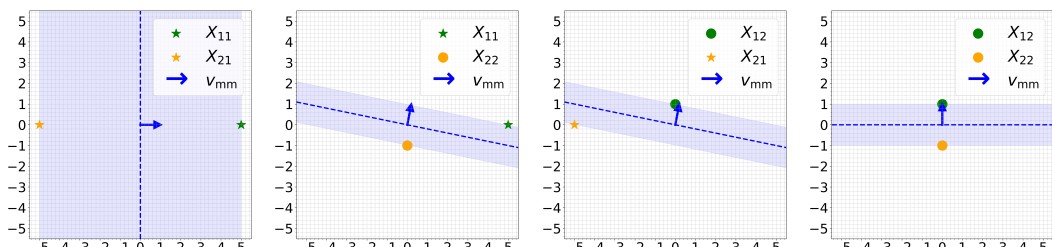

Figure 3: Effect of token selection on margin size in ($\ell_p$-SVM) for Example 1. The first plot shows the largest class margin with optimal tokens $X_{11}$ and $X_{21}$. In subsequent plots, as different tokens are used, the class margin (light blue shaded area) decreases, reflecting suboptimal class separation.

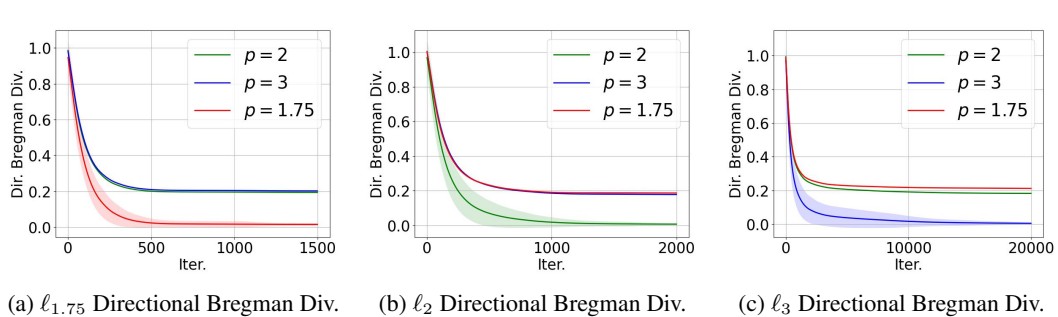

(a) $\ell_{1.75}$ Directional Bregman Div.     (b) $\ell_2$ Directional Bregman Div.     (c) $\ell_3$ Directional Bregman Div.

Figure 4: Average directional Bregman divergence between the (a) $\ell_{1.75}$, (b) $\ell_2$, and (c) $\ell_3$ optimization paths and the ($\ell_p$-AttSVM) solutions for $p = 1.75, 2$, and $3$ at each training iteration from 100 trials. The shaded area represents the standard deviation of the directional Bregman divergence.

following *joint* updates for $W$ and $v$ applied to (ERM), respectively: For all $i, j \in [d]$:

$$\begin{cases} [W(k+1)]_{ij} \leftarrow \left| [W(k)]_{ij}^+ \right|^{\frac{1}{p-1}} \cdot \text{sign}\left( [W(k)]_{ij}^+ \right), \\ [W(k)]_{ij}^+ := |[W(k)]_{ij}|^{p-1}\text{sign}([W(k)]_{ij}) - \eta_W[\nabla_W \mathcal{L}(W(k), v(k))]_{ij}, \\ [v(k+1)]_i \leftarrow \left| [v(k)]_i^+ \right|^{\frac{1}{p-1}} \cdot \text{sign}([v(k)]_i^+), \\ [v(k)]_i^+ := |[v(k)]_i|^{p-1}\text{sign}([v(k)]_i) - \eta_v[\nabla_v \mathcal{L}(W(k), v(k))]_i. \end{cases} \quad (\ell_p\text{-JointGD})$$

We discuss the implicit bias and convergence for $v(k)$ below. From previous results (Azizan et al., 2021), one can expect $v(k)$ to converge to the $\ell_p$-SVM solution, i.e., the max-margin classifier separating the set of samples $\{(X_{i\alpha_i}, y_i)\}_{i=1}^n$, where $X_{i\alpha_i}$ denote the (locally) optimal token for each $i \in [n]$. Consequently, we consider the following hard-margin SVM problem,

$$v_{\text{mm}} = \arg\min_{v \in \mathbb{R}^d} \|v\|_p \quad \text{subj. to} \quad y_i X_{i\alpha_i}^\top v \geq 1 \quad \text{for all} \quad i \in [n]. \quad (\ell_p\text{-SVM})$$

In ($\ell_p$-SVM), define the *label margin* as $1/\|v_{\text{mm}}\|_p$. The label margin quantifies the distance between the separating hyperplane and the nearest data point in the feature space. A larger label margin indicates better generalization performance of the classifier, as it suggests that the classifier has a greater separation between classes. From ($\ell_p$-SVM) and Definitions 2 and 3, an additional intuition by Tarzanagh et al. (2024) behind optimal tokens is that they maximize the label margin when selected; see Figure 3 in the appendix for a visualization. Selecting the locally optimal token indices $\alpha = (\alpha_i)_{i=1}^n$ from each input data sequence achieves the largest label margin, meaning that including other tokens will reduce the label margin as defined in ($\ell_p$-SVM). In the Appendix G, we show that $W$ and $v$ generated by $\ell_p$-JointRP converge to their respective max-margin solutions under suitable geometric conditions (Theorem 5 in the appendix).

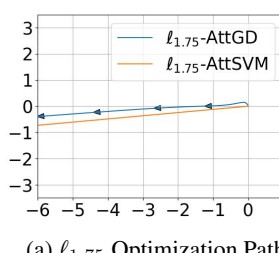
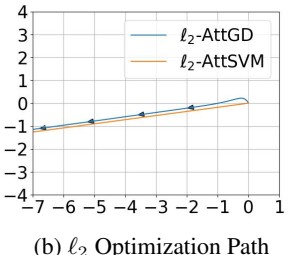
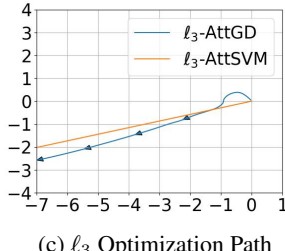

(a) $\ell_{1.75}$ Optimization Path      (b) $\ell_2$ Optimization Path      (c) $\ell_3$ Optimization Path

Figure 5: Direction of change of two entries of $W$ updated by $\ell_p$-AttGD with $p = 1.75$, $p = 2$, and $p = 3$ for one trial, shown in (a), (b), and (c). Each axis represents a different entry. The orange line shows the direction of ($\ell_p$-AttSVM).

## 4 EXPERIMENTAL RESULTS

### 4.1 SYNTHETIC DATA EXPERIMENTS

We describe the setup of the experiments for $\ell_p$-AttGD and $\ell_p$-JointGD and their results.

$\ell_p$-**AttGD Experiment.** To measure the directional distance between $W_\alpha^{\mathrm{mm}}$ (solution of ($\ell_p$-AttSVM)) and $W(k)$ (output of $\ell_p$-AttGD), we use a directional Bregman divergence, defined as $D_\psi(W/\|W\|_{p,p}, V/\|V\|_{p,p})$ for $W, V \in \mathbb{R}^{d \times d}$. We compare the ($\ell_p$-AttSVM) solution with the $\ell_q$ optimization path for all $p, q \in \{1.75, 2, 3\}$ for synthetically generated data (described in detail in the Appendix). The experiment is repeated 100 times, and the average directional Bregman divergence is reported. A closer look at one sample trial is also provided.

Figure 4 shows the directional Bregman divergence between the ($\ell_p$-AttSVM) solution and the $\ell_q$ optimization path for each pair $p, q \in \{1.75, 2, 3\}$. In Figure 4a, the divergence converges to 0 only for the ($\ell_p$-AttSVM) ($p = 1.75$) solution, indicating that the $\ell_{1.75}$ path does not converge to the $p = 2$ or 3 solutions. The shrinking standard deviation shows consistent behavior. Similarly, Figures 4b and 4c show the divergence converging to 0 for the corresponding ($\ell_p$-AttSVM) solution, demonstrating that the $\ell_p$ optimization path converges to the ($\ell_p$-AttSVM) solution, with the direction of convergence changing with $p$. In addition to the directional Bregman divergence, we can also observe the convergence in direction for one of the trials directly by plotting how two of the entries of $W$ change during training simultaneously and plotting it on a Cartesian graph, then showing that the path it follows converges to the direction of the ($\ell_p$-AttSVM) solution. As we can see in Figure 5, each of the $\ell_p$ optimization paths follows the direction of the corresponding ($\ell_p$-AttSVM) solution.

$\ell_p$-**JointGD Experiments.** We use the data from Example 2 in the Appendix to train a model using $\ell_p$-JointGD for $p = 1.75, 2$, and 3. The comparison between the iterates and the SVM solutions in Figure 6 shows that the iterates of $W$ and $v$ converge to the $\ell_p$-AttSVM and $\ell_p$-SVM directions, respectively, for each of $p = 1.75, 2$, and 3. These convergence are similar to Theorem 5, as in both this experiment and that theorem, we get that the iterates converge to the SVM problem solutions. In addition to these iterates, we record the evolution of the average softmax probability of the optimal token, along with the average logistic probability of the model, which we define to be $1/n \sum_{i=1}^n 1/(1 + e^{-\gamma_i \alpha_i})$. As shown in Figure 7, each average softmax probability converges to 1, indicating that the attention mechanism produces a softmax vector converging to a one-hot vector during different $\ell_p$-JointGD training. Moreover, the average logistic probability also converges to 1, indicating the model's prediction reaches 100% accuracy.

### 4.2 REAL DATA EXPERIMENTS

This section presents evidence of improved generalization and token selection from training an attention network with MD instead of GD, along with a hypothesis for this improvement.

We trained a transformer classification model on the Stanford Large Movie Review Dataset (Maas et al., 2011) using MD with $\ell_{1.1}$, $\ell_2$, and $\ell_3$ potentials. The models are similar to the encoder module in Vaswani et al. (2017), with the last layer being a linear classification layer on the feature representation of the first [CLS] token. We put the details of the classification model in the Appendix. Table 1 summarizes the resulting test accuracy of several variants of that model when trained with the three algorithms, which shows that the $\ell_{1.1}$ potential MD outperforms the other MD algorithms, including the

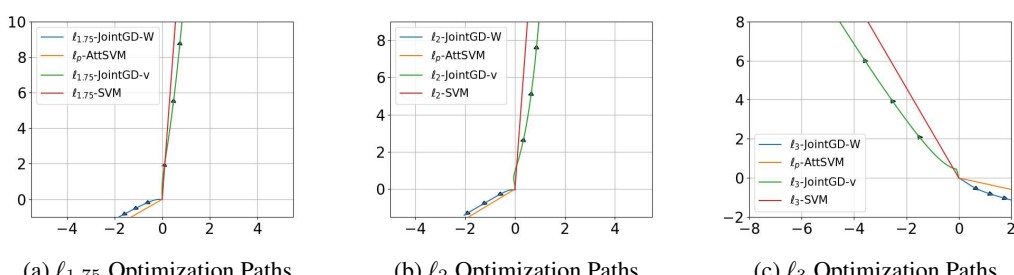

(a) $\ell_{1.75}$ Optimization Paths      (b) $\ell_2$ Optimization Paths      (c) $\ell_3$ Optimization Paths

Figure 6: Iterates of the $W$ and $v$ parameters of the model as they are trained using $\ell_p$-`JointGD` for $p = 1.75, 2$, and 3, along with the corresponding $\ell_p$-AttSVM and $\ell_p$-SVM directions.

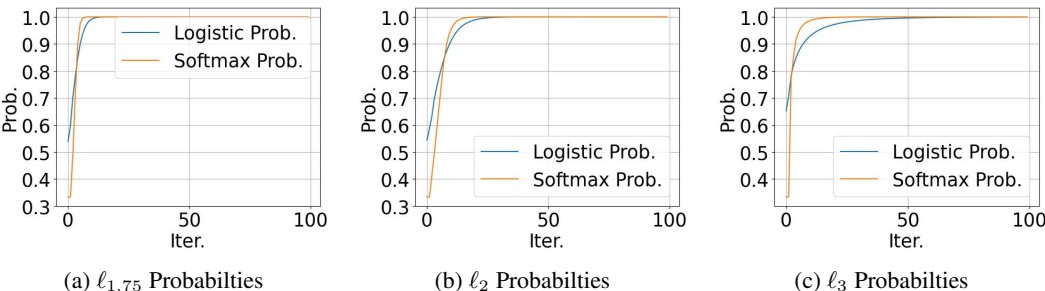

(a) $\ell_{1.75}$ Probabilties      (b) $\ell_2$ Probabilties      (c) $\ell_3$ Probabilties

Figure 7: Softmax probability evolution of the optimal token and logistic probability evolution for $p = 1.75, 2$, and 3.

| Algorithm | Model Size 3 | Model Size 4 | Model Size 6 |
|---|---|---|---|
| $\ell_{1.1}$-MD | **$83.47 \pm 0.09\%$** | **$83.36 \pm 0.13\%$** | **$83.65 \pm 0.13\%$** |
| $\ell_2$-MD | $81.66 \pm 0.09\%$ | $81.05 \pm 0.17\%$ | $82.22 \pm 0.13\%$ |
| $\ell_3$-MD | $82.57 \pm 0.09\%$ | $82.40 \pm 0.12\%$ | $81.97 \pm 0.10\%$ |

Table 1: Test accuracies of transformer classification models trained with $\ell_{1.1}$, $\ell_2$, and $\ell_3$-MD on the **Stanford Large Movie Review Dataset**. The model sizes refers to the number of layers in the transformer model and the number of attention heads per layer. $\ell_{1.1}$-MD provides superior generalization performance.

one with the $\ell_2$ potential, which is equivalent to the GD. To investigate this further, we look at how the attention layers of the model select the tokens from simple reviews that GPT-4o generated and investigate how much the attention layer focuses on a particular token that truly determines whether the whole review was a positive one or a negative one. We chose these pivotal tokens using GPT-4o as well. We do this procedure to the model trained using $\ell_{1.1}$−MD and the GD and tabulate the full results in Appendix H (we provide five of the results in Figure 8). We can see that the $\ell_{1.1}$−MD also outperforms the GD in token selection.

Finally, we collect the training weights from the resulting models trained by $\ell_{1.1}$−MD and the GD and plot a histogram of their absolute values in Figure 9. Specifically, we take the histogram of the components of the key, query, and value matrices. The figures show that the resulting model that was trained using $\ell_{1.1}$−MD is sparser than the one trained using GD, which could hint at a potential explanation as to why $\ell_{1.1}$−MD can outperform the standard GD algorithm when it is used to train attention-based models.

## 5 CONCLUSION

We explored the optimization dynamics and generalization performance of a family of `MD` algorithms for softmax attention mechanisms, focusing on $\ell_p$−`AttGD`, which generalizes GD by using the $p$-th power of the $\ell_p$-norm as the potential function. Our theoretical analysis and experiments show that

| Label | Optimal Token | $\ell_{1.1}$-MD Token Selection | GD Token Selection | Better Selector |
|---|---|---|---|---|
| + | fantastic | the movie was fantastic | the movie was fantastic | 1.1 |
| - | hated | i hated the movie | i hated the movie | 1.1 |
| - | boring | the plot was boring | the plot was boring | 2 |
| + | love | i love this movie | i love this movie | 2 |
| - | terrible | the plot was terrible | the plot was terrible | 1.1 |

Figure 8: The attention map generated by the resulting models that were trained using $\ell_{1.1}$–MD and GD for five sample sentences. For three out of five of the sample sentences, the model trained using $\ell_{1.1}$–MD selects the optimal token better than the model trained using GD.

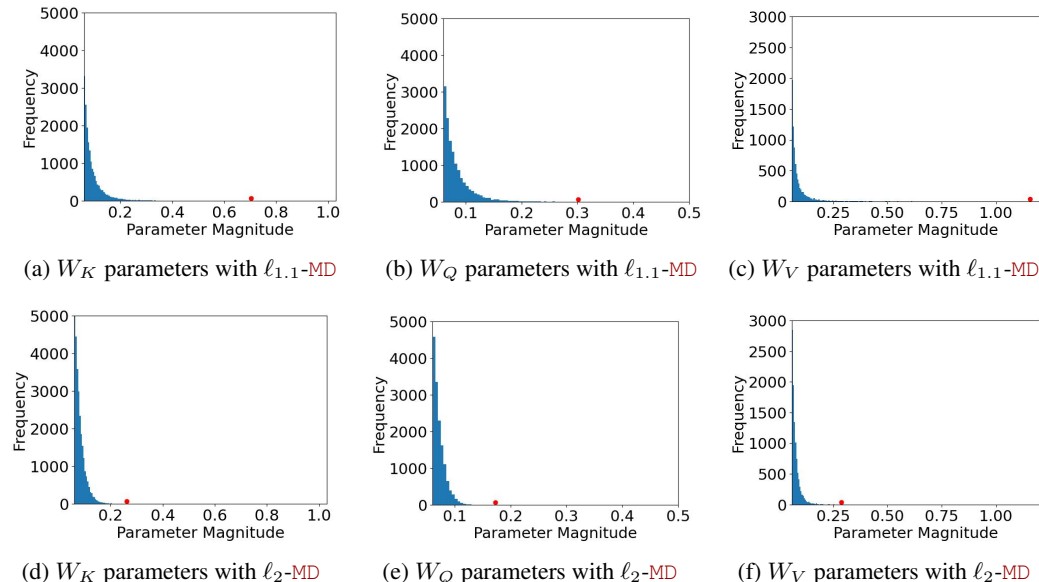

(a) $W_K$ parameters with $\ell_{1.1}$-MD  (b) $W_Q$ parameters with $\ell_{1.1}$-MD  (c) $W_V$ parameters with $\ell_{1.1}$-MD

(d) $W_K$ parameters with $\ell_2$-MD  (e) $W_Q$ parameters with $\ell_2$-MD  (f) $W_V$ parameters with $\ell_2$-MD

Figure 9: Histogram of the absolute values of the $W_K, W_Q$, and $W_V$ components of transformer models trained with $\ell_{1.1}$ and $\ell_2$-MD on the **Stanford Large Movie Review Dataset**. Only large parameters ($\geq 0.06$) are shown, with the maximum magnitude component marked by a red dot. The $\ell_{1.1}$-MD model has $18,206$ components in $W_K$, $13,964$ in $W_Q$, and $7,643$ in $W_V$ with magnitudes $\geq 0.06$, while the $\ell_2$-MD model has $27,224$ in $W_K$, $14,654$ in $W_Q$, and $10,127$ in $W_V$ with such magnitudes. These results imply that the $\ell_{1.1}$-MD algorithm yields sparser parameters and that it would have a stronger token selection ability.

$\ell_p$-AttGD converges to the solution of a generalized hard-margin SVM with an $\ell_p$-norm objective in classification tasks using a single-layer softmax attention model. This generalized SVM separates optimal from non-optimal tokens via linear constraints on token pairs. We also examined the joint problem under logistic loss with $\ell_p$-norm regularization and proved that $W$ and $v$ generated by $\ell_p$-norm regularization path converge to their respective generalized max-margin solutions. Finally, our numerical experiments on real data demonstrate that MD algorithms improve generalization over standard GD and excel in optimal token selection.

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

CONTENTS

## A  ADDITIONAL RELATED WORK

**Transformers Optimization.** Recently, the study of optimization dynamics of attention mechanisms has garnered significant attention (Deora et al., 2023; Huang et al., 2023; Tian et al., 2023b; Fu et al., 2023; Li et al., 2024; Tarzanagh et al., 2024; 2023; Vasudeva et al., 2024a; Sheen et al., 2024; Deng et al., 2023; Makkuva et al., 2024; Jeon et al., 2024; Zheng et al., 2023; Collins et al., 2024; Chen & Li, 2024; Li et al., 2023; Sheen et al., 2024; Ildiz et al., 2024; Vasudeva et al., 2024b; Bao et al., 2024; Chen et al., 2024a; Huang et al., 2024; Wang et al., 2024; Zhao et al., 2024). We discuss the works most closely related to this paper. Studies such as Sahiner et al. (2022); Ergen et al. (2022) investigate the optimization of attention models through convex relaxations. Jelassi et al. (2022) demonstrate that Vision Transformers (ViTs) identify spatial patterns in binary classification via gradient methods. Li et al. (2023) provide sample complexity bounds and discuss attention sparsity in SGD for ViTs. Nguyen et al. (2024) provided static primal-dual formulations for attention mechanisms, connecting self-attention to support vector regression (SVR). Nguyen et al. (2022) connects self-attention to kernel methods to enhance Transformers. Chen et al. (2024b) provided a novel attention mechanism that optimizes self-attention in Transformers using asymmetric Kernel Singular Value Decomposition (KSVD) in the primal representation, achieving improved efficiency and performance through low-rank approximations and regularization techniques. However, these works do not examine optimization dynamics, the role of descent algorithms, or the implications of implicit bias in training, which are the main focus of this work.

Oymak et al. (2023) and Deora et al. (2023) explore optimization dynamics in prompt-attention and multi-head attention models, respectively. Tian et al. (2023a;b) study SGD dynamics and multi-

layer transformer training. Tarzanagh et al. (2024; 2023) explored GD's implicit bias for attention optimization. Vasudeva et al. (2024a) discusses the global directional convergence and convergence rate of GD for attention optimization under specific data conditions. Sheen et al. (2024) notes that gradient flow not only achieves minimal loss but also minimizes the nuclear norm of the key-query weight $W$. Thrampoulidis (2024); Li et al. (2024); Zhao et al. (2024) also studied the optimization dynamics of attention mechanisms and provided the implicit bias of GD for next token prediction. Our work extends these findings and those of Tarzanagh et al. (2024; 2023), focusing on the implicit bias of the general class of MD algorithms for attention training.

**Implicit Bias of First Order Methods.** In recent years, significant progress has been made in understanding the implicit bias of gradient descent on separable data, particularly highlighted by the works of Soudry et al. (2018); Ji & Telgarsky (2018). For linear predictors, Nacson et al. (2019); Ji & Telgarsky (2021); Ji et al. (2021) demonstrated that gradient descent methods rapidly converge to the max-margin predictor. Extending these insights to MLPs, Ji & Telgarsky (2020); Lyu & Li (2019); Chizat & Bach (2020) have examined the implicit bias of GD and gradient flow using exponentially-tailed classification losses, and show convergence to the Karush-Kuhn-Tucker (KKT) points of the corresponding max-margin problem, both in finite Ji & Telgarsky (2020); Lyu & Li (2019) and infinite width scenarios Chizat & Bach (2020). Further, the implicit bias of GD for training ReLU and Leaky-ReLU networks has been investigated, particularly on orthogonal data Phuong & Lampert (2020); Frei et al. (2022). Additionally, the implicit bias towards rank minimization in regression settings with square loss has been explored in Vardi & Shamir (2021); Arora et al. (2019); Li et al. (2020).

Our work is closely related to the implicit bias of MD (Gunasekar et al., 2018; Azizan & Hassibi, 2018) for regression and classification, respectively. Specifically, Sun et al. (2022) extended the findings of Gunasekar et al. (2018); Azizan & Hassibi (2018) to classification problems, and developed a class of algorithms exhibiting an implicit bias towards a generalized SVM with $\ell_p$ norms that effectively separates samples based on their labels; for a survey, we refer to Vardi (2023).

# B  AUXILIARY LEMMAS

## B.1  ADDITIONAL NOTATIONS

Consider the following constants for the proofs, depending on the dataset $(X_i, Y_i, z_i)_{i=1}^{n}$, the parameter $v$, and the locally optimal token $(\alpha_i)_{i=1}^{n}$:

$$\delta' := \frac{1}{2} \min_{i \in [n]} \min_{\tau \in \bar{\mathcal{T}}_i} \left( (X_{i\alpha_i} - X_{i\tau})^\top W_{\mathrm{mm}}^{\alpha} z_i - 1 \right)$$

$$\leq \frac{1}{2} \min_{i \in [n]} \min_{t \in \mathcal{T}_i, \tau \in \bar{\mathcal{T}}_i} \left( (X_{it} - X_{i\tau})^\top W_{\mathrm{mm}}^{\alpha} z_i \right); \tag{6a}$$

$$\delta := \min\{0.25, \delta'\}. \tag{6b}$$

When $\bar{\mathcal{T}}_i = \emptyset$ for all $i \in [n]$ (i.e. globally-optimal indices), we set $\delta' = \infty$ as all non-neighbor related terms will disappear. Further, recalling Definition 4 and using $W_{\mathrm{mm}}^{\alpha}$—i.e., the minimizer of ($\ell_p$-AttSVM), we set

$$A' := \|W_{\mathrm{mm}}^{\alpha}\|_{p,p} \max_{i \in [n], t \in [T]} \|X_{it} z_i^\top\|_{\frac{p}{p-1}, \frac{p}{p-1}};$$

$$A := \max\{1, A'\}. \tag{7}$$

Recalling Definition 5, we provide the following initial radius $\mu = \mu_0$ which will be used later in Lemma 9:

$$\mu_0 := \begin{cases} \dfrac{1}{p} \left( \dfrac{\delta}{8A} \right)^p & \text{if } p \geq 2, \\[3mm] \dfrac{1}{p} \left( \dfrac{\delta(p-1)}{4A d^{\frac{2}{p}-1}} \right)^2 & \text{otherwise.} \end{cases} \tag{8}$$

Furthermore, define the following sums for $W$:

$$S_i(W) := \sum_{t \in \mathcal{T}_i} [\sigma(X_i W z_i)]_t, \quad \text{and} \quad Q_i(W) := \sum_{t \in \bar{\mathcal{T}}_i} [\sigma(X_i W z_i)]_t.$$

For the samples $i$ with non-empty supports $\mathcal{T}_i$, let

$$\gamma_i^{\text{gap}} := \gamma_{i\alpha_i} - \max_{t \in \mathcal{T}_i} \gamma_{it}, \quad \text{and} \quad \bar{\gamma}_i^{\text{gap}} := \gamma_{i\alpha_i} - \min_{t \in \mathcal{T}_i} \gamma_{it}. \tag{9}$$

Furthermore, we define the *global score gap* as

$$\Gamma := \sup_{i \in [n], t, \tau \in [T]} |\gamma_{it} - \gamma_{i\tau}|. \tag{10}$$

### B.2 LEMMA FOR ANALYZING THE $\ell_p$-NORM

In this section of the Appendix, we provide some analysis on comparing the $\ell_p$-norm, the $\ell_p$ Bregman divergence, and the $\ell_2$-norm of matrices. Since the $\ell_2$-norm of matrices are much easier to analyze and use, like in the inner product Cauchy-Schwarz inequality, having this comparison is valuable when analyzing the $\ell_p$-`AttGD`.

**Lemma 1.** *For any $d \times d$ matrix $W$, let $w$ denote its vectorization. Then,*

$$\|w\|_p \in \left[ d^{\frac{2}{p}-1} \|w\|_2, \|w\|_2 \right]$$

*for $p \geq 2$, and for $1 < p \leq 2$, $\|w\|_p$ is in a similar interval, with the two ends switched.*

*Proof.* Let $w_1, w_2, ..., w_{d^2}$ be the entries of $w$. Therefore, for $p \geq 2$,

$$\|w\|_p = \sqrt[p]{\sum_{i=1}^{d^2} |w_i|^p}$$

$$= \sqrt[p]{\sum_{i=1}^{d^2} (|w_i|^2)^{p/2}},$$

and because $\frac{p}{2} \geq 1$, we would have

$$\sqrt[p]{\sum_{i=1}^{d^2} (|w_i|^2)^{p/2}} \leq \sqrt[p]{\left( \sum_{i=1}^{d^2} |w_i|^2 \right)^{p/2}}$$

$$= \sqrt[p]{\|w\|_2^p} = \|w\|_2.$$

Therefore, $\|w\|_p \leq \|w\|_2$ whenever $p \geq 2$. A similar argument will get us $\|w\|_p \geq \|w\|_2$ whenever $1 < p \leq 2$, so one end of the interval is solved for each case, now for the other end.

Using the power-mean inequality, we can get that whenever $p \geq 2$,

$$\sqrt[p]{\frac{1}{d^2} \sum_{i=1}^{d^2} |w_i|^p} \geq \sqrt{\frac{1}{d^2} \sum_{i=1}^{d^2} |w_i|^2},$$

$$d^{-\frac{2}{p}} \|w\|_p \geq d^{-1} \|w\|_2,$$

$$\|w\|_p \geq d^{\frac{2}{p}-1} \|w\|_2.$$

Similarly, for $1 < p \leq 2$,

$$\|w\|_p \leq d^{\frac{2}{p}-1} \|w\|_2.$$

$\square$

**Lemma 2.** *Let $W_1, W_2 \in \mathbb{R}^{d \times d}$ be two matrices such that $\|W_1\|_{p,p} = \|W_2\|_{p,p} = 1$. Then, the following inequalities hold:*

    L1. *For $p \geq 2$,*

$$D_\psi(W_1, W_2) \geq \frac{1}{p \times 2^p} \|W_1 - W_2\|_{p,p}^p,$$

L2. *For $p \in (1, 2)$,*

$$D_\psi(W_1, W_2) \geq \frac{(p-1)^2}{p} \|W_1 - W_2\|_{2,2}^2.$$

*Here, $D_\psi(\cdot, \cdot)$ denotes the Bregman divergence given in Definition 1.*

*Proof.* Let $W_1 = (x_{ij})_{i,j \in [d]}$ and $W_2 = (y_{ij})_{i,j \in [d]}$, then from Definition 1, we have

$$D_\psi(W_1, W_2) = \frac{1}{p} \sum_{i,j \in [d]} |x_{ij}|^p - \frac{1}{p} \sum_{i,j \in [d]} |y_{ij}|^p - \sum_{i,j \in [d]} |y_{ij}|^{p-1}(x_{ij} - y_{ij}) \operatorname{sign}(y_{ij})$$

$$= \sum_{i,j \in [d]} \left( \frac{1}{p} |x_{ij}|^p + \frac{p-1}{p} |y_{ij}|^p - |y_{ij}|^{p-1} |x_{ij}| \operatorname{sign}(x_{ij} y_{ij}) \right).$$

Therefore, it is enough to prove that whenever $x, y \in [-1, 1]$, the expression

$$\frac{1}{p} |x|^p + \frac{p-1}{p} |y|^p - |x||y|^{p-1} \operatorname{sign}(xy) \tag{11}$$

is at least $\frac{1}{p 2^p} |x - y|^p$ if $p \geq 2$, or is at least $\frac{(p-1)^2}{p} |x - y|^2$ if $p \in (1, 2)$. We split the argument into two cases, the first is when the signs of $x$ and $y$ are the same, and the second for when they are not.

**Case 1:** $\operatorname{sign}(xy) = 1$, so both $x$ and $y$ have the same sign, WLOG both are non-negative. Let us fix the value $\Delta \in [-1, 1]$ and find the minimum value of (11) when we constraint $x$ and $y$ to be positive and $x - y = \Delta$. Therefore, that expression can be written as

$$\frac{(y + \Delta)^p + (p - 1)y^p}{p} - (y + \Delta)y^{p-1},$$

the first derivative with respect to $y$ is

$$(y + \Delta)^{p-1} + (p - 1)y^{p-1} - y^{p-1} - (p - 1)(y + \Delta)y^{p-2}$$
$$= (y + \Delta)^{p-1} - y^{p-1} - (p - 1)\Delta y^{p-2}.$$

Since the function $t \mapsto t^{p-1}$ is convex for $p \geq 2$, and concave for $p \in (1, 2)$, then that derivative is always non-negative when $p \geq 2$ and always negative when $p \in (1, 2)$.

**Sub-Case 1.1:** $p \geq 2$. In this subcase, (11) reaches its minimum when $(x, y) = (\Delta, 0)$ or $(0, -\Delta)$, depending on the sign of $\Delta$, plugging them in gets us the minimum, which is $\frac{1}{p} |\Delta|^p$ when $\Delta \geq 0$ or $\frac{p-1}{p} |\Delta|^p$ otherwise.

**Sub-Case 1.2:** $p \in (1, 2)$. In this subcase, (11) reaches its minimum when $(x, y) = (1, 1 - \Delta)$ if $\Delta$ is non-negative or $(1 + \Delta, 1)$ otherwise. When $\Delta$ is non-negative, the desired minimum is

$$\frac{1 + (p-1)(1-\Delta)^p}{p} - (1-\Delta)^{p-1} = \frac{1}{p}(1 - (1-\Delta)^{p-1} - (p-1)\Delta(1-\Delta)^{p-1})$$

$$\geq \frac{1}{p}((p-1)\Delta - (p-1)\Delta(1-\Delta)^{p-1})$$

$$= \frac{(p-1)\Delta}{p}(1 - (1-\Delta)^{p-1}) \geq \frac{(p-1)^2}{p}\Delta^2.$$

Combining the results from the subcases, we get that the expression in (11) is lower-bounded by $\frac{1}{p}|x - y|^p$ when $p \geq 2$, or $\frac{(p-1)^2}{p}|x - y|^2$ otherwise, which sufficiently satisfies the desired bounds for case 1.

**Case 2:** $\operatorname{sign}(xy) = -1$, so $x$ and $y$ has opposite sign. The expression in (11) can be simplified to

$$\frac{1}{p}|x|^p + \frac{p-1}{p}|y|^p + |x||y|^{p-1},$$

and we want to prove that it is at least $\frac{1}{p2^p}(|x| + |y|)^p$ when $p \geq 2$, or is at least $\frac{(p-1)^2}{p}(|x| + |y|)^2$ when $p \in (1,2)$. In the case that $p \geq 2$, one of $|x|$ or $|y|$ is at least $\frac{|x|+|y|}{2}$, so the above is at least $\frac{1}{p}\left(\frac{|x|+|y|}{2}\right)^p = \frac{1}{p2^p}(|x| + |y|)^p$. Otherwise,

$$
\begin{aligned}
\frac{1}{p}|x|^p + \frac{p-1}{p}|y|^p + |x||y|^{p-1} &= \frac{|x|(|x|^{p-1} + |y|^{p-1}) + (p-1)|y|^{p-1}(|x| + |y|)}{p} \\
&\geq \frac{(|x| + |y|)(|x| + (p-1)|y|^{p-1})}{p} \\
&\geq \frac{(|x| + |y|)((p-1)|x| + (p-1)|y|)}{p} \\
&= \frac{p-1}{p}(|x| + |y|)^2 \geq \frac{(p-1)^2}{p}(|x| + |y|)^2.
\end{aligned}
$$

Therefore, we have proven the bound for this case. □

**Lemma 3.** *For any $x \geq y \geq 0$, we we have*

$$
\frac{p-1}{p}x^p - \frac{p-1}{p}y^p \geq y(x^{p-1} - y^{p-1}).
$$

*Proof.*

$$
\frac{d}{dx}\left(\frac{p-1}{p}x^p - \frac{p-1}{p}y^p\right) = (p-1)x^{p-1},
$$

$$
\frac{d}{dx}y(x^{p-1} - y^{p-1}) = (p-1)x^{p-2}y \leq (p-1)x^{p-1},
$$

so as we increase $x$, the left side grows faster than the right side, so we simply need to prove that the inequality holds at $x = y$, which is trivially true. □

**Lemma 4.** *For any $x \geq y \geq 0$, we we have that if $q \geq 1$*

$$
x^q - y^q \leq qx^{q-1}(x - y),
$$

*and if $0 < q < 1$,*

$$
x^q - y^q \leq qy^{q-1}(x - y)
$$

*Proof.*

$$
\frac{d}{dx}(x^q - y^q) = qx^{q-1},
$$

$$
\frac{d}{dx}qx^{q-1}(x - y) = q(q-1)x^{q-2}(x - y) + qx^{q-1}, \text{ and}
$$

$$
\frac{d}{dx}qy^{q-1}(x - y) = qy^{q-1}.
$$

When $q \geq 1$,

$$
\frac{d}{dx}(x^q - y^q) \geq \frac{d}{dx}qx^{q-1}(x - y),
$$

so because we have

$$
x^q - y^q = qx^{q-1}(x - y) = 0
$$

when $x = y$, then

$$
x^q - y^q \geq qx^{q-1}(x - y)
$$

when $x \geq y \geq 0$ if $q \geq 1$. We can use a similar argument for the $0 < q < 1$ case. □

### B.3   LEMMA FOR ANALYZING ERM OBJECTIVE AND ITS GRADIENT

In this section of the Appendix, we analyze the objective function. We especially want to know about its gradient and the inner product of this gradient with the matrices of the cone set, as was mentioned before in the main body of the paper. The first one bounds the loss objective,

**Lemma 5.** *Under Assumption A, $\mathcal{L}(W)$ is bounded from above by $\mathcal{L}_{max}$ and below by $\mathcal{L}_{min}$ for some dataset-dependent constants $\mathcal{L}_{max}$ and $\mathcal{L}_{min}$ that are finite.*

*Proof.* It is enough to show the same thing for each of the loss contributions of each sample, $l_i(y_i v^\top X_i^\top \sigma(X_i W z_i))$. By Assumption A, we simply need to show that $y_i v^\top X_i^\top \sigma(X_i W z_i)$ is bounded by dataset-dependent bounds. However, $W$ only affects the softmax, so the above expression is bounded above by $\max_{t \in [T]} \gamma_{it}$ and bounded below by $\min_{t \in [T]} \gamma_{it}$, which are dataset dependent. $\square$

**Lemma 6.** *If we denote $h_i := X_i W z_i$ and $l_i' := l'(\gamma_i^\top \sigma(h_i))$, then*

$$\nabla \mathcal{L}(W) = \frac{1}{n} \sum_{i=1}^n l_i' X_i^\top (\mathrm{diag}(\sigma(h_i)) - \sigma(h_i)\sigma(h_i)^\top)\gamma_i z_i^\top,$$

*where $\mathcal{L}(W)$ denotes the objective of (ERM) with fixed $v$.*

*Proof.* We first calculate the derivatives of each term in the sum of $\mathcal{L}(W)$. The derivative of the $i$-th term for the $W_{j_1 j_2}$ component is

$$\frac{\partial}{\partial W_{j_1 j_2}} l(y_i v^\top X_i^\top \sigma(X_i W z_i)) = l_i' \gamma_i^\top \frac{\partial}{\partial W_{j_1 j_2}} \sigma(X_i W z_i)$$
$$= l_i' \gamma_i^\top \nabla \sigma(h_i) X_{i,:,j_1}^\top z_{ij_2}$$
$$= l_i' X_{i,:,j_1} \nabla \sigma(h_i)^\top \gamma_i z_{ij_2}.$$

Therefore, the derivative for the $j_2$-th row of $W$ is

$$l_i' X_i^\top \nabla \sigma(h_i)^\top \gamma_i z_{ij_2}.$$

Next, the full gradient for the $i$-th term equals

$$l_i' X_i^\top \nabla \sigma(h_i)^\top \gamma_i z_i^\top.$$

To finish the proof, we calculate the derivative of $\sigma(h_i)$. The derivative of the $j_1$-th component of $\sigma(h_i)$ with respect to $h_{ij_2}$ is

$$\frac{\partial}{\partial h_{ij_2}} \left( \frac{e^{h_{ij_1}}}{\sum_{l=1}^T e^{h_{il}}} \right) = \frac{e^{h_{ij_1}} 1_{j_1 = j_2}}{\sum_{l=1}^T e^{h_{il}}} - \frac{e^{h_{ij_1}} e^{h_{ij_2}}}{\left( \sum_{l=1}^T e^{h_{il}} \right)^2}$$
$$= \sigma(h_i)_{j_1} 1_{j_1 = j_2} - \sigma(h_i)_{j_1} \sigma(h_i)_{j_2}.$$

Thus, the derivative of $\sigma(h_i)$ is a matrix in $\mathbb{R}^{T \times T}$ defined as

$$\mathrm{diag}(\sigma(h_i)) - \sigma(h_i)\sigma(h_i)^\top.$$

Therefore, the full gradient is

$$\frac{1}{n} \sum_{i=1}^n l_i' X_i^\top (\mathrm{diag}(\sigma(h_i)) - \sigma(h_i)\sigma(h_i)^\top)\gamma_i z_i^\top.$$

$\square$

**Lemma 7.** *Under Assumption A, $\|\nabla \mathcal{L}(W)\|_{p,p}$ is bounded by a dataset-dependent constant L.*

*Proof.* Using the expression in Lemma 6, since $l'$ is bounded and the entries in $\sigma(h_i)$ is always between 0 and 1, then the entries of $\nabla \mathcal{L}(W)$ is bounded by a dataset-dependent bounded, which directly implies this lemma statement. $\square$

In the following lemma, we analyze the behaviors of the ($\ell_p$-AttSVM) constraint $(X_{it} - X_{i\tau})^\top W z_i$ for all $W \in S_{p,\mu_0}(W_{\text{mm}}^\alpha)$ satisfying $\|W\|_{p,p} = \|W_{\text{mm}}^\alpha\|_{p,p}$, the result of which is a generalization of (Tarzanagh et al., 2023, Equation 64) for a general $\ell_p$ norm.

**Lemma 8.** *Let $\alpha = (\alpha_i)_{i=1}^n$ be locally optimal tokens as per Definition 3, and let $W_{\text{mm}}^\alpha$ be the ($\ell_p$-AttSVM) solution. Let $(\mathcal{T}_i)_{i=1}^n$ be the index set of all support tokens per Definition 3. Let $\bar{\mathcal{T}}_i = [T] - \mathcal{T}_i - \{\alpha_i\}$. For any $W \in S_{p,\mu_0}(W_{\text{mm}}^\alpha)$ with $\mu_0$ defined in (8) and $\|W\|_{p,p} = \|W_{\text{mm}}^\alpha\|_{p,p}$, we have*

$$(X_{it} - X_{i\tau})^\top W z_i \geq \frac{3}{2}\delta > 0, \tag{12a}$$

$$(X_{i\alpha_i} - X_{i\tau})^\top W z_i \geq 1 + \frac{3}{2}\delta, \tag{12b}$$

$$1 + \frac{1}{2}\delta \geq (X_{i\alpha_i} - X_{it})^\top W z_i \geq 1 - \frac{1}{2}\delta, \tag{12c}$$

*for all $t \in \mathcal{T}_i$ and $\tau \in \bar{\mathcal{T}}_i$*

*Proof.* Let

$$\bar{W} := \frac{W}{\|W\|_{p,p}} \quad \text{and} \quad \bar{W}_{\text{mm}}^\alpha := \frac{W_{\text{mm}}^\alpha}{\|W_{\text{mm}}^\alpha\|_{p,p}}.$$

Using Lemma 2 and the definition of $S_{p,\mu_0}(W_{\text{mm}}^\alpha)$ in (4a), when $p \geq 2$,

$$\|\bar{W} - \bar{W}_{\text{mm}}^\alpha\|_{p,p}^p \leq 2^p p D_\psi(\bar{W}_{\text{mm}}^\alpha, \bar{W})$$
$$\leq 2^p p \mu_0$$
$$= \left(\frac{\delta}{4A}\right)^p,$$

which implies that

$$\|\bar{W} - \bar{W}_{\text{mm}}^\alpha\|_{p,p} \leq \frac{\delta}{4A}.$$

When $p \in (1, 2)$, we can also use Lemmas 1 and 2 to obtain

$$\|\bar{W} - \bar{W}_{\text{mm}}^\alpha\|_{p,p} \leq d^{\frac{2}{p}-1}\|\bar{W} - \bar{W}_{\text{mm}}^\alpha\|_{2,2}$$
$$\leq d^{\frac{2}{p}-1}\frac{\sqrt{p}}{p-1}\sqrt{D_\psi(\bar{W}_{\text{mm}}^\alpha, \bar{W})}$$
$$\leq d^{\frac{2}{p}-1}\frac{\sqrt{p}}{p-1}\sqrt{\mu_0} = \frac{\delta}{4A},$$

where the last inequality uses the definition of $S_{p,\mu_0}(W_{\text{mm}}^\alpha)$ in (4a).

Therefore, either way, we have

$$\|W - W_{\text{mm}}^\alpha\|_{p,p} \leq \frac{\delta}{4A}\|W_{\text{mm}}^\alpha\|_{p,p}.$$

We will proceed to show a bound on $(X_{it_1} - X_{it_2})^\top (W - W_{\text{mm}}^\alpha) z_i$ for any $i \in [n]$ and any token indices $t_1, t_2 \in [T]$. To do that, let us focus on the term $X_{it_1}^\top (W - W_{\text{mm}}^\alpha) z_i$ first,

$$\left| X_{it_1}^\top (W - W_{\text{mm}}^\alpha) z_i \right| = \left| \langle W - W_{\text{mm}}^\alpha, X_{it_1} z_i^\top \rangle \right|$$
$$\leq \|W - W_{\text{mm}}^\alpha\|_{p,p} \cdot \|X_{it_1} z_i^\top\|_{\frac{p}{p-1}, \frac{p}{p-1}}$$
$$\leq \frac{\delta}{4A}\|W_{\text{mm}}^\alpha\|_{p,p} \cdot \|X_{it_1} z_i^\top\|_{\frac{p}{p-1}, \frac{p}{p-1}}$$
$$\leq \frac{\delta}{4A} \cdot A$$
$$= \frac{\delta}{4}.$$

The first inequality above uses Hölder's Inequality. We now have

$$\left|(X_{it_1} - X_{it_2})^\top (W - W_{\mathrm{mm}}^\alpha) z_i\right| \leq \frac{1}{2}\delta.$$

To obtain the first inequality of the lemma in (12a), for all $t \in \mathcal{T}_i$ and $\tau \in \bar{\mathcal{T}}_i$, we have

$$(X_{it} - X_{i\tau})^\top W z_i \geq (X_{it} - X_{i\tau})^\top W_{\mathrm{mm}}^\alpha z_i + (X_{it} - X_{i\tau})^\top (W - W_{\mathrm{mm}}^\alpha) z_i$$
$$\geq 2\delta' - \frac{1}{2}\delta \geq \frac{3}{2}\delta.$$

To get the second inequality in (12b), for all $\tau \in \bar{\mathcal{T}}_i$, we have

$$(X_{i\alpha_i} - X_{i\tau})^\top W z_i \geq (X_{i\alpha_i} - X_{i\tau})^\top W_{\mathrm{mm}}^\alpha z_i + (X_{i\alpha_i} - X_{i\tau})^\top (W - W_{\mathrm{mm}}^\alpha) z_i$$
$$\geq 1 + 2\delta' - \frac{1}{2}\delta \geq 1 + \frac{3}{2}\delta.$$

Finally, to get the last inequality in (12c), for all $t \in \mathcal{T}_i$, we have

$$\left|(X_{i\alpha_i} - X_{it})^\top W z_i - 1\right| = \left|(X_{i\alpha_i} - X_{it})^\top W_{\mathrm{mm}}^\alpha z_i + (X_{i\alpha_i} - X_{it})^\top (W - W_{\mathrm{mm}}^\alpha) z_i - 1\right|$$
$$= |(X_{i\alpha_i} - X_{it})^\top (W - W_{\mathrm{mm}}^\alpha) z_i| \leq \frac{1}{2}\delta,$$

which implies that

$$1 + \frac{1}{2}\delta \geq (X_{i\alpha_i} - X_{it})^\top W z_i \geq 1 - \frac{1}{2}\delta.$$

$\square$

The following two lemmas aim at bounding the correlation between the gradient and the attention matrix parameter, each of which is a generalization of (Tarzanagh et al., 2023, Lemmas 13 and 14) for the generalized $\ell_p$ norm.

**Lemma 9.** *Suppose Assumption A holds. Let $\alpha = (\alpha_i)_{i=1}^n$ be locally optimal tokens as per Definition 3, and let $W_{\mathrm{mm}}^\alpha$ be the solution to ($\ell_p$-AttSVM). There exists a dataset-dependent constant $R_\delta = \mathcal{O}(1/\delta)$ such that for all $W, V \in C_{p,\mu_0,R_\delta}(W_{\mathrm{mm}}^\alpha)$ with $\|V\|_{p,p} = \|W_{\mathrm{mm}}^\alpha\|_{p,p}$, $\delta$ and $\mu_0$ defined in (6) and (8), respectively,*

$$-\langle \nabla\mathcal{L}(W), V\rangle = \Omega\left(\exp\left(-\frac{\|W\|_{p,p}}{\|W_{\mathrm{mm}}^\alpha\|_{p,p}}\left(1 + \frac{1}{2}\delta\right)\right)\right) > 0.$$

*Proof.* Let

$$h_i := X_i W z_i, \quad \tilde{h}_i := X_i V z_i, \quad l_i' := l'(\gamma_i^\top \sigma(h_i)), \quad \text{and} \quad s_i = \sigma(h_i).$$

Therefore,

$$\langle \nabla\mathcal{L}(W), V\rangle = \frac{1}{n}\sum_{i=1}^n l_i'\langle X_i^\top(\mathrm{diag}(s_i) - s_i s_i^\top)\gamma_i z_i^\top, V\rangle$$

$$= \frac{1}{n}\sum_{i=1}^n l_i'\langle(\mathrm{diag}(s_i) - s_i s_i^\top)\gamma_i, X_i V z_i\rangle$$

$$= \frac{1}{n}\sum_{i=1}^n l_i'\langle(\mathrm{diag}(s_i) - s_i s_i^\top)\gamma_i, \tilde{h}_i\rangle$$

$$= \frac{1}{n}\sum_{i=1}^n l_i'\tilde{h}_i^\top(\mathrm{diag}(s_i) - s_i s_i^\top)\gamma_i,$$

$$-\langle \nabla\mathcal{L}(W), V\rangle = \frac{1}{n}\sum_{i=1}^n (-l_i')\tilde{h}_i^\top(\mathrm{diag}(s_i) - s_i s_i^\top)\gamma_i. \tag{13}$$

The value $\gamma_i^\top \sigma(h_i)$ for any $i \in [n]$ must be bounded, and the bound is only dataset-dependent, so by Assumption A, $l_i'$ is bounded for any $i \in [n]$ by some bound that is dataset-dependent. Furthermore, because $l$ is decreasing, $-l'$ is always non-negative, so an easier approach is to lower-bound the following for each $i \in [n]$,

$$\tilde{h}_i^\top s_i s_i^\top \gamma_i - \tilde{h}_i^\top \operatorname{diag}(s_i)\gamma_i.$$

Next, we can get for all $i \in [n]$ and $t \in [T]$ that

$$\tilde{h}_{it} = X_{it}^\top V z_i = \langle X_{it} z_i^\top, V \rangle$$
$$\leq \|V\|_{p,p} \|X_{it} z_i^\top\|_{\frac{p}{p-1}}$$
$$\leq A,$$

where $A$ is defined in (7).

Therefore, if we drop the $i$ notation and let $\alpha_i = 1$, and use (Tarzanagh et al., 2023, Lemma 7),

$$\left| \tilde{h}^\top s s^\top \gamma - \tilde{h}^\top \operatorname{diag}(s)\gamma - \sum_{t=2}^T (\tilde{h}_1 - \tilde{h}_t)s_t(\gamma_1 - \gamma_t) \right| \leq 2\Gamma A(1 - s_1)^2.$$

Let us attempt to remove the non-support tokens from the sum above by bounding the sum of the term for the non-supports,

$$\left| \sum_{t \in \bar{\mathcal{T}}} (\tilde{h}_1 - \tilde{h}_t)s_t(\gamma_1 - \gamma_t) \right| \leq 2 \max_{t \in [T]}\{|\tilde{h}_t|\}Q(W)\Gamma \leq 2AQ(W)\Gamma.$$

Therefore,

$$\left| \tilde{h}^\top s s^\top \gamma - \tilde{h}^\top \operatorname{diag}(s)\gamma - \sum_{t \in \mathcal{T}} (\tilde{h}_1 - \tilde{h}_t)s_t(\gamma_1 - \gamma_t) \right| \leq 2\Gamma A((1 - s_1)^2 + Q(W)),$$

which implies that

$$\tilde{h}^\top s s^\top \gamma - \tilde{h}^\top \operatorname{diag}(s)\gamma \geq \sum_{t \in \mathcal{T}} (\tilde{h}_1 - \tilde{h}_t)s_t(\gamma_1 - \gamma_t) - 2\Gamma A((1 - s_1)^2 + Q(W)).$$

Using Lemma 8, we have

$$\tilde{h}^\top s s^\top \gamma - \tilde{h}^\top \operatorname{diag}(s)\gamma \geq \left(1 - \frac{1}{2}\delta\right) \sum_{t \in \mathcal{T}} s_t(\gamma_1 - \gamma_t) - 2\Gamma A((1 - s_1)^2 + Q(W)). \quad (14)$$

To proceed, we can upper-bound $1 - s_1$ and $Q(W)$. For bounding $1 - s_1$, let $\tau > 1$ be some index that maximizes $X_\tau^\top W z$, so

$$1 - s_1 = \frac{\sum_{t=2}^T e^{X_t^\top W z}}{\sum_{t=1}^T e^{X_t^\top W z}} \leq \frac{(T-1)e^{X_\tau^\top W z}}{(T-1)e^{X_\tau^\top W z} + e^{X_1^\top W z}}$$
$$\leq \frac{T}{T + e^{(X_1 - X_\tau)^\top W z}}$$
$$\leq \frac{T}{T + e^{\frac{\|W\|_{p,p}}{\|W_{\mathrm{mm}}^\alpha\|_{p,p}}(1 - \frac{1}{2}\delta)}}$$
$$\leq \frac{T}{e^{\frac{\|W\|_{p,p}}{\|W_{\mathrm{mm}}^\alpha\|_{p,p}}(1 - \frac{1}{2}\delta)}},$$

with the last inequality using the third inequality Lemma 8.

For ease of notation, denote

$$R' := \frac{\|W\|_{p,p}}{\|W_{\mathrm{mm}}^\alpha\|_{p,p}}. \quad (15)$$

To upper bound $Q(W)$, we use a method similar to that for $1 - s_1$, but we utilize the second inequality of Lemma 8 instead of the first. This gives:

$$Q(W) \leq \frac{T}{T + e^{(1 + \frac{3}{2}\delta)R'}} \leq \frac{T}{e^{(1 + \frac{3}{2}\delta)R'}}.$$

Therefore, we have

$$2\Gamma A((1 - s_1)^2 + Q(W)) \leq 2\Gamma A \left( \frac{T^2}{e^{(2-\delta)R'}} + \frac{T}{e^{(1+\frac{3}{2}\delta)R'}} \right)$$

$$\leq \frac{2\Gamma A T(T+1)}{e^{(1+\frac{3}{2}\delta)R'}}. \tag{16}$$

Now it is time to lower-bound the sum on the right side of Equation (14). When the set of supports is empty, that sum is zero. However, if it is not empty,

$$\sum_{t \in \mathcal{T}} s_t(\gamma_1 - \gamma_t) \geq S(W)\gamma^{\text{gap}}.$$

If we let $\tau \in \mathcal{T}$ be the support index that minimizes $X_\tau^\top W z$, then

$$S(W) = \frac{\sum_{t \in \mathcal{T}} e^{X_t^\top W z}}{\sum_{t=1}^{T} e^{X_t^\top W z}} \geq \frac{e^{X_\tau^\top W z}}{T e^{X_1^\top W z}} = \frac{1}{T e^{(X_1 - X_\tau)^\top W z}}$$

$$\geq \frac{1}{T e^{(1+\frac{1}{2}\delta)R'}},$$

with the last inequality coming from the third inequality of Lemma 8.

Therefore,

$$\sum_{t \in \mathcal{T}} s_t(\gamma_1 - \gamma_t) \geq \frac{\gamma^{\text{gap}}}{T e^{(1+\frac{1}{2}\delta)R'}} > 0.$$

Using Equation (14), we get that if the support index set is empty,

$$\tilde{h}^\top s s^\top \gamma - \tilde{h}^\top \operatorname{diag}(s)\gamma \geq -\frac{2\Gamma A T(T+1)}{e^{(1+\frac{3}{2}\delta)R'}},$$

otherwise,

$$\tilde{h}^\top s s^\top \gamma - \tilde{h}^\top \operatorname{diag}(s)\gamma \geq \frac{\gamma^{\text{gap}}}{T e^{(1+\frac{1}{2}\delta)R'}} \left(1 - \frac{1}{2}\delta\right) - \frac{2\Gamma A T(T+1)}{e^{(1+\frac{3}{2}\delta)R'}}.$$

Plugging everything back into Equation (13), and considering that some samples will have non-empty support index sets, we have:

$$-\langle \mathcal{L}(W), V \rangle \geq -\frac{\min_{i \in \mathcal{T}_i}\{\gamma_i^{\text{gap}}\}}{nT e^{(1+\frac{1}{2}\delta)R'}} \left(1 - \frac{1}{2}\delta\right) \max_{i=1}^{n}\{l_i'\}$$

$$+ \frac{2\Gamma A T(T+1)}{e^{(1+\frac{3}{2}\delta)R'}} \sum_{i=1}^{n} l_i' = \Omega \left( e^{-(1+\frac{1}{2}\delta)R'} \right). \tag{17}$$

Let

$$\bar{L} := \frac{\sum_{i=1}^{n} l_i'}{\max_{i=1}^{n}\{l_i'\}}. \tag{18}$$

Note that using Assumption A, $\bar{L}$ is positive. Hence, using (18) and (17), the term $-\langle \mathcal{L}(W), V \rangle$ is positive when

$$R' \geq \frac{1}{\delta} \log \left( \frac{2\Gamma \bar{L} A T^2 (T+1) n}{\min_{i \in \mathcal{T}_i}\{\gamma_i^{\text{gap}}\} \left(1 - \frac{1}{2}\delta\right)} \right),$$

or equivalently, from (15), we have

$$\|W\|_{p,p} \geq \frac{\|W_{\text{mm}}^\alpha\|_{p,p}}{\delta} \log \left( \frac{2\Gamma \bar{L} A T^2 (T+1)}{\min_{i \in \mathcal{T}_i}\{\gamma_i^{\text{gap}}\} \left(1 - \frac{1}{2}\delta\right)} \right).$$

$\square$

Finally, we introduce the following lemma to help understand the correlation between the gradient of the objective and the parameter.

**Lemma 10.** *Suppose Assumption A holds. Let $\alpha = (\alpha_i)_{i=1}^n$ be locally optimal tokens as per Definition 3, let $W_{\mathrm{mm}}^\alpha$ be the ($\ell_p$-AttSVM) solution, and let $R_\delta$ be the constant from Lemma 9. For any choice of $\pi \in (0,1)$, there exists $R_\pi$ that depends on $\pi$ defined as*

$$R_\pi := \max \left\{ R_\delta, \mathcal{O}\left( \frac{1}{\pi\delta} \log \frac{\delta}{\pi} \right) \right\},$$

*such that for all $W \in C_{p,\mu_0,R_\pi}(W_{\mathrm{mm}}^\alpha)$,*

$$\left\langle \nabla \mathcal{L}(W), \frac{W}{\|W\|_{p,p}} \right\rangle \geq (1+\pi) \left\langle \nabla \mathcal{L}(W), \frac{W_{\mathrm{mm}}^\alpha}{\|W_{\mathrm{mm}}^\alpha\|_{p,p}} \right\rangle.$$

*Proof.* Let

$$h_i := X_i W z_i, \quad \tilde{h}_i := X_i W_{\mathrm{mm}}^\alpha z_i, \quad l_i' := l'(\gamma_i^\top \sigma(h_i)),$$

$$s_i := \sigma(h_i), \quad \bar{W} := \frac{\|W_{\mathrm{mm}}^\alpha\|_{p,p} W}{\|W\|_{p,p}}, \quad \text{and} \quad \bar{h}_i := X_i \bar{W} z_i. \tag{19}$$

By decomposing $\mathcal{L}(W)$ into its sum and using Lemma 6, the main inequality is equivalent to the following,

$$\sum_{i=1}^n (-l_i')\langle X_i^\top (\mathrm{diag}(s_i) - s_i s_i^\top)\gamma_i z_i^\top, \bar{W}\rangle$$

$$\leq (1+\pi) \sum_{i=1}^n (-l_i')\langle X_i^\top (\mathrm{diag}(s_i) - s_i s_i^\top)\gamma_i z_i^\top, W_{\mathrm{mm}}^\alpha\rangle,$$

which implies that

$$\sum_{i=1}^n (-l_i')\langle (\mathrm{diag}(s_i) - s_i s_i^\top)\gamma_i, X_i \bar{W} z_i\rangle$$

$$\leq (1+\pi) \sum_{i=1}^n (-l_i')\langle (\mathrm{diag}(s_i) - s_i s_i^\top)\gamma_i, X_i W_{\mathrm{mm}}^\alpha z_i\rangle.$$

Using (19), we get

$$\sum_{i=1}^n (-l_i')\langle (\mathrm{diag}(s_i) - s_i s_i^\top)\gamma_i, \bar{h}_i\rangle \leq (1+\pi) \sum_{i=1}^n (-l_i')\langle (\mathrm{diag}(s_i) - s_i s_i^\top)\gamma_i, \tilde{h}_i\rangle,$$

which gives

$$\sum_{i=1}^n (-l_i')\bar{h}_i^\top (\mathrm{diag}(s_i) - s_i s_i^\top)\gamma_i \leq (1+\pi) \sum_{i=1}^n (-l_i')\tilde{h}_i^\top (\mathrm{diag}(s_i) - s_i s_i^\top)\gamma_i.$$

Hence,

$$\sum_{i=1}^n (-l_i')\left[ (1+\pi)\left( \tilde{h}_i^\top \mathrm{diag}(s_i)\gamma_i - \tilde{h}_i^\top s_i s_i^\top \gamma_i \right) - \left( \bar{h}_i^\top \mathrm{diag}(s_i)\gamma_i - \bar{h}_i^\top s_i s_i^\top \gamma_i \right) \right] \geq 0.$$

Using a similar technique as the one we used to prove Lemma 9,

$$\left| \tilde{h}_i^\top \mathrm{diag}(s_i)\gamma_i - \tilde{h}_i^\top s_i s_i^\top \gamma_i - \sum_{t \in \mathcal{T}_i} (\tilde{h}_{i\alpha_i} - \tilde{h}_{it})s_{it}(\gamma_{i\alpha_i} - \gamma_{it}) \right|$$

$$\leq 2\Gamma A((1 - s_{i\alpha_i})^2 + Q_i(W)).$$

Similarly,

$$\left| \bar{h}_i^\top \operatorname{diag}(s_i)\gamma_i - \bar{h}_i^\top s_i s_i^\top \gamma_i - \sum_{t \in \mathcal{T}_i}(\bar{h}_{i\alpha_i} - \bar{h}_{it})s_{it}(\gamma_{i\alpha_i} - \gamma_{it}) \right|$$
$$\leq 2\Gamma A((1 - s_{i\alpha_i})^2 + Q_i(W)).$$

Therefore, it is enough to prove that

$$\sum_{i=1}^n (-l_i') \left( (1 + \pi) \left( \sum_{t \in \mathcal{T}_i} (\tilde{h}_{i\alpha_i} - \tilde{h}_{it})s_{it}(\gamma_{i\alpha_i} - \gamma_{it}) - 2\Gamma A((1 - s_{i\alpha_i})^2 + Q_i(W)) \right) \right.$$
$$\left. - \left( \sum_{t \in \mathcal{T}_i} (\bar{h}_{i\alpha_i} - \bar{h}_{it})s_{it}(\gamma_{i\alpha_i} - \gamma_{it}) + 2\Gamma A((1 - s_{i\alpha_i})^2 + Q_i(W)) \right) \right), \tag{20}$$

Using the fact that $\pi < 1$ and using Equation (16), we get another lower-bound

$$\sum_{i=1}^n \sum_{t \in \mathcal{T}_i} (-l_i')(1 + \pi - (\bar{h}_{i\alpha_i} - \bar{h}_{it}))s_{it}(\gamma_{i\alpha_i} - \gamma_{it}) + \frac{6\Gamma AT(T+1)}{e^{(1 + \frac{3}{2}\delta)R'}} \sum_{i=1}^n l_i', \tag{21}$$

with $R'$ again being $\frac{\|W\|_{p,p}}{\|W_{\mathrm{mm}}^\alpha\|_{p,p}}$. Next, we analyze the softmax probability $s_{it}$, and lower and upper-bound them in terms of $R'$ and $\bar{h}_{i\alpha_i} - \bar{h}_{it}$. For the lower-bound,

$$s_{it} = \frac{e^{\bar{h}_{it}R'}}{\sum_{\tau \in [T]} e^{\bar{h}_{i\tau}R'}} \geq \frac{e^{\bar{h}_{it}R'}}{Te^{\bar{h}_{i\alpha_i}R'}}$$
$$= \frac{1}{T}e^{-(\bar{h}_{i\alpha_i} - \bar{h}_{it})R'}.$$

For the upper-bound,

$$s_{it} = \frac{e^{\bar{h}_{it}R'}}{\sum_{\tau \in [T]} e^{\bar{h}_{i\tau}R'}} \leq \frac{e^{\bar{h}_{it}R'}}{e^{\bar{h}_{i\alpha_i}R'}}$$
$$= e^{-(\bar{h}_{i\alpha_i} - \bar{h}_{it})R'}.$$

In both bounds, the main inequality derivation stems from the fact that $\bar{h}_{i\alpha_i} > \bar{h}_{i\tau}$ for all $\tau \in [T]$, which we obtain from Lemma 8. Now, we analyze the left double-summation in Equation (21). To analyze the sum, let $\mathcal{I}$ be the subset of $[n] \times [T]$ that contains all $(i, t)$ such that $t \in \mathcal{T}_i$. Furthermore, let

$$\mathcal{I}_1 := \left\{ (i, t) \in \mathcal{I} \mid \bar{h}_{i\alpha_i} - \bar{h}_{it} \leq 1 \right\},$$
$$\mathcal{I}_2 := \left\{ (i, t) \in \mathcal{I} \mid 1 < \bar{h}_{i\alpha_i} - \bar{h}_{it} \leq 1 + \pi \right\},$$
$$\mathcal{I}_3 := \left\{ (i, t) \in \mathcal{I} \mid \bar{h}_{i\alpha_i} - \bar{h}_{it} > 1 + \pi \right\}.$$

Therefore, we can split the sum above into the sum over $\mathcal{I}_1$, $\mathcal{I}_2$, and $\mathcal{I}_3$. The set $\mathcal{I}_1$ in particular must be non-empty because $\|\bar{W}\|_{p,p} = \|W_{\mathrm{mm}}^\alpha\|_{p,p}$, meaning that one of the constraints in the $\ell_p$-AttSVM problem must either be fulfilled exactly or violated.

The sum over $\mathcal{I}_1$ must be positive and is at least

$$-\frac{\pi}{T} \min_{i \in \mathcal{I}_1}\{\gamma_i^{gap}\}e^{-R'} \max_{i=1}^n\{l_i'\}.$$

The sum over $\mathcal{I}_2$ must be non-negative, and the sum over $\mathcal{I}_3$ is negative can be bounded from below using Lemma 8

$$\frac{1}{2}\delta \max_{i \in \mathcal{I}_3}\{\bar{\gamma}_i^{gap}\}Te^{-(1+\pi)R'} \sum_{i=1}^n l_i'.$$

Putting things together into Equation (21), we get that we want the following to be non-negative

$$-\frac{\pi}{T}\min_{i\in\mathcal{I}_1}\{\gamma_i^{gap}\}e^{-R'}\max_{i=1}^n\{l_i'\} + \frac{1}{2}\delta\max_{i\in\mathcal{I}_3}\{\bar{\gamma}_i^{gap}\}Te^{-(1+\pi)R'}\sum_{i=1}^n l_i'$$

$$+ 6\Gamma AT(T+1)e^{-(1+\frac{3}{2}\delta)R'}\sum_{i=1}^n l_i'.$$

This can be achieved when

$$R' \geq \frac{1}{\min\{\pi,\frac{3}{2}\delta\}}\log\left(\frac{\frac{1}{2}\delta\max_{i\in\mathcal{I}_3}\{\bar{\gamma}_i^{gap}\}T^2 + 6\Gamma AT^2(T+1)}{\pi\min_{i\in\mathcal{I}_1}\{\gamma_i^{gap}\}\max_{i=1}^n\{l_i'\}}\sum_{i=1}^n l_i'\right),$$

or equivalently,

$$\|W\|_{p,p} \geq \frac{\|W_{\mathrm{mm}}^\alpha\|_{p,p}}{\min\{\pi,\frac{3}{2}\delta\}}\log\left(\frac{\frac{1}{2}\delta\max_{i\in\mathcal{I}_3}\{\bar{\gamma}_i^{gap}\}T^2 + 6\Gamma AT^2(T+1)}{\pi\min_{i\in\mathcal{I}_1}\{\gamma_i^{gap}\}\max_{i=1}^n\{l_i'\}}\sum_{i=1}^n l_i'\right),$$

which means that such dataset dependent $R_\pi$ exists. $\qquad\square$

### B.4 LEMMA FOR ANALYZING $\ell_p$-ATTGD

We introduce the lemmas for analyzing $\ell_p$-AttGD. The first we prove is Lemma 11, which describes the lower bound of the $W$ parameter at every iterate.

**Lemma 11.** *Suppose Assumption A holds. For the sequence $\{W(k)\}_{k\geq 0}$ generated by $\ell_p$-AttGD, we have*

$$\|W(k+1)\|_{p,p}^{p-1} \geq \|W(k)\|_{p,p}^{p-1} + \frac{\eta}{\|W(k)\|_{p,p}}\langle-\nabla\mathcal{L}(W(k)), W(k)\rangle.$$

*Proof.* With $\psi(W) = \frac{1}{p}\|W\|_{p,p}$, the derivative $\nabla\psi(\cdot)$ is computed as follows:

$$\nabla\psi(W) = (\mathrm{sign}(W_{ij})|W_{ij}|^{p-1})_{1\leq i,j\leq d}.$$

Thus, we have

$$\langle\nabla\psi(W), W\rangle = \sum_{i,j}\mathrm{sign}(W_{ij})|W_{ij}|^{p-1}W_{ij} = \|W\|_{p,p}^p.$$

Using this fact, we take the inner product of both sides of (23) with $W(k)$:

$$\langle\nabla\psi(W(k+1)), W(k)\rangle = \langle\nabla\psi(W(k)), W(k)\rangle + \eta\langle-\nabla\mathcal{L}(W(k)), W(k)\rangle,$$

$$\langle\nabla\psi(W(k+1)), W(k)\rangle = \|W(k)\|_{p,p}^p + \eta\langle-\nabla\mathcal{L}(W(k)), W(k)\rangle. \qquad(22)$$

The left side of the above equation is upper-bounded by

$$\sum_{i,j}\mathrm{sign}(W_{ij}(k+1))|W_{ij}(k+1)|^{p-1}W_{ij}(k) \leq \sum_{i,j}|W_{ij}(k+1)|^{p-1}|W_{ij}(k)|.$$

Using Hölder's inequality:

$$\sum_{i,j}|W_{ij}(k+1)|^{p-1}|W_{ij}(k)| \leq \left(\sum_{i,j}(|W_{ij}(k+1)|^{p-1})^{\frac{p}{p-1}}\right)^{\frac{p-1}{p}}\left(\sum_{i,j}|W_{ij}(k)|^p\right)^{\frac{1}{p}}$$

$$= \|W(k+1)\|_{p,p}^{p-1}\|W(k)\|_{p,p}.$$

Combining this result with (22), we get:

$$\|W(k+1)\|_{p,p}^{p-1} \geq \|W(k)\|_{p,p}^{p-1} + \frac{\eta}{\|W(k)\|_{p,p}}\langle-\nabla\mathcal{L}(W(k)), W(k)\rangle.$$

$\qquad\square$

Next, we show several tools for analyzing the algorithm further and for analyzing the Bregman divergence. The following three specifically are from Sun et al. (2022, Lemma 18, 3) and Azizan & Hassibi (2018), and so the proofs are omitted.

**Lemma 12.** *For any $W \in \mathbb{R}^{d \times d}$, the following identities hold for MD:*

$$D_\psi(W, W(k)) = D_\psi(W, W(k+1)) + D_{\psi - \eta \mathcal{L}}(W(k+1), W(k))$$
$$- \eta \langle \nabla \mathcal{L}(W(k)), W - W(k) \rangle - \eta \mathcal{L}(W(k)) + \eta \mathcal{L}(W(k+1)). \quad (23)$$

**Lemma 13.** *Suppose Assumptions A hold and $\eta$ is small enough. For the sequence $\{W(k)\}_{k \geq 0}$ generated by $\ell_p$-AttGD, we have*

$$\frac{p-1}{p} \|W(k+1)\|_{p,p}^p - \frac{p-1}{p} \|W(k)\|_{p,p}^p + \eta \mathcal{L}(W(k+1)) - \eta \mathcal{L}(W(k))$$
$$\leq \langle -\eta \nabla \mathcal{L}(W(k)), W(k) \rangle. \quad (24)$$

**Lemma 14.** *Suppose Assumptions A hold. Consider the sequence $W(k)$ generated by Algorithm $\ell_p$-AttGD. Given that the step size $\eta$ is sufficiently small, then the ERM objective $\mathcal{L}(W(k))$ is decreasing in $k$.*

This following is a well-known lemma, so the proof is omitted.

**Lemma 15** (Bregman Divergences Cosine Law). *For any $w, w', w''$ that are all vectors or matrices with the same dimensionalities, we have*

$$D_\psi(w, w') = D_\psi(w, w'') + D_\psi(w'', w') - \langle \nabla \psi(w') - \nabla \psi(w''), w - w'' \rangle.$$

The following is adapted from Sun et al. (2022, Equation 12) for the case of our attention model. Our proof is quite similar, except that we use our version of the gradient correlation lemma.

**Lemma 16.** *Suppose Assumptions A hold. Consider the sequence $W(k)$ generated by Algorithm $\ell_p$-AttGD. For any $\pi \in (0,1)$, if $W(k) \in C_{p,\mu_0,R_\pi}(W_{\mathrm{mm}}^\alpha)$, with $R_\pi$ being the constant from Lemma 10, then for a small enough step size $\eta$,*

$$\langle \nabla \psi(W(k+1)) - \nabla \psi(W(k)), \bar{W}_{\mathrm{mm}}^\alpha \rangle \geq \frac{1}{1+\pi} (\|W(k+1)\|_{p,p}^{p-1} - \|W(k)\|_{p,p}^{p-1})$$
$$+ \frac{\eta}{\|W(k)\|_{p,p}} (\mathcal{L}(W(k+1)) - \mathcal{L}(W(k))). \quad (25)$$

*Proof.* Let $\bar{W}_{\mathrm{mm}}^\alpha = \frac{W_{\mathrm{mm}}^\alpha}{\|W_{\mathrm{mm}}^\alpha\|_{p,p}}$. Using the $\ell_p$-AttGD algorithm equation,

$$\langle \nabla \psi(W(k+1)) - \nabla \psi(W(k)), \bar{W}_{\mathrm{mm}}^\alpha \rangle = \langle -\eta \nabla \mathcal{L}(W(k)), \bar{W}_{\mathrm{mm}}^\alpha \rangle.$$

Then, using Lemma 10, we get that

$$\langle -\eta \nabla \mathcal{L}(W(k)), \bar{W}_{\mathrm{mm}}^\alpha \rangle \geq \frac{1}{(1+\pi)\|W(k)\|_{p,p}} \langle -\eta \nabla \mathcal{L}(W(k)), W(k) \rangle,$$

and using Lemma 13, we get that this is lower-bounded by

$$\frac{p-1}{p(1+\pi)\|W(k)\|_{p,p}} (\|W(k+1)\|_{p,p}^p - \|W(k)\|_{p,p}^p) + \frac{\eta}{(1+\pi)\|W(k)\|_{p,p}} (\mathcal{L}(W(k+1)) - \mathcal{L}(W(k))).$$

By Lemma 9, $\langle -\eta \nabla \mathcal{L}(W(k)), W(k) \rangle > 0$, so by Lemma 11, $\|W(k+1)\|_{p,p} \geq \|W(k)\|_{p,p}$. Therefore, we can use Lemma 3 to get that the above is lower-bounded by

$$\frac{1}{1+\pi} (\|W(k+1)\|_{p,p}^{p-1} - \|W(k)\|_{p,p}^{p-1}) + \frac{\eta}{(1+\pi)\|W(k)\|_{p,p}} (\mathcal{L}(W(k+1)) - \mathcal{L}(W(k))).$$

From Lemma 14, we get that we can lower-bound the above further using the right hand side of (25). $\qquad \square$

With all these lemmas in hand, we provide the following Lemma 17.

**Lemma 17.** *Suppose Assumptions A holds and that the step size $\eta$ is sufficiently small. For any $\mu \in (0, \mu_0]$ and any locally optimal tokens $(\alpha_i)_{i=1}^n$ as per Definition 3, there exists constants $R_\mu$ and $\mu' \in (0, \mu]$ that depends on the dataset and $\mu$ such that if $C_1$ is the wider cone $C_{p,\mu,R_\mu}(W_{\mathrm{mm}}^\alpha)$ and $C_2$ is the thinner cone $C_{p,\mu',R_\mu}(W_{\mathrm{mm}}^\alpha)$, then if $W(0) \in C_2$, then $W(k) \in C_1$ for all positive indices $k$.*

*Proof.* Let $\pi$ be some positive real number that we determine later, and let $R_\pi$ be as described in Lemma 10.

For the proof, we use induction with the assumption that $W(k) \in C_{p,\mu,R_\pi}(W_{\mathrm{mm}}^\alpha)$ for all $k = 0, \ldots, K-1$. We aim to find the correct $\mu'$ and $R_\mu$ such that $W(K) \in C_{p,\mu,R_\pi}(W_{\mathrm{mm}}^\alpha)$.

Denote $\bar{W}(k) := \frac{W(k)}{\|W(k)\|_{p,p}}$, so

$$D_\psi(\bar{W}_{\mathrm{mm}}^\alpha, \bar{W}(k)) = \frac{1}{p}\|\bar{W}_{\mathrm{mm}}^\alpha\|_{p,p} - \frac{1}{p}\|\bar{W}(k)\|_{p,p} - \langle \nabla\psi(\bar{W}(k)), \bar{W}_{\mathrm{mm}}^\alpha - \bar{W}(k)\rangle$$

$$= 1 - \langle \nabla\psi(\bar{W}(k)), \bar{W}_{\mathrm{mm}}^\alpha\rangle.$$

So now, let us analyze the term $\langle \nabla\psi(\bar{W}(K)), \bar{W}_{\mathrm{mm}}^\alpha\rangle$ using the inductive hypothesis on $k = 0, 1, ..., K-1$. Lemma 16 tells us that

$$\langle \nabla\psi(W(k+1)) - \nabla\psi(W(k)), \bar{W}_{\mathrm{mm}}^\alpha\rangle \geq \frac{\|W(k+1)\|_{p,p}^{p-1} - \|W(k)\|_{p,p}^{p-1}}{(1+\pi)} \tag{26}$$

$$+ \frac{\eta}{\|W(k)\|_{p,p}}(\mathcal{L}(W(k+1)) - \mathcal{L}(W(k))).$$

Since this is true for all $k = 0, 1, ..., K-1$, and since $\|W(k)\|_{p,p}$ is increasing in $k$, we can sum all the above inequalities and get the following,

$$\langle \nabla\psi(W(K)) - \nabla\psi(W(0)), \bar{W}_{\mathrm{mm}}^\alpha\rangle \geq \frac{\|W(K)\|_{p,p}^{p-1} - \|W(0)\|_{p,p}^{p-1}}{(1+\pi)}$$

$$+ \frac{\eta}{\|W(0)\|_{p,p}}(\mathcal{L}(W(K)) - \mathcal{L}(W(0))).$$

Rearranging this, we get

$$\|W(K)\|_{p,p}^{p-1} - \langle \nabla\psi(W(K)), \bar{W}_{\mathrm{mm}}^\alpha\rangle \leq \|W(0)\|_{p,p}^{p-1} - \langle \nabla\psi(W(0)), \bar{W}_{\mathrm{mm}}^\alpha\rangle$$

$$+ \frac{\pi}{1+\pi}(\|W(K)\|_{p,p}^{p-1} - \|W(0)\|_{p,p}^{p-1})$$

$$+ \frac{\eta}{\|W(0)\|_{p,p}}(\mathcal{L}(W(0)) - \mathcal{L}(W(K))).$$

Dividing by $\|W(K)\|_{p,p}^{p-1}$, we get

$$D_\psi(\bar{W}_{\mathrm{mm}}^\alpha, \bar{W}(K)) \leq \frac{\|W(0)\|_{p,p}^{p-1}}{\|W(K)\|_{p,p}^{p-1}}D_\psi(\bar{W}_{\mathrm{mm}}^\alpha, \bar{W}(0)) + \frac{\pi}{1+\pi}\left(1 - \frac{\|W(0)\|_{p,p}^{p-1}}{\|W(K)\|_{p,p}^{p-1}}\right)$$

$$+ \frac{\eta}{\|W(K)\|_{p,p}^{p-1}\|W(0)\|_{p,p}}(\mathcal{L}(W(0)) - \mathcal{L}(W(K))) \tag{27}$$

$$\leq \mu' + \pi + \frac{\eta(\mathcal{L}(W(0)) - \mathcal{L}(W(K)))}{R_\mu^p}.$$

Therefore, we can simply choose $\mu' = \frac{1}{3}\mu$, $\pi$ be any real number below $\frac{1}{3}\mu$, and have $R_\mu$ big enough so that $\frac{\eta(\mathcal{L}(W(0))-\mathcal{L}(W(K)))}{R_\mu^p} \leq \frac{1}{3}\mu$ and $R_\mu \geq R_\pi$, such $R_\mu$ exists because $\mathcal{L}$ is bounded. $\qquad\square$

### B.5 LEMMA FOR ANALYZING RATE OF CONVERGENCE

**Lemma 18.** *Suppose Assumptions A holds. Let $R_\delta$ be from Lemma 9, let $c$ be from Lemma 16, let $\mu'$ and $R_\mu$ be from Lemma 17 when $\mu = \mu_0$, and let $R := \max\{R_\mu, R_\delta, e^{1/c}\}$. For any $W \in \mathbb{R}^{d\times d}$,*

denote $\bar{W} := W/\|W\|_{p,p}$. *If the initialization $W(0)$ is in $C_{p,\mu',R}(W_{\mathrm{mm}}^\alpha)$, then for a sufficiently small step size $\eta$, the sequence $\{W(k)\}_{k\geq 0}$ generated by $\ell_p$-AttGD satisfies*

$$D_\psi(\bar{W}_{\mathrm{mm}}^\alpha, \bar{W}(k)) = \begin{cases} \mathcal{O}\left(\dfrac{\log\|W(k)\|_{p,p}}{\|W(k)\|_{p,p}}\right) & \text{if } p > 2, \\[2mm] \mathcal{O}\left(\dfrac{(\log\|W(k)\|_{p,p})^2}{\|W(k)\|_{p,p}}\right) & \text{if } p = 2, \\[2mm] \mathcal{O}\left(\dfrac{1}{\|W(k)\|_{p,p}^{p-1}}\right) & \text{otherwise.} \end{cases} \tag{28}$$

*Proof.* Using Lemma 10, setting $c$ as the dataset dependent constant hidden by the $\mathcal{O}$ notation for $R_\pi$, we can get that by setting $\pi = \min\{\frac{c\log\|W(k)\|_{p,p}}{\delta\|W(k)\|_{p,p}}, 1\}$, we can use the result of Lemma 16 on $k$, so rearranging that result, we get

$$\|W(k+1)\|_{p,p}^{p-1} - \langle\nabla\psi(W(k+1)), \bar{W}_{\mathrm{mm}}^\alpha\rangle \leq \|W(k)\|_{p,p}^{p-1} - \langle\nabla\psi(W(k)), \bar{W}_{\mathrm{mm}}^\alpha\rangle$$
$$+ \frac{\pi}{1+\pi}(\|W(k+1)\|_{p,p}^{p-1} - \|W(k)\|_{p,p}^{p-1})$$
$$+ \frac{\eta}{\|W(k)\|_{p,p}}(\mathcal{L}(W(k)) - \mathcal{L}(W(k+1))).$$

From Lemma 9 and Lemma 11, $\|W(k)\|_{p,p}$ is increasing, so focusing on the second line, we can use Lemma 4 and get

$$\frac{\pi}{1+\pi}(\|W(k+1)\|_{p,p}^{p-1} - \|W(k)\|_{p,p}^{p-1}) \leq \pi(\|W(k+1)\|_{p,p}^{p-1} - \|W(k)\|_{p,p}^{p-1})$$
$$\leq \frac{cp}{\delta\|W(k)\|_{p,p}}\max\{\|W(k)\|_{p,p}^{p-2}, \|W(k+1)\|_{p,p}^{p-2}\}$$
$$\times \log\|W(k)\|_{p,p}$$
$$\times (\|W(k+1)\|_{p,p} - \|W(k)\|_{p,p}).$$

From Lemma 7, we know that for all index $k$,

$$\|W(k+1)\|_{p,p} \leq \|W(k)\|_{p,p} + \eta L, \tag{29}$$

so we can use integral approximation when bounding the sums of $\Delta(k)$'s. Let

$$\Delta(k) = \frac{cp}{\delta\|W(k)\|_{p,p}}\max\{\|W(k)\|_{p,p}^{p-2}, \|W(k+1)\|_{p,p}^{p-2}\}\log\|W(k)\|_{p,p}$$
$$\times (\|W(k+1)\|_{p,p} - \|W(k)\|_{p,p}),$$

so we can get that

$$\|W(K)\|_{p,p}^{p-1} - \langle\nabla\psi(W(K)), \bar{W}_{\mathrm{mm}}^\alpha\rangle \leq \|W(0)\|_{p,p}^{p-1} - \langle\nabla\psi(W(0)), \bar{W}_{\mathrm{mm}}^\alpha\rangle$$
$$+ \sum_{k=0}^{k-1}\Delta(k) + \frac{\eta}{c}(\mathcal{L}(W(0)) - \mathcal{L}(W(K))),$$

$$\|W(K)\|_{p,p}^{p-1}D_\psi(\bar{W}_{\mathrm{mm}}^\alpha, \bar{W}(K)) \leq \|W(0)\|_{p,p}^{p-1}D_\psi(\bar{W}_{\mathrm{mm}}^\alpha, \bar{W}(0))$$
$$+ \sum_{k=0}^{k-1}\Delta(k) + \frac{\eta}{c}(\mathcal{L}(W(0)) - \mathcal{L}(W(K))). \tag{30}$$

When $p > 2$, we have

$$\Delta(k) = \frac{cp}{\delta\|W(k)\|_{p,p}}(\|W(k)\|_{p,p} + \eta L)^{p-2}\log\|W(k)\|_{p,p}(\|W(k+1)\|_{p,p} - \|W(k)\|_{p,p}).$$

We can see that

$$\frac{d}{dx}(x + \eta L)^{p-2}(\log x - \log c) > \frac{p-2}{x}(x + \eta L)^{p-2}\log x$$

for all $x > 0$, so from Equation (29), we can get that

$$\sum_{k=0}^{K-1} \Delta(k) = O(\|W(K)\|^{p-2} \log \|W(K)\|_{p,p}).$$

When $p = 2$, we have

$$\Delta(k) = \frac{cp}{\|W(k)\|_{p,p}} \log \|W(k)\|_{p,p}(\|W(k+1)\|_{p,p} - \|W(k)\|_{p,p}).$$

We can see that

$$\frac{d}{dx}(\log x)^2 > \frac{2}{x}(\log x)$$

for all $x \geq c$, so from Equation (29), we can get that

$$\sum_{k=0}^{K-1} \Delta(k) = O((\log \|W(K)\|_{p,p})^2).$$

When $p < 2$, we have

$$\Delta(k) = cp\|W(k)\|_{p,p}^{p-3} \log \|W(k)\|_{p,p}(\|W(k+1)\|_{p,p} - \|W(k)\|_{p,p}).$$

From Equation (29), we can get that

$$\sum_{k=0}^{K-1} \Delta(k) = O(1).$$

Combining the above cases with Equation (30), we get that

$$\|W(K)\|_{p,p}^{p-1} D_\psi(\bar{W}_{\mathrm{mm}}^\alpha, \bar{W}(K)) = \begin{cases} O(\|W(K)\|_{p,p}^{p-2} \log \|W(K)\|_{p,p}) & \text{if } p > 2, \\ O((\log \|W(K)\|_{p,p})^2) & \text{if } p = 2, \\ O(1) & \text{otherwise} \end{cases}$$

Dividing both sides by $\|W(K)\|_{p,p}^{p-1}$ gives (28). $\qquad \square$

**Lemma 19.** *Suppose Assumptions A holds. Let $\mu'$ be that from Lemma 17 if $\mu = \mu_0$, and let $R$ the maximum of the $R_\mu$ from 17 and $R_\delta$ 9. Let $\{W(k)\}_{k \geq 0}$ be the sequence generated by $\ell_p$-$\mathtt{AttGD}$. If the initialization $W(0)$ is in $C_{p,\mu',R}(W_{\mathrm{mm}}^\alpha)$, then with a small enough step size $\eta$, we have the following for each $k \geq 0$,*

$$\|W(k)\|_{p,p} = \Omega(\log k).$$

*Proof.* For each $k \geq 0$, Lemma 11 gives

$$\|W(k+1)\|_{p,p}^{p-1} \geq \|W(k)\|_{p,p}^{p-1} + \frac{\eta}{\|W(k)\|_{p,p}} \langle -\nabla\mathcal{L}(W(k)), W(k) \rangle.$$

Lemma 17 gives us that $W(k) \in C_{p,\mu,R}(W_{\mathrm{mm}}^\alpha)$ for each $k \geq 0$, so by Lemma 9,

$$\frac{\eta}{\|W(k)\|_{p,p}} \langle -\nabla\mathcal{L}(W(k)), W(k) \rangle = \Omega\left(e^{-\frac{\|W(k)\|_{p,p}}{\|W_{\mathrm{mm}}^\alpha\|_{p,p}}(1+\frac{1}{2}\delta)}\right),$$

so there exists dataset dependent constants $R_1, R_2 > 0$ such that

$$\frac{\eta}{\|W(k)\|_{p,p}} \langle -\nabla\mathcal{L}(W(k)), W(k) \rangle \geq R_1 e^{-R_2\|W(k)\|_{p,p}},$$

so for each $k \geq 0$,

$$\|W(k+1)\|_{p,p}^{p-1} \geq \|W(k)\|_{p,p}^{p-1} + R_1 e^{-R_2\|W(k)\|_{p,p}}.$$

Set $k_0 = 0$, and let $k_{i+1}$ be the lowest indices such that $\|W(k_{i+1})\|_{p,p} \geq \|W(k_i)\|_{p,p} + 1$ for all index $i \geq 0$. Therefore,

$$k_{i+1} - k_i \leq \frac{(\|W(k_i)\|_{p,p} + 1)^{p-1} - \|W(k_i)\|_{p,p}^{p-1}}{R_1 e^{-R_2(\|W(k_i)\|_{p,p}+1)}} = e^{O(\|W(k_i)\|_{p,p})}.$$

Therefore,

$$\|W(k)\|_{p,p} = \Omega(\log k).$$

$\qquad \square$

## C  PROOF OF THEOREM 1

*Proof.* The proof is similar to the proof of (Tarzanagh et al., 2024, Theorem 1). Specifically, we need to show that $f(X) = v^\top X^\top \sigma(XW)$ satisfies the assumptions of (Tarzanagh et al., 2024, Lemma 14), where the nonlinear head is replaced by the linear term $v$. This holds independently of the choice of algorithm or the attention SVM solution. Thus, we omit the details and refer to the proof of (Tarzanagh et al., 2024, Theorem 1). ◻

## D  PROOF OF THEOREM 2

*Proof.* It is enough to show the existence of such constants $\mu, R > 0$ such that if $W(0)$ is in $C_{p,\mu,R}(W_{\mathrm{mm}}^\alpha)$, then the norm diverges to infinity. As discussed in Lemma 11, for any timestep $k$,

$$\|W(k+1)\|_p^{p-1} \geq \|W(k)\|_p^{p-1} - \frac{\eta}{\|W(k)\|_p}\langle\nabla\mathcal{L}(W(k)), W(k)\rangle. \tag{31}$$

Let $R_1$ be the $R$ from Lemma 9, set $\mu$ and $R_2$ to be the $\mu'$ and $R$ for $\mu = \mu_0$ of Lemma 17, and set $R := \max\{R_1, R_2\}$. From Lemma 17, we know that $W(k) \in C_{p,\mu_0,R}(W_\alpha^{\mathrm{mm}})$ for any timestep $k$, so from Lemma 9,

$$\langle\nabla\mathcal{L}(W(k)), W(k)\rangle < 0,$$

for all timesteps $k$.

Therefore, the $l_p$-norm is always increasing, but this does not immediately imply that the $l_p$-norm will approach infinity; it could converge to a finite value. However, if $\|W(k)\|_p$ converges to a finite value, then again by Lemma 9, we get a lower bound for $-\frac{\eta}{\|W(k)\|_p}\langle\nabla\mathcal{L}(W(k)), W(k)\rangle$ at any timestep $k$. Therefore, by Equation (31),

$$\lim_{k\to\infty}\|W(k)\|_p^{p-1} = \infty,$$

a contradiction, so $\|W(k)\|_p$ converges to infinity. ◻

## E  PROOF OF THEOREM 3

*Proof.* This is a direct consequence of Theorem 4. ◻

## F  PROOF OF THEOREM 4

*Proof.* Let $R$ be the one from Lemma 18. Given $W(0) \in C_{p,\mu,R}(W_{\mathrm{mm}}^\alpha)$, by Lemma 18, we have

$$D_\psi(\bar{W}_{\mathrm{mm}}^\alpha, \bar{W}(k)) = \begin{cases} \mathcal{O}\left(\dfrac{\log\|W(k)\|_{p,p}}{\|W(k)\|_{p,p}}\right) & \text{if } p > 2, \\[3mm] \mathcal{O}\left(\dfrac{(\log\|W(k)\|_{p,p})^2}{\|W(k)\|_{p,p}}\right) & \text{if } p = 2, \\[3mm] \mathcal{O}\left(\dfrac{1}{\|W(k)\|_{p,p}^{p-1}}\right) & \text{otherwise.} \end{cases}$$

From Lemma 19, we know that

$$\|W(k)\|_{p,p} = \Omega(\log k).$$

The derivative $\frac{d}{dx}\left(\frac{\log x}{x}\right) = \frac{1-\log x}{x^2}$ is negative when $x > e$, so $\frac{\log x}{x}$ is decreasing when $x > e$. Similarly, $\frac{(\log x)^2}{x}$ is decreasing when $x > e^2$.

Thus when $p > 2$, for a large enough $k$,

$$D_\psi(\bar{W}_{\mathrm{mm}}^\alpha, \bar{W}(k)) = O\left(\frac{\log\log k}{\log k}\right). \tag{32a}$$

Similarly, when $p = 2$, for a large enough $k$,

$$D_\psi(\bar{W}_{\text{mm}}^\alpha, \bar{W}(k)) = O\left(\frac{(\log \log k)^2}{\log k}\right). \tag{32b}$$

Finally, when $1 < p < 2$,

$$D_\psi(\bar{W}_{\text{mm}}^\alpha, \bar{W}(k)) = O\left(\frac{1}{(\log k)^{p-1}}\right). \tag{32c}$$

$\square$

# G  On the Convergence of the $\ell_p$ Regularization Path for Joint $W$ and $v$

In this section, we extend the results of Theorem 1 to the case of joint optimization of head $v$ and attention weights $W$ using a logistic loss function.

**Assumption B.** *Let $\Gamma, \Gamma' > 0$ denote the label margins when solving ($\ell_p$-SVM) with $X_{i\alpha_i}$ and its replacement with $X_i^\top \sigma(X_i W z_i)$, for all $i \in [n]$, respectively. There exists $\nu > 0$ such that for all $i \in [n]$ and $W \in \mathbb{R}^{d \times d}$,*

$$\Gamma - \Gamma' \geq \nu \cdot (1 - s_{i\alpha_i}), \quad \text{where} \quad s_{i\alpha_i} = [\sigma(X_i W z_i)]_{\alpha_i}.$$

Assumption B is similar to Tarzanagh et al. (2024) and highlights that selecting optimal tokens is key to maximizing the classifier's label margin. When attention features, a weighted combination of all tokens, are used, the label margin shrinks based on how much attention is given to the optimal tokens. The term $\nu \cdot (1 - s_{i\alpha_i})$ quantifies this minimum shrinkage. If the attention mechanism fails to focus on these tokens (i.e., low $s_{i\alpha_i}$), the margin decreases, reducing generalization. This assumption implies that optimal performance is achieved when attention converges on the most important tokens, aligning with the max-margin attention SVM solution.

Similar to how we provided the characterization of convergence for the regularization path of $\ell_p$-AttGD, we offer a similar characterization here for $\ell_p$-JointGD.

**Theorem 5** (Joint $\ell_p$–norm Regularization Path)**.** *Consider (ERM) with a logistic loss $l(x) = \log(1 + e^{-x})$, and define*

$$(v^{(r)}, W^{(R)}) := \underset{(v,W)}{\arg\min} \ \mathcal{L}(v, W) \quad \text{subj. to} \quad \|W\|_{p,p} \leq R \text{ and } \|v\|_p \leq r. \quad (\ell_p\text{-JointRP})$$

*Suppose there are token indices $\alpha = (\alpha_i)_{i=1}^m$ for which $W_{\text{mm}}^\alpha$ of ($\ell_p$-AttSVM) exists and Assumption B holds for some $\Gamma, \nu > 0$. Then,*

$$\lim_{(r,R) \to (\infty,\infty)} \left(\frac{v^{(r)}}{r}, \frac{W^{(R)}}{R}\right) = \left(\frac{v_{\text{mm}}}{\|v_{\text{mm}}\|_p}, \frac{W_{\text{mm}}^\alpha}{\|W_{\text{mm}}^\alpha\|_{p,p}}\right). \tag{33}$$

*Here, $v_{\text{mm}}$ and $W_{\text{mm}}^\alpha$ are the solution of ($\ell_p$-SVM) and ($\ell_p$-AttSVM), respectively.*

*Proof.* The proof follows a similar approach to (Tarzanagh et al., 2024, Theorem 5), adapted to the $\ell_p$-norm. We provide the revised version for the generalized attention SVM, tracking the required changes. Without loss of generality, we set $\alpha_i = 1$ for all $i \in [n]$, and we use $W_{\text{mm}}$ instead of $W_{\text{mm}}^\alpha$. Suppose the claim is incorrect, meaning either $W^{(R)}/R$ or $v^{(r)}/r$ fails to converge as $R$ and $r$ grow. Define

$$\Xi = \frac{1}{\|\bar{W}_{\text{mm}}\|_{p,p}}, \qquad \Gamma = \frac{1}{\|v_{\text{mm}}\|_p},$$
$$\bar{W}_{\text{mm}} := R\Xi W_{\text{mm}}, \qquad \bar{v}_{\text{mm}} := r\Gamma v_{\text{mm}} \tag{34}$$

Our strategy is to show that $(\bar{v}_{\text{mm}}, \bar{W}_{\text{mm}})$ is a strictly better solution compared to $(v^{(r)}, W^{(R)})$ for large $R$ and $r$, leading to a contradiction.

● **Case 1:** $W^{(R)}/R$ **does not converge to** $\bar{W}_{\mathrm{mm}}/R$. In this case, there exists $\delta, \gamma = \gamma(\delta) > 0$ such that we can find arbitrarily large $R$ with

$$\|W^{(R)}/R - \bar{W}_{\mathrm{mm}}/R\| \geq \delta$$

and the margin induced by $W^{(R)}/R$ is at most $\Xi(1 - \gamma)$.

We bound the amount of non-optimality $q_i^*$ of $\bar{W}_{\mathrm{mm}}$:

$$q_i^* := \frac{\sum_{t \neq \alpha_i} \exp(X_{it}^\top \bar{W}_{\mathrm{mm}} z_i)}{\sum_{t \in [T]} \exp(X_{it}^\top \bar{W}_{\mathrm{mm}} z_i)} \leq \frac{\sum_{t \neq \alpha_i} \exp(X_{it}^\top \bar{W}_{\mathrm{mm}} z_i)}{\exp(X_{i\alpha_i}^\top \bar{W}_{\mathrm{mm}} z_i)}$$

$$\leq T \exp(-\Xi R).$$

Thus,

$$q_{\max}^* := \max_{i \in [n]} q_i^* \leq T \exp(-\Xi R). \tag{35a}$$

Next, assume without loss of generality that the first margin constraint is $\gamma$-violated by $W^{(R)}$, meaning

$$\min_{t \neq \alpha_1} (X_{1\alpha_1} - X_{1t})^\top W^{(R)} z_1 \leq \Xi R(1 - \gamma).$$

Denoting the amount of non-optimality of the first input of $W^{(R)}$ as $\hat{q}_1$, we find

$$\hat{q}_1 := \frac{\sum_{t \neq \alpha_1} \exp(X_{1t}^\top W^{(R)} z_1)}{\sum_{t \in [T]} \exp(X_{1t}^\top W^{(R)} z_1)} \geq \frac{1}{T} \frac{\sum_{t \neq \alpha_1} \exp(X_{1t}^\top W^{(R)} z_1)}{\exp(X_{1\alpha_1}^\top W^{(R)} z_1)}$$

$$\geq T^{-1} \exp(-(1 - \gamma) R \Xi).$$

This implies that

$$\hat{q}_{\max} := \max_{i \in [n]} q_i^* \geq T^{-1} \exp(-\Xi R(1 - \gamma)). \tag{35b}$$

We similarly have

$$q_{\max}^* \geq T^{-1} \exp(-\Xi R). \tag{35c}$$

Thus, (35) gives the following relationship between the upper and lower bounds on non-optimality:

$$-(1 - \gamma)\Xi R - \log T \leq \log(\hat{q}_{\max}),$$
$$-\Xi R - \log T \leq \log(q_{\max}^*) \leq -\Xi R + \log T. \tag{36}$$

In other words, $\bar{W}_{\mathrm{mm}}$ has exponentially less non-optimality compared to $W^{(R)}$ as $R$ grows. To proceed, we need to upper and lower bound the logistic loss of $(\bar{v}_{\mathrm{mm}}, \bar{W}_{\mathrm{mm}})$ and $(v^{(r)}, W^{(R)})$ respectively, to establish a contradiction.

● **Sub-Case 1.1: Upper bound for** $\mathcal{L}(\bar{v}_{\mathrm{mm}}, \bar{W}_{\mathrm{mm}})$. We now bound the logistic loss for the limiting solution. Set $\bar{r}_i = X_i^\top \sigma(X_i \bar{W}_{\mathrm{mm}} z_i)$. If $\|\bar{r}_i - X_{i1}\|_p \leq \epsilon_i$, then $v_{\mathrm{mm}}$ satisfies the SVM constraints on $\bar{r}_i$ with $Y_i \cdot \bar{r}_i^\top v_{\mathrm{mm}} \geq 1 - \epsilon_i/\Gamma$. Setting $\epsilon_{\max} = \sup_{i \in [n]} \epsilon_i$, $v_{\mathrm{mm}}$ achieves a label-margin of $\Gamma - \epsilon_{\max}$ on the dataset $(Y_i, \bar{r}_i)_{i \in [n]}$. Let $M = \sup_{i \in [n], t, \tau \in [T]} \|X_{it} - X_{i\tau}\|_p$. Recalling (36), the worst-case perturbation is

$$\epsilon_{\max} \leq M \exp(-\Xi R + \log T) = MT \exp(-\Xi R).$$

This implies the upper bound on the logistic loss:

$$\mathcal{L}(\bar{v}_{\mathrm{mm}}, \bar{W}_{\mathrm{mm}}) \leq \max_{i \in [n]} \log(1 + \exp(-Y_i \bar{r}_i^\top \bar{v}_{\mathrm{mm}}))$$

$$\leq \max_{i \in [n]} \exp(-Y_i \bar{r}_i^\top \bar{v}_{\mathrm{mm}})$$

$$\leq \exp(-r\Gamma + r\epsilon_{\max})$$

$$\leq e^{rMT \exp(-\Xi R)} e^{-r\Gamma}. \tag{37}$$

● **Sub-Case 1.2: Lower bound for** $\mathcal{L}(v^{(r)}, W^{(R)})$. We now bound the logistic loss for the finite solution. Set $\bar{r}_i = X_i^\top \sigma(X_i W^{(R)} z_i)$. Using Assumption B, solving ($\ell_p$-SVM) on $(y_i, \bar{r}_i)_{i \in [n]}$ achieves at most $\Gamma - \nu e^{-(1-\gamma)\Xi R}/T$ margin. Consequently, we have:

$$\mathcal{L}(v^{(r)}, W^{(R)}) \geq \frac{1}{n} \max_{i \in [n]} \log(1 + \exp(-Y_i \bar{r}_i^\top v^{(r)}))$$

$$\geq \left( \frac{1}{2n} \max_{i \in [n]} \exp(-Y_i \bar{r}_i^\top v^{(r)}) \right) \wedge \log 2$$

$$\geq \left( \frac{1}{2n} \exp(-r(\Gamma - \nu e^{-(1-\gamma)\Xi R}/T)) \right) \wedge \log 2$$

$$\geq \left( \frac{1}{2n} e^{r(\nu/T)\exp(-(1-\gamma)\Xi R)} e^{-r\Gamma} \right) \wedge \log 2.$$

Observe that this lower bound dominates the upper bound from (37) when $R$ is large, specifically when (ignoring the multiplier $1/2n$ for simplicity):

$$(\nu/T)e^{-(1-\gamma)\Xi R} \geq MTe^{-\Xi R} \implies R \geq \frac{1}{\gamma\Xi} \log\left( \frac{MT^2}{\nu} \right).$$

Thus, we obtain the desired contradiction since such a large $R$ is guaranteed to exist when $W^{(R)}/R \nrightarrow \bar{W}_{\mathrm{mm}}$. Therefore, $W^{(R)}/R$ must converge to $\bar{W}_{\mathrm{mm}}/R$.

• **Case 2: Suppose $v^{(r)}/r$ does not converge.** In this case, there exists $\delta > 0$ such that we can find arbitrarily large $r$ obeying $\mathrm{dist}(v^{(r)}/r, \bar{v}_{\mathrm{mm}}/r) \geq \delta$. If $\mathrm{dist}(W^{(R)}/R, \Xi W_{\mathrm{mm}}) \nrightarrow 0$, then "Case 1" applies. Otherwise, we have $\mathrm{dist}(W^{(R)}/R, \Xi W_{\mathrm{mm}}) \to 0$, thus we can assume $\mathrm{dist}(W^{(R)}/R, \Xi W_{\mathrm{mm}}) \leq \epsilon$ for an arbitrary choice of $\epsilon > 0$.

On the other hand, due to the strong convexity of ($\ell_p$-AttSVM), for some $\gamma := \gamma(\delta) > 0$, $v^{(r)}$ achieves a margin of at most $(1-\gamma)\Gamma r$ on the dataset $(Y_i, X_{i1})_{i \in [n]}$, where $X_{i1}$ denotes the optimal token for each $i \in [n]$. Additionally, since $\mathrm{dist}(W^{(R)}/R, \Xi W_{\mathrm{mm}}) \leq \epsilon$, $W^{(R)}$ strictly separates all optimal tokens (for small enough $\epsilon > 0$) and $\hat{q}_{\max} := f(\epsilon) \to 0$ as $R \to \infty$. Note that $f(\epsilon)$ quantifies the non-optimality of $W^{(R)}$ compared to $W_{\mathrm{mm}}$; as $\epsilon \to 0$, meaning $W^{(R)}/R$ converges to $\Xi W_{\mathrm{mm}}/R$, $f(\epsilon) \to 0$. Consequently, setting $r_i = X_i^\top \sigma(X_i W^{(R)} z_i)$, for sufficiently large $R > 0$ and setting $M = \sup_{i \in [n], t \in [T]} \|X_{it}\|$, we have that

$$\min_{i \in [n]} y_i (v^{(r)})^\top r_i \leq \min_{i \in [n]} y_i (v^{(r)})^\top X_{i1} + \sup_{i \in [n]} |(v^{(r)})^\top (X_{it} - X_{i1})|$$

$$\leq (1-\gamma)\Gamma r + M f(\epsilon) r$$

$$\leq (1 - \gamma/2)\Gamma r. \tag{38}$$

This in turn implies that logistic loss is lower bounded by

$$\mathcal{L}(v^{(r)}, W^{(R)}) \geq \left( \frac{1}{2n} e^{\gamma\Gamma r/2} e^{-\Gamma r} \right) \wedge \log 2.$$

Now, using (37), this exponentially dominates the upper bound of $(\bar{W}_{\mathrm{mm}}, \bar{v}_{\mathrm{mm}})$ whenever $rMT \exp(-\Xi R) < r\gamma\Gamma/2$, completing the proof. $\qquad\square$

# H  IMPLEMENTATION DETAILS

The experiments were run on an Intel i7 core and a single V100 GPU using the pytorch and huggingface libraries. They should be runnable on any generic laptop.

## H.1  COMPUTATIONAL OVERHEAD OF ALGORITHM

When compared to the standard gradient descent algorithm, the mirror descent based algorithms, such as the $\ell_p$-MD, $\ell_p$-AttGD, and $\ell_p$-JointGD have additional computational overhead. We claim that the overhead is linear in the size of the parameters for both time complexity and space complexity. For the analysis, we focus on the $\ell_p$-AttGD algorithm, but the same analysis can be applied for the other algorithms.

Let $D = d^2$, the number of entries in $W(k)$. The gradient descent update rule entails the calculation of the gradient $\nabla \mathcal{L}(W(k))$, and then updating the parameters by subtracting from it $\eta$ times that gradient, giving us $W(k) - \eta \nabla \mathcal{L}(W(k))$.

In $\ell_p$-`AttGD`, computing all entries of $[W(k)]^+$ can be done in the following way: The first step

---

**Algorithm 1** Compute $[W(k)]^+$

---

  1: Apply $x \mapsto |x|^{p-1} \operatorname{sign}(x)$ on each entry of $W(k)$ to get $W(k)'$
  2: $[W(k)]^+ \leftarrow W(k)' - \eta \nabla \mathcal{L}(W(k))$

---

takes $O(D)$, while the second step would require the same amount of time as the GD algorithm. For computing $W(k+1)$, we just have to apply the mapping $y \mapsto |y|^{1/(p-1)} \operatorname{sign}(y)$ entry-wise to $[W(k)]^+$, which would take $O(D)$ time. In total, we have an $O(D)$ time overhead, and for space complexity, we can see that we only need to hold a constant amount of additional space for the computation of each entry, so it requires $O(D)$ additional space.

### H.2 $\ell_p$-A T T GD EXPERIMENT

The dataset $(X_i, Y_i, z_i)_{i=1}^n$ is generated randomly: $X_i$ and $z_i$ are sampled from the unit sphere, and $Y_i$ is uniformly sampled from $\{\pm 1\}$. Additionally, $v$ is randomly selected from the unit sphere. We use $n = 6$ samples, $T = 8$ tokens per sample, and $d = 10$ dimensions per token, fulfilling the overparameterized condition for the $\ell_p$-AttSVM problem to be almost always feasible.

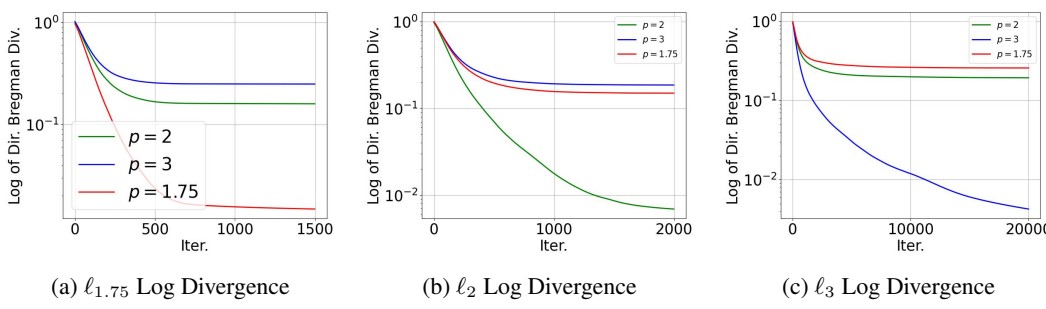

(a) $\ell_{1.75}$ Log Divergence      (b) $\ell_2$ Log Divergence      (c) $\ell_3$ Log Divergence

Figure 10: Graph of the log of directional Bregman divergence between the (a) $\ell_{1.75}$, (b) $\ell_2$, and (c) $\ell_3$ optimization paths and the ($\ell_p$-AttSVM). This shows how the divergence behaves when viewed from the log-space, adding an extra detail to Figure 4

The model is trained with parameters initialized near the origin, using Algorithms $\ell_p$-`AttGD` with $p = 1.75, 2$, and $3$, and a learning rate of $0.1$. Training lasted for $1,500$ epochs for $p = 1.75$, $2,000$ epochs for $p = 2$, and $20,000$ epochs for $p = 3$. Gradients are normalized to accelerate convergence without altering results significantly. We refer to the parameter histories as the $\ell_{1.75}, \ell_2$, and $\ell_3$ optimization paths. We compute the chosen tokens $(\alpha_i)_{i=1}^n$ for the ($\ell_p$-AttSVM) problem by selecting the token with the highest softmax probability for each sample. This process is repeated for $p = 1.75, 2$, and $3$.

### H.3 $\ell_p$-J O I N TGD EXPERIMENT

We use the following example dataset for the experiment on joint optimization.

**Example 2.** *Let $n = 2$, $T = 3$, $d = 2$. Let $y_1 = 1$, $y_2 = -1$. Let:*

$$X_1 = \begin{pmatrix} X_{11} \\ X_{12} \\ X_{13} \end{pmatrix} = \begin{pmatrix} -5.4 & 2.4 \\ 2.8 & 4.2 \\ 2.6 & -0.2 \end{pmatrix}, \quad \text{and} \quad X_2 = \begin{pmatrix} X_{21} \\ X_{22} \\ X_{23} \end{pmatrix} = \begin{pmatrix} 0.8 & -4.4 \\ -2.2 & -0.8 \\ 1.8 & 0.2 \end{pmatrix}. \quad (39)$$

*Let $z_1 = X_{11}$, $z_2 = X_{21}$.*

We use learning rates $0.1$ and we trained the model for $1,500$ epochs for when $p = 1.75$, $2,000$ epochs for $p = 2$, and $20,000$ epochs for $p = 3$. As it was done in the previous experiment, we obtain the parameter history for each $p$, and compute the optimal token for the ($\ell_p$-AttSVM) and $\ell_p$-SVM problems.

## H.4 ARCHITECTURE DETAILS FOR STANFORD LARGE MOVIE REVIEW CLASSIFICATION

The architecture for the model used to perform the semantic analysis task on the Stanford Large Movie Review dataset follows the transformer encoder architecture from Vaswani et al. (2017) with a linear classifier as the last layer.

The embedding layer has trainable token embedding $E$ and position encoding $P$. The model's vocabulary size is 30522 with maximum token length of 512 and embedding dimension 384, so $E \in \mathbb{R}^{30522 \times 384}$ and $P \in \mathbb{R}^{512 \times 384}$. If a token $t$ is in position $i$, its embedding will be $X_i = E_t + P_i$, where $E_t$ and $P_i$ denote the $t^{th}$ and $i^{th}$ rows of $E$ and $P$, respectively.

Then, the token features get passed through the encoding blocks, each of which consists of a multi-head self-attention layer MultiHead, two layer-normalization layers LayerNorm$_1$ and LayerNorm$_2$, and a Multilayer Perceptron layer (MLP). If the sequence of input token features for the encoding block are $X_1, \ldots, X_T$, and if we denote MultiHead$(X_1, \ldots, X_T)_i$ as the $i^{th}$ token feature from the multi-head self-attention, the output of the encoding block for the $i^{th}$ token is LayerNorm$_2(X_i' +$ MLP$(X_i'))$, where $X_i' = $ LayerNorm$_1(X_i + $ MultiHead$(X_1, \ldots, X_T)_i)$. When training the model, we apply dropout of $0.2$ for regularization.

The MultiHead attention is a variant of what is known as the single-head attention that horizontally stacks several instances of single-head attention within the same layer. A single-head attention is equivalent to (2), but with the vector $z$ replaced with the matrix $X^\top$, and the vector $v$ replaced with a matrix $V$.

We experimented with having 3 encoding blocks with 3 attention heads each, 4 encoding blocks with 4 attention heads each, and 6 encoding blocks with 6 attention heads each. Finally, we pass the feature vector of the first token from the last encoding layer into a linear classifier.

## H.5 ADDITIONAL EXPERIMENT WITH ADAM

We run an additional experiment to compare the $\ell_{1.1}$-MD training algorithm with the Adam algorithm. We train a Vision Transformer (ViT) architecture (Dosovitskiy et al., 2020) on CIFAR-10.

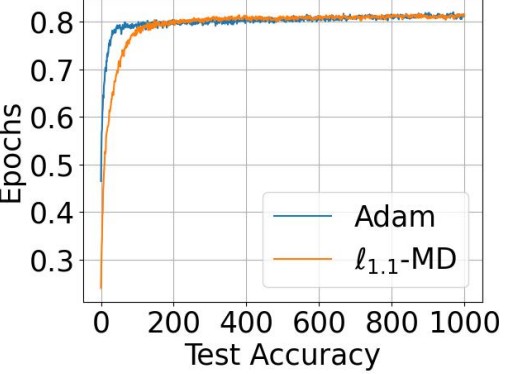

Figure 11: The test accuracy of training a ViT network using $\ell_{1.1}$-MD and Adam (Kingma & Ba, 2014) training algorithms. The resulting test accuracies that the two algorithms approach are similar.

Specifically, the ViT architecture used a patch size of 4, 512 dimensional token feature, with 6 layers of attention blocks and 8 attention heads per attention layer, and a two-layer GeLU network to make the final classification layer on the [CLS] patch token feature that has 512 as the hidden layer size. The embedding layer of the architecture follows the work of Dosovitskiy et al. (2020), where the embedding layer learns the [CLS] token embedding, the linear map for embedding each image patch, and a positional embedding for each possible position on the image. The details of how the attention layers are implemented are similar to that of the architecture used for the Stanford Large Movie dataset, just that we apply the layer normalization before the multihead attention and the MLP instead of after the residual connection. Furthermore, we apply a dropout of $0.1$ when training the network.

The resulting test accuracies for the two algorithms are reported in Figure 11. As we can see, the two algorithms achieve a similar level of test accuracy at the end, which means that our algorithm can reach the SOTA performance.

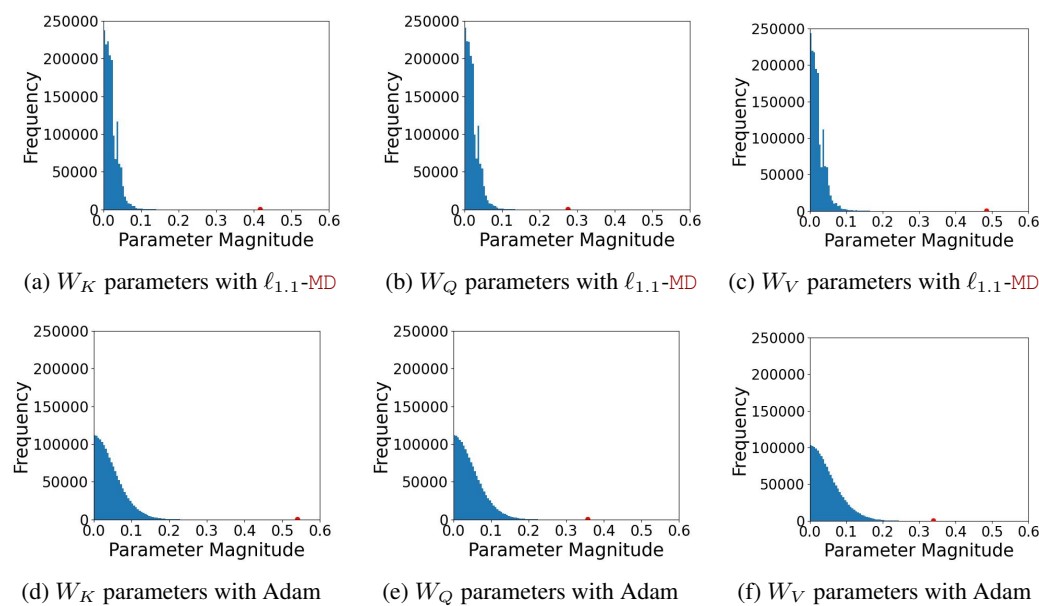

(a) $W_K$ parameters with $\ell_{1.1}$-MD    (b) $W_Q$ parameters with $\ell_{1.1}$-MD    (c) $W_V$ parameters with $\ell_{1.1}$-MD

(d) $W_K$ parameters with Adam    (e) $W_Q$ parameters with Adam    (f) $W_V$ parameters with Adam

Figure 12: Histogram of the absolute values of the $W_K, W_Q,$ and $W_V$ components of ViT models trained with $\ell_{1.1}$-MD and Adam on CIFAR-10. These results show that $\ell_{1.1}$-MD can be more explainable compared to Adam because it produces sparser parameters, which should induce better token selection.

We also plot the weights' absolute value of the resulting two models that were trained by MD and Adam, specifically the weights from the attention layer, which consists of the key, query, and value matrices. Just as it was in Figure 9, $\ell_{1.1}$-MD creates sparser parameter set, this time compared to the one generated by Adam, which shows a potential explanation of $\ell_{1.1}$-MD.

## H.6 Addendum to the Attention Map Results

| Label | Optimal Token | $\ell_{1.1}$-MD Token Selection | GD Token Selection | Better Selector |
|---|---|---|---|---|
| + | fantastic | the movie was fantastic | the movie was fantastic | 1.1 |
| - | hated | i hated the movie | i hated the movie | 1.1 |
| - | boring | the plot was boring | the plot was boring | 2 |
| + | love | i love this movie | i love this movie | 2 |
| - | terrible | the plot was terrible | the plot was terrible | 1.1 |
| + | great | this movie is great | this movie is great | 1.1 |
| - | dirty | the scenes were dirty | the scenes were dirty | 2 |
| + | satisfied | i m satisfied with movie | i ' m satisfied with movie | 2 |
| - | late | the dvd arrived late | the dvd arrived late | 1.1 |
| + | perfectly | the sub ##titles work perfectly | the sub ##titles work perfectly | 1.1 |
| - | disappointing | the movie was disappointing | the movie was disappointing | 1.1 |
| - | unreliable | the pacing is unreliable | the pacing is unreliable | 1.1 |
| + | friendly | the cast were friendly | the cast were friendly | 2 |
| - | slow | the script is slow | the script is slow | 1.1 |
| + | great | the movie was great | the movie was great | 1.1 |
| - | poor | the dvd was poor | the dvd was poor | 1.1 |
| + | fascinating | the plot was fascinating | the plot was fascinating | 1.1 |
| + | sturdy | the set was sturdy | the set was sturdy | 2 |
| - | ruined | the cinematography was ruined | the cinematography was ruined | 1.1 |
| + | engaging | the documentary was engaging | the documentary was engaging | 1.1 |
| - | crashes | the dvd crashes often | the dvd crashes often | 1.1 |
| + | delicious | the scenes were delicious | the scenes were delicious | 1.1 |
| - | broke | the dvd broke down | the dvd broke down | 2 |
| + | prompt | the service was prompt | the service was prompt | 2 |
| - | predictable | the plot was predictable | the plot was predictable | 1.1 |
| + | excellent | the service was excellent | the service was excellent | 2 |
| + | scenic | the theater is scenic | the theater is scenic | 2 |
| - | stopped | the project ##or stopped | the project ##or stopped | 1.1 |
| + | vibrant | the festival was vibrant | the festival was vibrant | 1.1 |
| + | fun | the movie was fun | the movie was fun | 1.1 |
| - | delayed | the screening was delayed | the screening was delayed | 1.1 |

| | | | | |
|---|---|---|---|---|
| + | pleasant | the impact was pleasant | the impact was pleasant | 2 |
| - | unstable | the streaming is unstable | the streaming is unstable | = |
| + | fresh | the snacks are fresh | the snacks are fresh | 2 |
| - | cracked | the dvd cracked | the dvd cracked | 2 |
| + | selection | the theater has selection | the theater has selection | = |
| - | difficult | the interface is difficult | the interface is difficult | 1.1 |
| + | spacious | the cinema is spacious | the cinema is spacious | 2 |
| - | broke | the equipment broke | the equipment broke | 2 |
| + | friendly | the staff are friendly | the staff are friendly | 2 |

Figure 13: The full attention map table that shows that $\ell_{1.1}$-MD provides strictly more attention to the pivotal token compared to $\ell_2$-MD, or equivalently GD, for 22 of the sample sentences. Out of the other 18 sentences, for 16 of which, GD strictly outperforms $\ell_{1.1}$-MD, while for the other 2, the two algorithms are equally as good.

