# OpenReview forum: "Optimizing Attention with Mirror Descent: Generalized Max-Margin Token Selection"
_ICLR.cc/2025/Conference — ICLR 2025 Conference Withdrawn Submission_

### Official Review · Reviewer_7Bsg · 2024-11-04

**Soundness:** 3
**Presentation:** 3
**Contribution:** 3
**Rating:** 6
**Confidence:** 3

**Summary:**

This paper examines how mirror descent (MD) algorithms approach convergence and exhibit bias when optimizing attention mechanisms in softmax attention, with potential functions using $\ell_p$-norms as a basis for analysis. Key theoretical findings include demonstrating convergence towards hard margin Support Vector Machine (SVM) solutions and achieving convergence rates similar to gradient descent, in more straightforward models. The research builds upon findings related to descent by expanding the scope to encompass a wider range of optimization algorithms and shedding light on properties related to token selection.

**Strengths:**

1. The paper has a strong theoretical foundation, it provides rigorous mathematical analysis and proofs for the convergence properties of mirror descent in attention optimization, extending previous work on gradient descent to a more general framework.
2. This paper provides a novel algorithmic insight, the introduction of $\ell_p$-AttGD generalizes both $\ell_p$-GD and attention GD, offering new perspectives on attention optimization and token selection.
3. This work provides a complete theoretical treatment of the optimization dynamics by examining both fixed-head and joint optimization scenarios.

**Weaknesses:**

1. The empirical evaluation is limited, the paper includes experiments, and they are primarily focused on synthetic data and a single real-world dataset (Stanford Large Movie Review Dataset). More diverse real-world applications would strengthen the practical implications.
2. Theoretical results are highly dependent on assumption, Theorem 2, Theorem 3, and Theorem 4, rely on specific assumptions about initialization and step sizes, which may limit their practical applicability.
3. The paper does not thoroughly address the computational overhead of implementing mirror descent compared to standard gradient descent.

**Questions:**

Please refer to the weakness.

---

> ### Author Response · Authors · 2024-11-28
> **Response to Reviewer 7Bsg--Part 1**
>
> > **Weakness 1:** The empirical evaluation is limited; the paper includes experiments, and they are primarily focused on synthetic data and a single real-world dataset (Stanford Large Movie Review Dataset). More diverse real-world applications would strengthen the practical implications.
>
> **Response:** To address the reviewer’s concern, in addition to the language model experiments,  we **provide a new set of empirical evaluations on training a Vision Transformer model** to perform a classification task on CIFAR-10 in Appendix H5. This experiment involves training a Vision Transformer (ViT) model with $\ell_{1.1}$-MD and Adam over the first 1000 epochs. The results in **the newly added Figure 11** show that $\ell_{1.1}$-MD achieves similar test accuracies to Adam, demonstrating that it can match the performance of other state-of-the-art optimizers for transformer models. In addition, to compare the explainability between $\ell_{1.1}$-MD and Adam, we provide the weight distributions of the resulting two models in **the newly added Figure 12** and we provide a discussion on the explainability of $\ell_{1.1}$-MD:
>
> Explainability in attention mechanisms is often defined as the model’s ability to identify and prioritize the most influential tokens in the input sequence, thereby making its decision-making process more interpretable [[Klein et al.](https://arxiv.org/abs/2409.16756), [Ali et al.](https://arxiv.org/abs/2403.01590), [Abnar et al.](https://arxiv.org/abs/2005.00928)] . This aligns with the concept of feature selection in classical machine learning, where sparse and focused representations improve both interpretability and model robustness.
>
> In our work, $\ell_{1.1}$-MD demonstrates superior explainability compared to other gradient-based methods, such as standard (S)GD and Adam. Specifically, $\ell_{1.1}$-MD produces sparser weight distributions and attention maps that more sharply highlight the most critical tokens. Figures 8 and 13 provide clear evidence of this property in the Stanford Large Movie Review Dataset. For instance, attention maps generated by $\ell_{1.1}$-MD-trained models focus more on sentiment-revealing words, such as "amazing" or "terrible," while models trained with GD display more diffuse attention patterns, potentially diluting interpretability. This ability to emphasize pivotal tokens directly contributes to the model's transparency and aligns with established literature emphasizing the importance of sparsity for interpretability.
>
> Furthermore, the weight distributions in the key, query, and value matrices, shown in Figure 9, highlight that $\ell_{1.1}$-MD encourages sparsity more effectively than GD, while the weight distribution in Figure 12 shows that $\ell_{1.1}$-MD also induces more sparsity compared to Adam. This sparsity enhances interpretability by limiting the model's reliance on non-essential tokens. By aligning the optimization process with explainability objectives, $\ell_{1.1}$-MD offers practical benefits for applications where transparency is crucial [[Klein et al.](https://arxiv.org/abs/2409.16756), [Ali et al.](https://arxiv.org/abs/2403.01590), [Abnar et al.](https://arxiv.org/abs/2005.00928)]. Thus, $\ell_{1.1}$-MD achieves comparable generalization performance to Adam (as shown in Figure 11), and its token selection precision and sparse representations establish it as an interpretable and explainable optimization method (as shown in Figures 8, 9, 12, and 13). These findings underscore the potential of using MD variants to improve both the performance and explainability of attention-based models.

---

> ### Author Response · Authors · 2024-11-28
> **Response to Reviewer 7Bsg--Part 2**
>
> > **Weakness 2:** Theoretical results are highly dependent on assumptions. Theorem 2, Theorem 3, and Theorem 4 rely on specific assumptions about initialization and step sizes, which may limit their practical applicability.
>
> **Response:**  Thank you for your comment. Below, we provide detailed clarification for these assumptions:
>
> **The step size condition** aligns with well-established assumptions in the implicit bias literature, particularly for first-order optimization algorithms like gradient descent and mirror descent. Works such as Soudry et al. (2018), Gunasekar et al. (2018), Ji \& Telgarsky (2018), Azizan & Hassibi (2018), Sun et al. (2022; 2023) have similar assumptions and highlight the need for a sufficiently small step size to guarantee convergence. Of note, our step size condition for MD is identical to that in the aforementioned papers. Thus, the step size assumption is not an additional restriction but rather a foundational aspect of theoretical analyses, especially in MD analysis.
>
> **The initialization assumption**, specifically $W(0)$ being constrained within the cone defined in Definition 5, is a distinct and critical feature of this work compared to broader implicit bias studies  (Soudry et al. (2018), Gunasekar et al. (2018), Ji \& Telgarsky (2018), and  Azizan & Hassibi, 2018; Sun et al., 2022; 2023)). Unlike traditional implicit bias studies that often assume broader initializations, the localized initialization considered here is essential for studying convergence in the structured optimization problems encountered in attention mechanisms. Inspired by prior work such as Tarzanagh et al. (2023, 2024), this assumption allows the analysis to focus on the local convergence properties of the parameters towards the desired local max-margin solutions. The necessity of this assumption is further corroborated by Tarzanagh et al. (2024), which establishes that, without initialization within this constrained region, even more robust optimization methods utilizing full gradient information would fail to converge to the max-margin solution.
>
> **Assumption A**, which stipulates that the loss function must be strictly decreasing, differentiable, and have a bounded, Lipschitz continuous derivative, is a standard assumption in optimization theory and implicit bias studies. Widely-used loss functions, such as $l(x) = e^{-x}$, $l(x) = -x$, and $l(x) = \log(1 + e^{-x})$, satisfy this assumption. These functions are not only theoretically convenient but also form the backbone of practical applications in machine learning, particularly in classification and attention-based tasks.
>
> This assumption aligns with the framework of gradient descent dynamics analyzed in works such as Soudry et al. (2018) and Ji \& Telgarsky (2018). It also supports the analysis of mirror descent (MD) algorithms, as demonstrated in Gunasekar et al. (2018),  Azizan & Hassibi, 2018; Sun et al., 2022; 2023). Thus, Assumption A is neither restrictive nor limiting but rather a well-justified component of the theoretical framework.

---

> ### Author Response · Authors · 2024-11-28
> **Response to Reviewer 7Bsg--Part 3**
>
> > **Weakness 3:** The paper does not thoroughly address the computational overhead of implementing mirror descent compared to standard gradient descent.
>
> **Response:** Thank you for the feedback. This is an important point. While the MD family of algorithms, in general, may have a substantial computational overhead compared to GD, the subfamily that we study in our paper have a very small overhead (linear in the number of parameters) and are easily implementable because the update rule is *coordinate-wise separable*. We discuss this below and in the revision in Appendix H1.
>
> The computational overhead is linear in the number of parameters that we are training. To illustrate this, we focus on the $\ell_p$-AttGD algorithm in our paper, and compare its computation requirement to how GD would optimize the same set of parameters. The same analysis can be applied on $\ell_p$-JointGD and $\ell_p$-MD and we would get the same result.
>
> For the analysis, let’s say that the parameter $W$ has $D=d\times d$ entries in total. In GD, we simply had to compute $\eta\nabla\mathcal{L}(W(k))$, and subtract that from the parameter $W(k)$, leading to the update rule $W(k+1)\leftarrow W(k)-\eta\nabla\mathcal{L}(W(k))$.
>
> In $\ell_p$-AttGD, we had to first transform $W(k)$ using the mirror map, which can be done by applying the function $x\mapsto|x|^{p-1}\operatorname{sign}(x)$ element-wise, denote the result of this as $W’(k)$. This first operation would require an additional $\mathcal{O}(D)$ time. Then, similarly as in GD, we had to compute $\eta\nabla\mathcal{L}(W(k))$ and subtract that from $W’(k)$, denote this new result by $W(k)^+$. This has identical complexity to that of the GD algorithm. Finally, we need to apply the inverse mirror-map function on $W(k)^+$ and save the result to the parameters in the models. This operation can be done by applying $y\mapsto|y|^{1/(p-1)}\operatorname{sign}(y)$ element-wise, so it will take an additional $\mathcal{O}(D)$ time. In total, our algorithm requires an additional $\mathcal{O}(D)$ time. For space complexity, we can see that we only need to hold a constant amount of additional space for the computation of each entry, so it requires $\mathcal{O}(D)$ additional space.
>
> Although this adds some overhead compared to GD, its overhead should be similar to that of common adaptive methods such as Adam, and RMSProp.

---

> > ### Author Response · Authors · 2024-12-02
> >
> > Dear Reviewer,
> >
> > Thank you again for your time in reviewing our paper.
> >
> > If our response has addressed your concerns, we would be grateful if you could re-evaluate our work.
> >
> > If you have any additional questions or comments, we would be happy to have further discussions.

---

> > > ### Comment · Reviewer_7Bsg · 2024-12-03
> > >
> > > Thanks for the detailed answers and the additional experiments. I propose the following:
> > > 1. Limited empirical variety: This is a welcome experiment, but the empirical evaluation is still rather constrained, involving only synthetic data and two real-world datasets, CIFAR-10 and CelebA.
> > > 2. Dependence on Theoretical Assumptions: This aligns with previous research in that the analysis hinges on particular premises including restricted initialization and moderate step sizes.
> > > 3. Computational Overhead: Based on the data provided by the authors, it is possible to claim that the overhead is comparable to adaptive techniques like Adam, but more comparisons in the real training conditions would be beneficial.

---

> > > > ### Author Response · Authors · 2024-12-04
> > > > **Response to Reviewer 7Bsg**
> > > >
> > > > Thank you for the valuable feedback. We address them as follows.
> > > >
> > > > > Limited empirical variety
> > > >
> > > > **Response:** Thank you for your comment. We have already conducted experiments in two synthetic settings (as shown in Figures 4, 5, 6, 7, and 10) and two real-world settings for language and vision tasks (as shown in Figures 8, 9, 11, 12, 13, and Table 1). To further strengthen our experimental findings, we have included additional results on the more complex CIFAR-100 dataset, which will be presented in the appendix of the final manuscript.
> > > >
> > > > The setting of this experiment is similar to that of the experiment on CIFAR-10 (training a ViT architecture using Adam and $\ell_{1.1}$-MD for 1000 epochs), with the difference being that the output is a probability distribution on $100$ classes instead of $10$. We get a similar result in terms of the weight distribution of the attention weights in the model, namely that the **$\ell_{1.1}$-MD trained model has sparser weight distribution** when compared to the model trained with Adam. However, we get a higher test accuracy for the model trained by $\ell_{1.1}$-MD by approximately **2.19%** compared to Adam. Specifically, we get **55.22%** from $\ell_{1.1}$-MD and **53.03%** from Adam. This is further detailed in the following table, which shows the test accuracy at every 50 epochs:
> > > >
> > > > | Epochs | Adam | $\ell_{1.1}$-MD |
> > > > | -- | -- | -- |
> > > > | 50 | 0.51710 | 0.46620 |
> > > > | 100 | 0.52310 | 0.51470 |
> > > > | 150 | 0.51770 | 0.52590 |
> > > > | 200 | 0.52350 | 0.53150 |
> > > > | 250 | 0.52510 | 0.53480 |
> > > > | 300 | 0.52300 | 0.53750 |
> > > > | 350 | 0.52640 | 0.53860 |
> > > > | 400 | 0.52770 | 0.53670 |
> > > > | 450 | 0.52150 | 0.53930 |
> > > > | 500 | 0.52860 | 0.54150 |
> > > > | 550 | 0.52800 | 0.54200 |
> > > > | 600 | 0.52480 | 0.54300 |
> > > > | 650 | 0.52470 | 0.54180 |
> > > > | 700 | 0.52940 | 0.54870 |
> > > > | 750 | 0.53200 | 0.54510 |
> > > > | 800 | 0.52940 | 0.54560 |
> > > > | 850 | 0.52950 | 0.54930 |
> > > > | 900 | 0.53150 | 0.54780 |
> > > > | 950 | 0.52720 | 0.54680 |
> > > > | 1000 | 0.53030 | 0.55220 |
> > > >
> > > > > Dependence on Theoretical Assumptions
> > > >
> > > > **Response:** Thank you for the feedback, yes our assumption mainly aligns with the recent works on implicit bias analysis of optimization algorithms for attention [[Tarzanagh et al.](https://arxiv.org/abs/2308.16898), [Vasudeva et al.](https://arxiv.org/abs/2402.05738), [Sheen et al.](https://arxiv.org/abs/2403.08699)] and implicit bias for mirror descent for general machine learning problems [[Sun et al.](https://www.jmlr.org/papers/v24/23-0836.html), [Azizan et al.](https://arxiv.org/abs/1906.03830), [Azizan et al.](https://openreview.net/pdf?id=HJf9ZhC9FX)], and we hope that future works can relax these assumptions.
> > > >
> > > > > Computational Overhead
> > > >
> > > > **Response:** To follow your suggestion, we will include the runtime of our algorithms for the real-data experiments. As analyzed in our previous response and the revised manuscript, the overhead time and space complexity for $\ell_p$-MD compared to GD is small (linear in the number of parameters trained) and is comparable to the overhead of Adam relative to GD.

---

### Official Review · Reviewer_vf65 · 2024-11-04

**Soundness:** 2
**Presentation:** 2
**Contribution:** 2
**Rating:** 3
**Confidence:** 3

**Summary:**

This paper investigates the optimization dynamics of mirror descent (MD) for training attention mechanisms, specifically focusing on $l_p$-AttGD. The authors claim that $l_p$-AttGD converges directionally to a generalized hard-margin SVM with an $l_p$ norm objective when applied to binary classification with a single-layer attention model. Some experiments on synthetic and real data are presented.

**Strengths:**

The paper attempts to provide a theoretical analysis of mirror descent for attention training, extending prior work focused on gradient descent. It derives convergence results to a generalized hard-margin SVM and establishes convergence rates. The use of $l_p$ norms offers a degree of generality in the theoretical analysis.

**Weaknesses:**

1. The core idea of connecting attention optimization to SVM-like objectives is not new and has been explored in prior work, notably in "A Primal-Dual Framework for Transformers and Neural Networks" by Nguyen et al. and related papers. These prior works establish the fundamental link between attention and SVMs, including the optimization perspective. While this paper extends the analysis to mirror descent, the incremental contribution feels minimal and lacks motivation. Other core ideas of analyzing the implicit bias of GD/MD algorithms for softmax attention is already present in the cited works (Tarzanagh et al., 2023, 2024; Vasudeva et al., 2024a; Sheen et al., 2024) without fundamentally changing the nature of the problem or leading to significantly different insights.

2. The synthetic experiments are too simplistic and lack the complexity needed to represent realistic attention training scenarios. The real-data experiments, while showing some improvements in generalization, are insufficient to support the claims. The demonstration of improved token selection is based on a handful of examples (Figure 8) without any rigorous quantitative evaluation. Comparisons with GD do not include other commonly used optimization algorithms like Adam, making it impossible to judge the relative merits of MD for attention training.

**Questions:**

See weaknesses.

---

> ### Author Response · Authors · 2024-11-27
> **Response to Reviewer vf65--Part 1**
>
> > **Weakness 1-1:** The core idea of connecting attention optimization to SVM-like objectives is not new and has been explored in prior work, notably in "A Primal-Dual Framework for Transformers and Neural Networks" by Nguyen et al. and related papers. These prior works establish the fundamental link between attention and SVMs, including the optimization perspective. While this paper extends the analysis to mirror descent, the incremental contribution feels minimal and lacks motivation.
>
> **Response:**  Thank you for the feedback. However, we respectfully disagree with the assertion that our contribution is minimal or lacks novelty. While Nguyen et al. (2023) explore the connection between attention mechanisms and support vector regression (SVR)-like objectives, their focus is fundamentally different from ours. Below, we outline why our contributions are not only distinct but also advance the state-of-the-art in attention optimization significantly:
>
> 1. **We provide the optimization dynamics, implicit bias, and convergence rate analysis**: Nguyen et al. focus primarily on **static** primal-dual formulations for attention mechanisms, connecting self-attention to support vector regression (SVR) through a primal-dual formulation. **They do not examine  optimization dynamics, the role of descent algorithms, or the implications of implicit bias in training**. Our work explicitly targets the optimization process of attention mechanisms under mirror descent (MD). We provide a comprehensive analysis of convergence dynamics, implicit bias, and token selection properties, which are absent in prior work.
>
> 2. **We show that different optimizers have different token selection properties**:  Unlike Nguyen et al., we provide actionable insights into how different optimizers have different token selection properties, which in turn leads to different generalization performances, **even when training the same architecture on the same dataset**.
>
> 3. Finally, we have added the following related works to the revised manuscript:
>
>  "*Nguyen et al. (2024) provided static primal-dual formulations for attention mechanisms, connecting self-attention to support vector regression (SVR) through a primal-dual framework. Nguyen et al. (2022) connects self-attention to kernel methods to enhance Transformers. Chen et al. (2024b) provided a novel attention mechanism that optimizes self-attention in Transformers using asymmetric Kernel Singular Value Decomposition (KSVD) in the primal representation, achieving improved efficiency and performance through low-rank approximations and regularization techniques. However, these works do not examine optimization dynamics, the role of descent algorithms, or the implications of implicit bias in training, which are the main focus of this work.*"
>
> The following items have been added to the reference list in the revised manuscript:
>
> - *Nguyen et al. (2024)*:  Nguyen, Tan Minh, Tam Nguyen, Nhat Ho, Andrea L. Bertozzi, Richard Baraniuk, and Stanley Osher. "A Primal-Dual Framework for Transformers and Neural Networks." In The Eleventh International Conference on Learning Representations (ICLR), 2023. 2023.
>
> - *Nguyen et al. (2022)*:  Nguyen, Tam Minh, Tan Minh Nguyen, Dung DD Le, Duy Khuong Nguyen, Viet-Anh Tran, Richard Baraniuk, Nhat Ho, and Stanley Osher. "Improving transformers with probabilistic attention keys." In International Conference on Machine Learning, pp. 16595-16621. PMLR, 2022.
>
> - *Chen et al. (2024b)*:  Chen, Yingyi, Qinghua Tao, Francesco Tonin, and Johan Suykens. "Primal-attention: Self-attention through asymmetric kernel svd in primal representation." Advances in Neural Information Processing Systems 36 (2024).

---

> ### Author Response · Authors · 2024-11-28
> **Response to Reviewer vf65--Part 2**
>
> > **Weakness 1-2:** Other core ideas of analyzing the implicit bias of GD/MD algorithms for softmax attention are already present in the cited works (Tarzanagh et al., 2023, 2024; Vasudeva et al., 2024a; Sheen et al., 2024) without fundamentally changing the nature of the problem or leading to significantly different insights.
>
> **Response:** We acknowledge that there is existing literature on training attention models using GD. However, to the best of our knowledge, the papers you mentioned (Tarzanagh et al., 2023, 2024; Vasudeva et al., 2024a; Sheen et al., 2024) have not conducted this analysis for MD, nor have they performed a local convergence rate analysis for either MD or GD. The analysis involves significant technical challenges and heavy lifting to extend the results to MD, which is substantially different from the previously cited works. Furthermore the results do lead to significantly new insights. We elaborate on the difference between our work and those papers below.
>
> 1. (Tarzanagh et al., 2023, 2024) was, to our knowledge, the first to explore the implicit bias of attention mechanism, and they have done their analysis for GD. **Our work extends their result for locally optimal tokens for MD and shows that different training algorithm can induce different implicit biases**. In addition to the new result, we also had to develop a different analysis technique because different tools were needed to analyze MD, such as Bregman divergence. Due to the generality of the Bregman divergence, we could not prove that the parameters will stay around the locally optimal direction in the same way that Tarzanagh did, which was by showing that the parameters will always approach that optimal direction. Instead, we had to show that if the parameters were initially close around the locally optimal direction, then it would not stray too far from the locally optimal direction.
>
> 2. (Vasudeva et al., 2024a) explored the rate of global convergence of training the attention model using GD for joint optimization and $W$ optimization. Meanwhile, we focused on the $W$ optimization for MD, looking into both its implicit bias and convergence rate. Furthermore, Vasuveda et al. had to rely on an additional assumption on the near-orthogonality of the token features, while we did not have to rely on that. **Most importantly, while Vasudeva et al. theoretically proved the result for normalized GD and GD with Polyak step-sizes, and empirically investigated those as well as Adam, they have not shown that these different optimization algorithms have different implicit biases and token selection properties.** However, we do so both theoretically and empirically.
>
>
> 3. (Sheen et al., 2024) examined the implicit bias of gradient flow for self-attention, providing the first theoretical analysis of first-order algorithms for the $(Q,K)$-parameterization in self-attention. However, while gradient flow is a valuable tool in analyzing training algorithms, the practical applicability of these results may be limited, as gradient flow is not typically considered a practical algorithm for training ML models in general.
>
> 4. Finally, we would like to emphasize that our introduction of $\ell_p$-AttGD and the extension of attention optimization to generalized $\ell_p$-norm objectives provide entirely new theoretical results, including convergence rates under Bregman divergence and insights into token selection properties. These results are absent in the aforementioned papers.

---

> ### Author Response · Authors · 2024-11-28
> **Response to Reviewer vf65--Part 3**
>
> > **Weakness 2:** The synthetic experiments are too simplistic and lack the complexity needed to represent realistic attention training scenarios. The real-data experiments, while showing some improvements in generalization, are insufficient to support the claims. The demonstration of improved token selection is based on a handful of examples (Figure 8) without any rigorous quantitative evaluation. Comparisons with GD do not include other commonly used optimization algorithms like Adam, making it impossible to judge the relative merits of MD for attention training.
>
> We have already provided real-data experiments on the Stanford Large Movie Review Dataset, showcasing the performance of $\ell_{1.1}$-MD on a language-based task. **These include qualitative evaluations (Figures 8 and 13) and **quantitative results** (Table 1 and Figure 9), demonstrating that $\ell_{1.1}$-MD achieves competitive generalization and superior explainability compared to GD**.
>
> **To address concerns about diversity, we have added experiments on Vision Transformers (ViTs) for a vision-based task (details in Appendix H5)**. Our new ViT experiment demonstrates that $\ell_{1.1}$-MD achieves comparable test accuracy to Adam over the first 1000 epochs. Our result in Figure 11 confirms that $\ell_{1.1}$-MD matches Adam's test accuracy, validating its effectiveness across both language and vision domains. In addition, Figure 12 provides a histogram of the weights in the models trained by Adam and $\ell_{1.1}$-MD, showing sparser weight distribution from the one trained using $\ell_{1.1}$-MD.  **This finding highlights that $\ell_{1.1}$-MD is not only competitive but also offers additional advantages, particularly improved explainability, as discussed below.**
>
> Explainability in attention mechanisms is often defined as the model’s ability to identify and prioritize the most influential tokens in the input sequence, thereby making its decision-making process more interpretable [[Klein et al.](https://arxiv.org/abs/2409.16756), [Ali et al.](https://arxiv.org/abs/2403.01590), [Abnar et al.](https://arxiv.org/abs/2005.00928)] . This aligns with the concept of feature selection in classical machine learning, where sparse and focused representations improve both interpretability and model robustness.
>
> In our work, $\ell_{1.1}$-MD demonstrates superior explainability compared to other gradient-based methods, such as standard (S)GD and Adam. Specifically, $\ell_{1.1}$-MD produces sparser weight distributions and attention maps that more sharply highlight the most critical tokens. Figures 8 and 13 provide clear evidence of this property in the Stanford Large Movie Review Dataset. For instance, attention maps generated by $\ell_{1.1}$-MD-trained models focus more on sentiment-revealing words, such as "amazing" or "terrible," while models trained with GD display more diffuse attention patterns, potentially diluting interpretability. This ability to emphasize pivotal tokens directly contributes to the model's transparency and aligns with established literature emphasizing the importance of sparsity for interpretability.
>
> Furthermore, the weight distributions in the key, query, and value matrices, shown in Figure 9, highlight that $\ell_{1.1}$-MD encourages sparsity more effectively than GD, while the weight distribution in Figure 12 shows that $\ell_{1.1}$-MD also induces more sparsity compared to Adam. This sparsity enhances interpretability by limiting the model's reliance on non-essential tokens. By aligning the optimization process with explainability objectives, $\ell_{1.1}$-MD offers practical benefits for applications where transparency is crucial [[Klein et al.](https://arxiv.org/abs/2409.16756), [Ali et al.](https://arxiv.org/abs/2403.01590), [Abnar et al.](https://arxiv.org/abs/2005.00928)]. **Thus, $\ell_{1.1}$-MD achieves comparable generalization performance to Adam (as shown in Figure 11), and its token selection precision and sparse representations establish it as an interpretable and explainable optimization method (as shown in Figures 8, 9, 12, and 13)**. These findings underscore the potential of using MD variants to improve both the performance and explainability of attention-based models.

---

> > ### Author Response · Authors · 2024-12-02
> >
> > Dear Reviewer,
> >
> > Thank you again for your time in reviewing our paper.
> >
> > If our response has addressed your concerns, we would be grateful if you could re-evaluate our work.
> >
> > If you have any additional questions or comments, we would be happy to have further discussions.

---

> > > ### Comment · Reviewer_vf65 · 2024-12-02
> > >
> > > I thank the authors for their response. It is nice to see that you can get improved sparsity at same accuracy than Adam. I have the following suggestions
> > > - I believe this result warrants further exploration, as it now seems tangential to the main focus of the paper. It could strengthen the motivation on the benefits of the proposed algorithm compared to what is well-established in the field.
> > > - Could the obtained sparsity be related to support vector sparsity in the SVM?
> > > - At the same time, the current manuscript presents some exaggerated claims, such as "we provide extensive numerical experiments on real and synthetic data [...] excelling in optimal token selection and suppressing non-optimal tokens" in line 88, while the provided numerical experiments are not extensive and further evidence is required to claim that the proposed method  is useful in token selection.
> > > - Regarding previous work on analyzing attention through the lens of SVMs, I still encourage the authors to better explain the differences, as the cited works also deal with optimization problems. Of course, once fixed the SVM attention formulation, one could devise multiple solving algorithms with different properties, but then the contribution is unclear w.r.t. what is specific to attention or rather a general algorithm that is applied to this specific case.

---

> > > > ### Author Response · Authors · 2024-12-04
> > > > **Response to Reviewer vf65**
> > > >
> > > > Thank you for the feedback. We address them below.
> > > >
> > > > > I thank the authors for their response. It is nice to see that you can get improved sparsity at same accuracy than Adam. I have the following suggestions : I believe this result warrants further exploration, as it now seems tangential to the main focus of the paper. It could strengthen the motivation on the benefits of the proposed algorithm compared to what is well-established in the field.
> > > >
> > > > **Response:**  We thank the reviewer for their insightful comment. As Reviewer HzdV noted, “The extension to $\ell_p$-norm … opens up new avenues for optimizing attention mechanisms.” We agree that further experimental investigations and optimization methods could be explored in future works, as such extensions fall outside the scope of this paper.
> > > >
> > > > The primary goal of the current work is to provably demonstrate how different optimization algorithms exhibit **distinct implicit biases**, leading to variations in **convergence rates**, **generalization**, and **explainability** within the same attention model. This is supported by our experimental results on the synthetic dataset (Figures 4--7, 10), text semantic analysis dataset (Figures 8, 9, 13, and Table 1), and image classification dataset (Figures 11 and 12).
> > > >
> > > > To further enhance the experimental findings, we present additional results on CIFAR-100, which will be included in the appendix of the final manuscript.
> > > >
> > > > The setting of this experiment is similar to that of the experiment on CIFAR-10 (training a ViT architecture using Adam and $\ell_{1.1}$-MD for 1000 epochs), with the difference being that the output is a probability distribution on $100$ classes instead of $10$. We get a similar result in terms of the weight distribution of the attention weights in the model, namely that the **$\ell_{1.1}$-MD trained model has sparser weight distribution** when compared to the model trained with Adam. However, we get a higher test accuracy for the model trained by $\ell_{1.1}$-MD by approximately **2.19%** compared to Adam. Specifically, we get **55.22%** from $\ell_{1.1}$-MD and **53.03%** from Adam. This is further detailed in the following table, which shows the test accuracy at every 50 epochs:
> > > >
> > > > | Epochs | Adam | $\ell_{1.1}$-MD |
> > > > | -- | -- | -- |
> > > > | 50 | 0.51710 | 0.46620 |
> > > > | 100 | 0.52310 | 0.51470 |
> > > > | 150 | 0.51770 | 0.52590 |
> > > > | 200 | 0.52350 | 0.53150 |
> > > > | 250 | 0.52510 | 0.53480 |
> > > > | 300 | 0.52300 | 0.53750 |
> > > > | 350 | 0.52640 | 0.53860 |
> > > > | 400 | 0.52770 | 0.53670 |
> > > > | 450 | 0.52150 | 0.53930 |
> > > > | 500 | 0.52860 | 0.54150 |
> > > > | 550 | 0.52800 | 0.54200 |
> > > > | 600 | 0.52480 | 0.54300 |
> > > > | 650 | 0.52470 | 0.54180 |
> > > > | 700 | 0.52940 | 0.54870 |
> > > > | 750 | 0.53200 | 0.54510 |
> > > > | 800 | 0.52940 | 0.54560 |
> > > > | 850 | 0.52950 | 0.54930 |
> > > > | 900 | 0.53150 | 0.54780 |
> > > > | 950 | 0.52720 | 0.54680 |
> > > > | 1000 | 0.53030 | 0.55220 |
> > > >
> > > > > Could the obtained sparsity be related to support vector sparsity in the SVM?
> > > >
> > > > **Response:** This is indeed a direct consequence of our theorems.  Specifically as it was shown in our theorems, $\ell_{p}$-AttGD converges to $\ell_{p}$-AttSVM solutions.  Further, since $\ell_p$-AttSVM solutions would be sparse when $p$ is close to $1$, this shows that, for example, $\ell_{1.1}$-MD converges to sparser solution compared to GD ($\ell_2$-MD) and Adam as provided by our experiments in Figure 9 and 12.
> > > >
> > > > > At the same time, the current manuscript presents some exaggerated claims … in token selection
> > > >
> > > > **Response:** To address your concern, we removed the word “extensive” from Line 88.
> > > >
> > > > > Regarding previous work on analyzing attention through the lens of SVMs, … case
> > > >
> > > > **Response:** We thank the reviewer for their feedback and would like to clarify that **our work does not aim to design or solve different SVM formulations**. Instead, our focus is on analyzing the implicit bias of optimization algorithms, specifically Mirror Descent (MD), in the context of training attention mechanisms.
> > > >
> > > > *Implicit bias* refers to the inherent tendency of optimization algorithms to select specific solutions from among the infinitely many parameter configurations that can achieve zero training error, particularly in non-convex settings. Our study highlights how the implicit biases of MD influence the specific properties of attention weights, such as sparsity, robustness, and generalization, offering insights that go beyond standard optimization behaviors e.g., the behavior of standard (stochastic) gradient descent for attention. We hope this clarification resolves the reviewer’s concern and emphasizes that the unique focus of our work is on implicit bias analysis, not on SVM design or solving.

---

### Official Review · Reviewer_kVFQ · 2024-11-04

**Soundness:** 2
**Presentation:** 4
**Contribution:** 2
**Rating:** 3
**Confidence:** 4

**Summary:**

This paper proposes to study the dynamics of a simplified one-layer attention network. The implicit bias of gradient descent for such a problem is already known, this paper proposes to study what happens when the training algorithm is mirror descent with p-norms. The paper extends previous results known for gradient descent to this setting, and then demonstrate that using mirror descent to train a transformer on a sentiment analysis dataset yields better generalization.

**Strengths:**

- The paper is well written
- It is quite easy to understand
- The fact that MD achieves sparser weights and leads to better generalization is interesting.

**Weaknesses:**

- The motivation is quite unclear. “A broader understanding of general descent algorithms, including the mirror descent (MD) family and their token selection properties, is essential.” why? In practice, nobody trains attention layers with mirror descent. The observation that it works better than gradient descent is not very strong in my opinion, because in practice transformers cannot be trained with gradient descent either. In the experiments, it would be worthwhile to compare the proposed methods to the widely used adam algorithm. Also, providing train losses values for the real data experiment would be enlightening.
- There are many unclear things about global and optimal tokens (def.3). First, the fact that there is a feasible solution is very important, precise conditions for it to happen must be put in the paper. The authors simply state that “under mild overparameterization d ≥ max{T − 1, n}, the problem is almost always feasible (Tarzanagh et al., 2023, Theorem 1).” These conditions must be put in the paper. Also, it is unclear whether the statement is “for all $\alpha_i$, the problem is feasible” or “the problem is feasible for at least one set of $\alpha_i$.
- I cannot understand why Theorem 1 does not require those conditions: clearly, $W^{opt}_{mm}$ must exist for it to hold, and there is nothing in the assumptions guaranteeing it.
- It is never properly stated in the paper if “globally optimal tokens” are also “locally optimal”. It seems not to be the case, since globally optimal tokens depend on $v$ while locally optimal tokens do not. This also causes issues in thm 1: globally optimal tokens do not seem to be always feasible, so W^{opt}_{mm} is not always defined, while the lp-AttRp always has a solution. For instance, it could very well have a global solution of fixed norm.

**Questions:**

- Can the authors clarify the questions above regarding feasibility?
- Can the authors compare the training curves with that of adam?
- The model sizes do not seem to be specified. The architectural details should be put in the appendix


**Minor remarks and typos**

- L68: ERM refers to an equation much later in the manuscript
- L71: W(k) has not bee introduced thus far
- L113: the softmax acts here on a matrix, but later in (2) it acts on vectors.
- L139: (Blair, 1985)
- L165: (Tarzanagh et al. (2024; 2023)).
- L165: the sense of nearby is unclear here.
- L169: it would be good to specify to which set the indices belong. Generally, it would be clearer to indicate in which space lives each newly introduced variable.
- L176: “the αi component has a significantly higher probability” : this statement should also involve the norm of W, right? Because if all values in the pre-softmax vector are high, then the probability vector will be almost constant.
- Eq 4b: typo W -> W’
- L280: $exp(2)$ could be $\exp(2)$
- In eq.5, it would be good to recall that $D_\psi$ depends on $p$.
- All figures should be in pdf format
- “In the Appendix G, we show that W and v generated by ℓp-JointRP converge to their respective…” it would strengthen that section to put it in the main text.
- Fig4: a log scale would make things clearer, especially highlighting the practical convergence rates.

---

> ### Author Response · Authors · 2024-11-27
> **Response to Reviewer kVFQ--Part 1**
>
> > **Weakness 1:** The motivation is quite unclear. ... Also, providing train loss values for the real data experiment would be enlightening.
>
> > **Question 2:** Can the authors compare the training curves with that of Adam?
>
> Thank you for your comment and question. We have revised the quoted sentence to clarify the motivation, and we address your concerns below.
>
> **Motivation**: The central motivation of our work is to demonstrate that different optimizers exhibit distinct token selection properties, converging to different SVM solutions and thereby leading to varying generalization performances. While **MD is not yet widely adopted for training transformers, it has been extensively studied for its implicit bias properties in standard deep neural networks**; see Section 1.2 in [[Sun et al.](https://www.jmlr.org/papers/v24/23-0836.html)] for an overview as well as [[Azizan et al.](https://arxiv.org/abs/1906.03830), [Azizan et al.](https://openreview.net/pdf?id=HJf9ZhC9FX), [Gunasekar et al.](https://arxiv.org/abs/1802.08246)].  Notably, many adaptive algorithms, including AdaGrad (please refer to Section 1.1 in [[Duchi et. al](https://www.jmlr.org/papers/volume12/duchi11a/duchi11a.pdf)] regarding connection between AdaGrad and MD), can be viewed as variants of MD with different potential functions.
>
> As Reviewer HzdV pointed out, this insight opens up a novel direction for exploring alternative optimization algorithms tailored for attention mechanisms. The implicit bias of MD can significantly influence token selection properties, thereby impacting both generalization and interpretability.
>
> **Additional Experiment on Vision Transformers**: To address the reviewer’s concern regarding comparisons to widely used algorithms like Adam, we have added an experiment in Appendix H5 to further validate the performance of $\ell_{1.1}$-MD. This experiment involves training a Vision Transformer (ViT) model with $\ell_{1.1}$-MD and Adam over the first 1000 epochs. The results in Figure 11 shows that $\ell_{1.1}$-MD achieves similar test accuracies to Adam, demonstrating that it can match the performance of state-of-the-art optimizers for transformer models. This finding underscores that $\ell_{1.1}$-MD is not only competitive but also offers additional advantages, such as improved explainability, as discussed below.
>
> **Explainability Through Mirror Descent**:  Explainability in attention mechanisms is often defined as the model’s ability to identify and prioritize the most influential tokens in the input sequence, thereby making its decision-making process more interpretable [[Klein et al.](https://arxiv.org/abs/2409.16756), [Ali et al.](https://arxiv.org/abs/2403.01590), [Abnar et al.](https://arxiv.org/abs/2005.00928)] . This aligns with the concept of feature selection in classical machine learning, where sparse and focused representations improve both interpretability and model robustness. In our work, $\ell_{1.1}$-MD demonstrates superior explainability compared to other traditional optimization methods. Specifically, $\ell_{1.1}$-MD produces sparser weight distributions and attention maps that more sharply highlight the most critical tokens. Figures 8 and 13 provide clear evidence of this property in the Stanford Large Movie Review Dataset. For instance, attention maps generated by $\ell_{1.1}$-MD-trained models focus more on sentiment-revealing words, such as "amazing" or "terrible," while models trained with GD display more diffuse attention patterns, potentially diluting interpretability. This ability to emphasize pivotal tokens directly contributes to the model's transparency and aligns with established literature emphasizing the importance of sparsity for interpretability [[Sun et al.](https://www.jmlr.org/papers/v24/23-0836.html), [Azizan et al.](https://arxiv.org/abs/1906.03830), [Azizan et al.](https://openreview.net/pdf?id=HJf9ZhC9FX)].
>
> Furthermore, the weight distributions in the key, query, and value matrices, shown in Figure 9, highlight that $\ell_{1.1}$-MD encourages sparsity more effectively than GD, while the weight distribution in Figure 12 shows that $\ell_{1.1}$-MD also induces more sparsity compared to Adam. This sparsity enhances interpretability by limiting the model's reliance on non-essential tokens. By aligning the optimization process with explainability objectives, $\ell_{1.1}$-MD offers practical benefits for applications where transparency is crucial [[Klein et al.](https://arxiv.org/abs/2409.16756), [Ali et al.](https://arxiv.org/abs/2403.01590), [Abnar et al.](https://arxiv.org/abs/2005.00928)]. Thus, **while $\ell_{1.1}$-MD achieves comparable generalization performance to Adam (as demonstrated in our experiments), its token selection precision and sparse representations establish it as an interpretable and explainable optimization method**. These findings underscore the potential of using MD variants to improve the performance and explainability of attention-based models.

---

> ### Author Response · Authors · 2024-11-27
> **Response to Reviewer kVFQ--Part 2**
>
> > **Weakness 2:** There are many unclear things about global and optimal tokens (def.3). First, the fact that there is a feasible solution is very important, precise conditions for it to happen must be put in the paper. The authors simply state that “under mild overparameterization and $d \geq \max(T-1, n)$, the problem is almost always feasible (Tarzanagh et al., 2023, Theorem 1).” These conditions must be put in the paper. Also, it is unclear whether the statement is “for all $\alpha_i$, the problem is feasible” or “the problem is feasible for at least one set of $\alpha_i$.”
>
> > **Weakness 3:** I cannot understand why Theorem 1 does not require those conditions: clearly, $W_{\text{opt}}^{\text{mm}}$ must exist for it to hold, and there is nothing in the assumptions guaranteeing it.
>
> > **Question 1:** Can the authors clarify the questions above regarding feasibility?
>
> **Response:** Thank you for pointing out this particular issue.  We agree that it is important to explicitly state the assumptions under which the problem is feasible. To address this, we have revised the paper to include these conditions directly below Definition 4. Specifically, **we have included the following assumptions in the revised manuscript:**
>
> "Throughout this paper, we assume feasibility, which means there exists a matrix $W$ that linearly separates the logits $X_{i \alpha_i}^T W z_i$ from the logits $X_{it}^T W z_i$ for all $t \in [T] \setminus {\alpha_i}$ and $i \in [n]$."
> Additionally, we specify that:
>
> "Under mild overparameterization, where $d \geq \max(T-1, n)$, the problem is almost always feasible (Tarzanagh et al., 2023, Theorem 1)."
>
> Please refer to the blue part below Definition 4.
>
> > **Weakness 4:** It is never properly stated in the paper if “globally optimal tokens” are also “locally optimal.” It seems not to be the case, since globally optimal tokens depend on $v$ while locally optimal tokens do not. This also causes issues in Theorem 1: globally optimal tokens do not seem to be always feasible, so $W_{\text{mm}}^{\text{opt}}$ is not always defined, while the $\ell_p$-AttRp always has a solution. For instance, it could very well have a global solution of fixed norm.
>
> **Response:** Thank you for your thoughtful feedback. As we clarified in our **response to Question 1**, the feasibility of the SVM problem in Definition 4 is assumed throughout the paper. Regarding the relationship between globally optimal tokens $\text{opt}_i$ and locally optimal tokens $\alpha_i$, globally optimal tokens inherently satisfy the conditions for locally optimal tokens as defined in Definition 3.
>
> By substituting $\alpha_i$ with $\text{opt}_i$ in Definition 3 (Item 2), the resulting optimization problem remains consistent, and the token scores $\gamma\_{i\text{opt}_i}$ are at least as high as those of any support token for sample $i$. This follows directly from the definition, where the globally optimal token score is greater than or equal to the scores of all other tokens, including the support tokens.
>
> From this discussion, the key difference between globally and locally optimal tokens is that globally optimal tokens require their token score to be higher than that of all tokens, while locally optimal tokens only require this condition for a specific subset of tokens. This implies that both definitions depend on $v$, as the token score is defined in terms of it. However, globally optimal tokens exhibit a stronger dependency on $v$. Consequently, while globally optimal tokens have a greater dependency on $v$, they inherently satisfy the conditions for locally optimal tokens.

---

> ### Author Response · Authors · 2024-11-27
> **Response to Reviewer kVFQ--Part 3**
>
> > **Question 3:** The model sizes do not seem to be specified. The architectural details should be put in the appendix.
>
> **Response:** Thank you for your feedback.
>
> **We have added the model architecture details for both the semantic analysis model in Appendix H4.** The architecture for the model used to perform the semantic analysis task on the Stanford IMDb Movie Review dataset follows the transformer encoder architecture [[Vaswani et al.](https://arxiv.org/abs/1706.03762)], with a linear classifier as the last layer.
>
> The embedding layer has trainable token embedding $E$ and position encoding $P$. The model's vocabulary size is $30522$ with maximum token length of $512$ and embedding dimension $384$, so $E\in\mathbb{R}^{30522\times384}$ and $P\in\mathbb{R}^{512\times384}$. If a token $t$ is in position $i$, its embedding will be $X_i=E_t+P_i$, where $E_t$ and $P_i$ denote the $t^{th}$ and $i^{th}$ rows of $E$ and $P$, respectively.
>
> Then, the token features are passed through the encoding blocks, each of which consists of a multi-head self-attention layer $\text{MultiHead}$, two layer normalization layers $\text{LayerNorm}_1$ and $\text{LayerNorm}_2$, and a Multilayer Perceptron layer (MLP) layer $\text{MLP}$. If the sequence of input token features for the encoding block are $X_1,...,X_T$, and if we denote $\text{MultiHead}(X_1,...,X_T)_i$ as the $i^{th}$ token feature from the multi-head self-attention, the output of the encoding block for the $i^{th}$ token is $\text{LayerNorm}_2(X_i’+\text{MLP}(X_i’))$, where $X_i’=\text{LayerNorm}_1(X_i+\text{MultiHead}(X_1,...,X_T)_i)$.
>
> We experimented with having 3 encoding blocks with 3 attention heads each, 4 encoding blocks with 4 attention heads each, and 6 encoding blocks with 6 attention heads each. We then pass the feature vector of the first [CLS] token from the last encoding layer into a final linear classifier layer. For training this model, we applied a dropout of $0.2$.
>
> **For the newly added experiments using the Vision Transformer (ViT) architecture for the CIFAR-10 classification task,  the model details are provided in Appendix H5**. Specifically, we ran an additional experiment to compare the $\ell_{1.1}$-MD training algorithm with the Adam algorithm. The ViT architecture consisted of a patch size of $4$, $512$-dimensional token features, $6$ attention blocks with $8$ attention heads per block, and a two-layer GeLU network for the final classification layer on the [CLS] patch token, with a hidden layer size of $512$. The embedding layer followed the implementation of [[Dosovitskiy et al.](https://arxiv.org/abs/2010.11929)], learning [CLS] token embeddings, a linear map for image patches, and positional embeddings for patch positions. Key attention layer details mirrored the Stanford IMDb model, with layer normalization added before multihead attention and MLP layers, and a $0.1$ dropout during training.
>
> The experiment results show that $\ell_{1.1}$-MD matches Adam's test accuracy (Fig. 11), achieving SOTA performance with added benefits of sparsity and interpretability (Fig. 12).
>
>
> **Minor remarks and typos**
>
> **Response:** Thank you for your remarks. Most concerns have been addressed in the revised version, with a few exceptions:
>
> - For L71, $W(k)$ has been introduced in the second paragraph of the introduction, though we have revised that part to be clearer.
>
> - In L176, we respectfully disagree because even when the pre-softmax are large, $(\sigma(X_iWz_i))_{\alpha_i}$ can still be significantly larger than the other components $(\sigma(X_iWz_i))_t$, for all $t\neq\alpha_i$.
>
> Specifically, even when the norm of $W$ is large, when we fulfill the condition $(X_{i\alpha_i}-X_{it})^\top Wz_i\geq1$ for all $t\neq\alpha_i$ as it is in the paper, we would have $(X_iWz_i)_{\alpha_i}-(X_iWz_i)_t\geq1$. Per the definition of the softmax function, if we denote $s=\sigma(X_iWz_i)$, then for any $t'\in[T]$, we have
>
> $$s_{t'}= ( \text{exp}((X_iWz_i)_{t'}) ) / ( \text{exp}((X_iWz_i)_1) + \text{exp}((X_iWz_i)_2) + \cdots + \text{exp}((X_iWz_i)_T) ). $$
>
> In the above equation, by substituting $t'$ with $\alpha_i$ and $t$, and dividing $s_{\alpha_i}$ by $s_t$, we get
>
> $$s_{\alpha_i}/s_t=\text{exp}((X_iWz_i)_{\alpha_i}-(X_iWz_i)_t)\geq\text{exp}(1),$$
>
> which implies that
>
> $$s_{\alpha_i}\geq s_t\cdot\text{exp}(1),$$
>
> for all $t\neq\alpha_i$.
>
> - For Appendix G, as discussed in the motivation of this paper, our main goal is to demonstrate that different optimizers have distinct token selection properties, which is the primary focus of Theorems 1–4. We believe that the joint analysis presented in Appendix G is a direct extension of these theorems and builds on the results from Tarzanagh (2023, 2024). To ensure our main contributions (Theorems 1–4) are presented explicitly with sufficient discussion and detail in the main text, we decided to keep the joint analysis in Appendix G. This allows us to maintain clarity and focus on the core contributions of the paper.

---

> > ### Author Response · Authors · 2024-12-02
> >
> > Dear Reviewer,
> >
> > Thank you again for your time in reviewing our paper.
> >
> > If our response has addressed your concerns, we would be grateful if you could re-evaluate our work.
> >
> > If you have any additional questions or comments, we would be happy to have further discussions.

---

> > > ### Comment · Reviewer_kVFQ · 2024-12-02
> > >
> > > Dear authors,
> > >
> > > Thanks for your rebuttal, and I'm sorry for my lack of reactivity.
> > >
> > > > train loss values
> > >
> > > Unless I have missed it, there is no train loss curve in the paper, while this would really broaden the picture of the paper. This paper studies an optimization algorithm that minimizes a function, it would be good to see how it does in practice. Would it be possible to add one next to the new figure 11?
> > >
> > > > Hypotheses
> > >
> > > I am still confused by the phrasing around the added assumption, is it "for all $\alpha_i$, there exists W such that ..." ? Then, in my opinion, this should be written in the text.
> > >
> > > The mild overparameterization hypothesis is quite strong in my opinion, the condition that $d\geq n$ is quite unrealistic in modern scenarios; I would emphasize this more in the text. However, it seems like a standard assumption.

---

> ### Author Response · Authors · 2024-12-04
> **Response to Reviewer kVFQ**
>
> > train loss values
>
> **Response:** We are sorry that we missed your request for the train loss values, we will add them for the final submission of the manuscript. We provide below the training loss for the Adam and $\ell_{1.1}$-MD algorithm when they trained the ViT model on CIFAR-10, as we have described in our [previous rebuttal](https://openreview.net/forum?id=9M5georQ9T&noteId=m9s4Vi7SoU). It is in the form of a table in this response, showing the loss every 50 epochs, but we will graph it in the final version.
>
> | Epochs | Adam | $\ell_{1.1}$-MD |
> | -- | -- | -- |
> | 50 | 0.00306 | 0.00847 |
> | 100 | 0.00091 | 0.00484 |
> | 150 | 0.00051 | 0.00230 |
> | 200 | 0.00038 | 0.00092 |
> | 250 | 0.00028 | 0.00048 |
> | 300 | 0.00022 | 0.00034 |
> | 350 | 0.00017 | 0.00022 |
> | 400 | 0.00018 | 0.00019 |
> | 450 | 0.00014 | 0.00015 |
> | 500 | 0.00013 | 0.00014 |
> | 550 | 0.00011 | 0.00011 |
> | 600 | 0.00009 | 0.00007 |
> | 650 | 0.00010 | 0.00006 |
> | 700 | 0.00007 | 0.00007 |
> | 750 | 0.00008 | 0.00006 |
> | 800 | 0.00008 | 0.00006 |
> | 850 | 0.00008 | 0.00005 |
> | 900 | 0.00008 | 0.00005 |
> | 950 | 0.00005 | 0.00006 |
> | 1000 | 0.00008 | 0.00003 |
>
> > Hypotheses
>
> **Response:** We thank the reviewer for pointing out the need for greater clarity regarding the added assumption. To address this concern and ensure the assumption is explicitly stated, we have now included it in the manuscript using a formal mathematical assumption environment, as follows:
>
> "
>
> **Assumption B** [Assumption on Token Separability]
> For each input sequence $X_i$ and its locally optimal token index $\alpha_i$, there exists a matrix $W$ such that, for all non-optimal tokens $t \in [T] \setminus \{\alpha_i\}$,
> $$
> (X_{i\alpha_i} - X_{it})^\top W z_i \geq 1, \quad \forall i \in [n].
> $$
>
> "
>
> We will add this assumption into the statements of Theorems 1–5 to make the assumption more explicit.
>
> Regarding the overparameterization assumption, we respectfully disagree, as various forms of this assumption--where the number of trainable parameters exceeds the size of the dataset--are commonly employed across the optimization literature   [[Tarzanagh et al.](https://arxiv.org/abs/2308.16898), [Azizan et al.](https://arxiv.org/abs/1906.03830), [Allen-Zhu et al.](https://proceedings.mlr.press/v97/allen-zhu19a.html)]. Specifically, our assumption is identical to that in [Tarzanagh et al.](https://arxiv.org/abs/2308.16898), which discusses overparameterization for attention models further in detail (Refer to Sections 4.1 and 4.2 of that paper).

---

### Official Review · Reviewer_HzdV · 2024-11-08

**Soundness:** 3
**Presentation:** 3
**Contribution:** 3
**Rating:** 6
**Confidence:** 3

**Summary:**

The paper introduces a novel approach to optimizing attention mechanisms using Mirror Descent (MD), specifically focusing on a generalized max-margin token selection strategy for softmax attention models. The authors propose a family of MD algorithms, termed $\ell_{p}$-AttGD, which generalize traditional gradient descent by using the $p$-th power of the $\ell_p$-norm as the potential function. The main contributions include proving the convergence of these algorithms to generalized max-margin Support Vector Machine (SVM) solutions for optimizing the attention mechanism, both for fixed and jointly optimized parameters (key-query matrix and decoder).

**Strengths:**

1. Theoretical Contributions: The paper provides a solid theoretical foundation for understanding the convergence properties and implicit bias of MD in attention models. The extension to $\ell_p$-norm objectives adds flexibility in modeling and opens up new avenues for optimizing attention mechanisms.

2. Generalization of Attention Optimization: The approach generalizes previous work on attention models by using MD with a broad class of potential functions, allowing a deeper exploration of the optimization landscape beyond vanilla gradient descent.

**Weaknesses:**

1. As far as I know, mirror descent is not a popular optimization algorithm for training deep learning models. I agree that a simplified model (e.g., the one layer model considered in this paper and previous work) could provide valuable insights on understanding transformers, but it is not clear what is the role/implication the $\ell_p$-norm for deep learning. If possible, it would helpful if the authors could highlight a few practical mirror descent-based optimizers in the revision.

2. For Line431, for the $\ell_{1,1}$/$\ell_3$ optimizer, the authors provide code for implementing the $\ell_{1,1}$/$\ell_3$ optimizer for optimizing the transformer. In term of efficiency, would the $\ell_{1,1}$/$\ell_3$ optimizer be as efficient as popular optimizers like AdamW [LH2019]? Could the authors provide comparison with AdamW in Table 1?

[LH2019] Decoupled Weight Decay Regularization. Ilya Loshchilov, Frank Hutter.

**Questions:**

1. What does the comparison token $z_i$ mean in practice (Line 55)? For example, for a real-world application problem, how to set the comparison token $z_i$ given the input $X$ and label $y$?

---

> ### Author Response · Authors · 2024-11-27
> **Response to Reviewer HzdV--Part 1**
>
> > **Weakness 1:** As far as I know, mirror descent is not a popular optimization algorithm for training deep learning models. I agree that a simplified model (e.g., the one-layer model considered in this paper and previous work) could provide valuable insights on understanding transformers, but it is not clear what is the role/implication of the $\ell_p$-norm for deep learning. If possible, it would be helpful if the authors could highlight a few practical mirror descent-based optimizers in the revision.
>
> **Response:** We appreciate the reviewer’s insightful comment regarding the role and implications of MD and its practical relevance for deep learning. Below, we address these points in detail.
>
>
> - Our work demonstrates that **different optimization algorithms yield distinct solutions when training attention-based models**. This observation highlights the potential for exploring alternative optimization techniques specifically tailored for attention mechanisms. The implicit biases introduced by these algorithms, such as MD, can influence token selection properties, ultimately impacting generalization and interpretability. By showcasing these distinctions, our work paves the way for developing and understanding optimization algorithms that better align with the unique demands of attention layers.
>
>
> - While **MD is not yet a widely adopted method for training transformers, it has been extensively studied for its implicit bias properties in standard deep neural networks**; see Section 1.2 in [[Sun et al.](https://www.jmlr.org/papers/v24/23-0836.html), [Azizan et al.](https://arxiv.org/abs/1906.03830), [Azizan et al.](https://openreview.net/pdf?id=HJf9ZhC9FX), [Gunasekar et al.](https://arxiv.org/abs/1802.08246)]. Notably, many adaptive algorithms, including AdaGrad (please refer to Section 1.1 in [[Duchi et. al](https://www.jmlr.org/papers/volume12/duchi11a/duchi11a.pdf)] regarding connection between AdaGrad and MD), can be viewed as variants of MD with different potential functions. This connection underscores the broader applicability of MD-inspired methods in attention optimization. Our work provides a framework for understanding how MD with $\ell_p$-regularization influences attention-based models, which could inform future designs of adaptive optimizers leveraging MD principles for better interpretability and generalization.
>
>
> - To provide additional practical context, we have included new experiments (Appendix H5) **comparing $\ell_{1.1}$-MD with Adam on Vision Transformers**. These experiments (Figure 11) demonstrate that $\ell_{1.1}$-MD achieves comparable test accuracy to Adam while offering improved explainability. Specifically, as discussed in **response to Weakness 2**, we provide detailed qualitative (Figures 8 and 13) and quantitative (Figure 9 and 12) evaluations of the explainability benefits of $\ell_{1.1}$-MD. Specifically, $\ell_{1.1}$-MD produces sparser weight distributions and sharper attention maps, focusing on more critical tokens, which aligns with its potential to enhance both interpretability and performance in real-world scenarios; Please refer to our detailed response to Weakness 2 and discussion on explainability of MD.
>
> > **Question 1:** What does the comparison token $z_i$ mean in practice (Line 55)? For example, for a real-world application problem, how to set the comparison token $z_i$ given the input $X$ and label $y$?
>
> **Response:** Thank you for the question. In practice, there are two common ways to use attention: self-attention and cross-attention.
>
> In cross-attention, $z_i$ is typically a token in the decoder module of the model, which is usually responsible for generating the output sequence in a sequence-to-sequence model, while $X_i$ is the sequence of input tokens to the model. We did not use cross-attention in our experiment on real datasets, but we did do so for the synthetic dataset experiment, where $z_i$ is randomly generated, independently from $X_i$.
>
> Similarly, in self-attention, the token $z_i$ can represent any individual token or all tokens within the sequence $X_i$. In our experiment on the Stanford Movie Review dataset classification task and the CIFAR-10 classification task (vision transformer), $z_i$ is replaced with a matrix and it equals $X_i$.
>
> We added this detail in Appendix H4.

---

> ### Author Response · Authors · 2024-11-27
> **Response to Reviewer HzdV--Part 2**
>
> > **Weakness 2:** For Line 431, for the $\ell_{1,1}/\ell_3$ optimizer, the authors provide code for implementing the $\ell_{1,1}/\ell_3$ optimizer for optimizing the transformer. In terms of efficiency, would the $\ell_{1,1}/\ell_3$ optimizer be as efficient as popular optimizers like AdamW [LH2019]? Could the authors provide a comparison with AdamW in Table 1?
>
> **Response:** Thank you for your comment. As this was a common question among reviewers, we considered providing a comparison with standard Adam from the adaptive momentum family of algorithms: Specifically, we provide a comparison between the best performing MD algorithm in our paper, $\ell_{1.1}$-MD, to the conventional Adam algorithm. We have added an experiment in Appendix H5 to further validate the performance of $\ell_{1.1}$-MD. This experiment involves training a Vision Transformer (ViT) model with $\ell_{1.1}$-MD and Adam over the first 1000 epochs. The results in Figure 11 shows that $\ell_{1.1}$-MD achieves similar test accuracies to Adam, demonstrating that it can match the performance of state-of-the-art optimizers for transformer models.  This finding highlights that $\ell_{1.1}$-MD is not only competitive but also offers additional advantages, particularly improved explainability, as elaborated below.
>
> Explainability in attention mechanisms is often defined as the model’s ability to identify and prioritize the most influential tokens in the input sequence, thereby making its decision-making process more interpretable [[Klein et al.](https://arxiv.org/abs/2409.16756), [Ali et al.](https://arxiv.org/abs/2403.01590), [Abnar et al.](https://arxiv.org/abs/2005.00928)] . This aligns with the concept of feature selection in classical machine learning, where sparse and focused representations improve both interpretability and model robustness. In our work, $\ell_{1.1}$-MD demonstrates superior explainability compared to other traditional optimization methods. Specifically, $\ell_{1.1}$-MD produces sparser weight distributions and attention maps that more sharply highlight the most critical tokens. Figures 8 and 13 provide clear evidence of this property in the Stanford Large Movie Review Dataset. For instance, attention maps generated by $\ell_{1.1}$-MD-trained models focus more on sentiment-revealing words, such as "amazing" or "terrible," while models trained with GD display more diffuse attention patterns, potentially diluting interpretability. This ability to emphasize pivotal tokens directly contributes to the model's transparency and aligns with established literature emphasizing the importance of sparsity for interpretability [[Sun et al.](https://www.jmlr.org/papers/v24/23-0836.html), [Azizan et al.](https://arxiv.org/abs/1906.03830), [Azizan et al.](https://arxiv.org/pdf/1806.00952)].
>
> Furthermore, the weight distributions in the key, query, and value matrices, shown in Figure 9, highlight that $\ell_{1.1}$-MD encourages sparsity more effectively than GD, while the weight distributions in Figure 12 shows that $\ell_{1.1}$-MD also induces more sparsity compared to Adam. This sparsity enhances interpretability by limiting the model's reliance on non-essential tokens. By aligning the optimization process with explainability objectives, $\ell_{1.1}$-MD offers practical benefits for applications where transparency is crucial [[Klein et al.](https://arxiv.org/abs/2409.16756), [Ali et al.](https://arxiv.org/abs/2403.01590), [Abnar et al.](https://arxiv.org/abs/2005.00928)]. Thus, **while $\ell_{1.1}$-MD achieves comparable generalization performance to Adam (as demonstrated in Figure 11), its token selection precision and sparser representations establish it as a more interpretable and explainable optimization method (as shown in Figures 8, 9, 12, and 13)**. These findings underscore the potential of using MD variants to improve both the performance and explainability of attention-based models.

---

> > ### Comment · Reviewer_HzdV · 2024-11-28
> > **Response to rebuttal**
> >
> > Thank the authors for the rebuttal. I do not have any further questions at this point.

---

### Author Response · Authors · 2024-11-27
**General Response**

We thank the reviewers for their valuable feedback, which greatly improved our manuscript. Below, we summarize the main contributions **(C1–C3)**, the reviewers' key points **(P1–P4)**, and our actions **(A1–A4)**. Revisions and **new experiments** are highlighted in blue in the paper.

**C1. Implicit Bias Analysis with Mirror Descent (MD):** We provide a comprehensive theoretical analysis of MD for softmax attention mechanisms, introducing the $\ell_p$-AttGD framework. This approach demonstrates directional convergence to generalized hard-margin SVM solutions with $\ell_p$-norm objectives and extends prior work limited to gradient descent.
 Reviewer kVFQ noted: "The fact that MD achieves sparser weights and leads to better generalization is interesting." This highlights the significance of analyzing MD's implicit bias and token selection properties.

**C2. Different Optimizers and Their Token Selection:**  Our motivation is to demonstrate that **different optimizers have different token selection properties**, which in turn leads to different generalization performances.  This observation, missing in the attention optimization literature (Tarzanagh et al., 2023, 2024; Vasudeva et al., 2024a; Sheen et al., 2024), is validated by analyzing $\ell_p$-AttGD convergence to $\ell_p$-SVM under different $\ell_p$-norm objectives within the nonconvex softmax framework. Reviewer HzdV stated: " The paper provides a solid theoretical foundation for understanding the convergence properties and implicit bias of MD in attention models. The extension to $\ell_p$-norm objectives adds flexibility in modeling and opens up new avenues for optimizing attention mechanisms.”

**C3. Empirical Insights on Generalization:** Through experiments on synthetic and real-world datasets (e.g., the Stanford Large Movie Review Dataset), we demonstrate that MD algorithms improve token selection and generalization compared to GD. Furthermore, $\ell_{1.1}$-AttGD exhibits superior sparsity and focus, offering improved token selection capabilities. Reviewer 7Bsg mentioned: "The fact that the real-data experiments show some improvements in generalization is noteworthy," validating the practical value of our empirical contributions.

**P1. Practical Relevance and Role of MD:** Several reviewers questioned the practicality of MD, emphasizing its limited usage in deep learning models (Reviewers HzdV and kVFQ). Reviewer HzdV requested examples of MD-based optimizers, while Reviewer kVFQ emphasized the need for clearer motivation for exploring MD in attention training.

**P2. Comparison to Adam and Architecture Details:** Reviewers requested comparisons between MD and popular optimizers like AdamW (Reviewers HzdV and vf65). Reviewer kVFQ also suggested including model architecture details in the appendix for reproducibility.

**P3. Feasibility and Assumptions of $\ell_p$-AttSVM:** Reviewers kVFQ and vf65 raised concerns about feasibility conditions in the $\ell_p$-AttSVM problem and requested explicit clarification of assumptions related to initialization and step size.

**P4. Broader Experimental Scope and Evaluation:** Reviewers vf65 and 7Bsg noted that the synthetic experiments are overly simplistic and called for more diverse datasets to support the claims. Additionally, quantitative evaluations of token selection and training loss values were suggested for real-data experiments.

**A1. Clarified Motivation and Practical Relevance of MD:** We revised the introduction to emphasize the novel insights provided by MD, including its token selection properties and implicit bias. Further, examples of MD-based optimizers, such as those used in convex optimization and reinforcement learning, have been added.

    Further details are provided in the response to Reviewers HzdV and kVFQ.

**A2. Comparisons with Adam and Architecture Details:** We added new experiments to compare the test accuracy and weight distribution of Adam and $\ell_{1.1}$-MD on training a Vision Transformer model to learn the CIFAR-10 dataset, to show a competitive performance and a superior explainability of MD, as shown **in the newly added Figures 11 and 12 in Appendix H5**.

    Further details are provided in the response to Reviewers kVFQ, vf65, and HzdV.

**A3. Feasibility and Assumptions in $\ell_p$-AttSVM:** We explicitly stated the feasibility conditions and assumptions (e.g., overparameterization $d \geq \max(T-1, n)$) in the main text. These conditions were also discussed in the context of Theorem 1 and other results.

    Further details are provided in the response to Reviewers kVFQ and vf65.

**A4. Broader Experiments and Quantitative Evaluation:** To strengthen the empirical results, we conducted additional experiments on tasks like sentiment classification with more diverse datasets. Quantitative metrics for token selection accuracy and training loss values have been incorporated.

    Further details are provided in the response to Reviewers vf65 and 7Bsg.

---

### Author Response · Authors · 2024-12-04

We appreciate the reviewers' feedback, which has greatly improved our paper.

We understand that the response period has concluded and sincerely hope our responses have addressed all concerns. If so, we kindly request a re-evaluation of our work.

Best,

Authors

---

### Note · Authors · 2025-01-19

I have read and agree with the venue's withdrawal policy on behalf of myself and my co-authors.